# A mechano-osmotic feedback couples cell volume to the rate of cell deformation

**Larisa Venkova[1,2†], Amit Singh Vishen[3], Sergio Lembo[4], Nishit Srivastava[1,2], Baptiste Duchamp[1,2], Artur Ruppel[5], Alice Williart[1,2], Stéphane Vassilopoulos[6], Alexandre Deslys[1,2], Juan Manuel Garcia Arcos[1,2], Alba Diz-Muñoz[4], Martial Balland[5], Jean-François Joanny[3], Damien Cuvelier[1,2,7], Pierre Sens[3]\*, Matthieu Piel[1,2]\***

[1]Institut Curie, PSL Research University, CNRS, UMR 144, Paris, France; [2]Institut Pierre Gilles de Gennes, PSL Research University, Paris, France; [3]Institut Curie, PSL Research University, CNRS, UMR 168, Paris, France; [4]Cell Biology and Biophysics Unit, European Molecular Biology Laboratory, Heidelberg, Germany; [5]Laboratoire Interdisciplinaire de Physique, Grenoble, France; [6]Sorbonne Université, Institut National de la Santé et de la Recherche Médicale, Institut de Myologie, Centre de Recherche en Myologie, Paris, France; [7]Sorbonne Université, INSERM, Paris, France

**Abstract** Mechanics has been a central focus of physical biology in the past decade. In comparison, how cells manage their size is less understood. Here, we show that a parameter central to both the physics and the physiology of the cell, its volume, depends on a mechano-osmotic coupling. We found that cells change their volume depending on the rate at which they change shape, when they spontaneously spread or when they are externally deformed. Cells undergo slow deformation at constant volume, while fast deformation leads to volume loss. We propose a mechanosensitive pump and leak model to explain this phenomenon. Our model and experiments suggest that volume modulation depends on the state of the actin cortex and the coupling of ion fluxes to membrane tension. This mechano-osmotic coupling defines a membrane tension homeostasis module constantly at work in cells, causing volume fluctuations associated with fast cell shape changes, with potential consequences on cellular physiology.

**\*For correspondence:**
pierre.sens@curie.fr (PS);
matthieu.piel@curie.fr (MP)

**Present address:** [†]Institute of Biochemistry and Cellular Genetics, University of Bordeaux, Paris, France

## Editor's evaluation

The paper by Venkova et al. is a comprehensive study of mammalian cell volume dynamics during the common cellular process of adhesion and spreading on a flat substrate, osmotic changes, and mechanical confinement. The paper reveals a complex interplay between cell water/ion regulation, cytoskeletal processes, and mechanical deformation of the cell. The topic is important in cell physiology and should be of considerable interest to cell biologists, mechanobiologists and biophysicists.

## Introduction

In recent years, in vivo imaging has revealed that, in a variety of physiological and pathological contexts, cells undergo large deformations (*Weigelin et al., 2014*), sometimes being squeezed to a tenth of their resting diameter. Migrating cells, in particular fast-moving immune or cancer cells, can deform to a large extent in only a few minutes (*Thiam et al., 2016*; *Vargas et al., 2017*; *Beunk et al., 2019*), for example, when they cross an endothelial barrier (*Subramanian et al., 2020*). Even faster deformations, below the second timescale, can be observed in circulating cells pushed through small blood capillaries. Altogether, these examples show that large cell deformations are physiological and

occur across a large range of timescales. Large cell shape changes must involve significant changes in volume, surface area, or both. But the number of studies on cell volume modulation upon cell deformation is still very small (*Guilak, 1995*; *Liu et al., 2020*). It is still not clear whether the material that cells are made of is rather poroelastic (*Moeendarbary et al., 2013*), losing volume when pressed, or behaves like a liquid droplet, extending its surface area at constant volume. Two articles, measuring volume using 3D reconstruction from confocal slices, report that cells that are more spread are smaller in volume (*Guo et al., 2017*; *Xie et al., 2018*), leading to a higher density and potential long-term effects on cell fate (*Guo et al., 2017*). On the other hand, another article, using volume measurements by fluorescence exclusion (FXm), reports no or slightly positive correlation between spreading area and cell volume (*Perez Gonzalez et al., 2018*), reflecting the fact that as cells grow larger, their spreading area also increases.

Different models have recently been proposed to explain a coupling between cells shape changes and cell volume modulation (*Guo et al., 2017*; *Xie et al., 2018*; *Adar and Safran, 2020*; *McEvoy et al., 2020*). Most of them are based on the same type of scenario: depending on the timescale and extent of the deformation, cell shape changes can stress the cell surface, including the membrane and the actin cortex (*Chugh and Paluch, 2018*). This stress can be relaxed due to cortex turnover, unfolding of membrane reservoirs (*Pietuch and Janshoff, 2013*), and detachment of the membrane from the cortex with the formation of blebs (*Tinevez et al., 2009*). Stress in these structures can also lead to the modulation of ion fluxes (*Jiang and Sun, 2013*) resulting in cell volume changes. Despite its broad relevance for cell mechanics and cell physiology, the consequences of this type of scenario have not been explored in depth experimentally.

Using FXm to accurately measure volume in live cells (*Zlotek-Zlotkiewicz et al., 2015*; *Cadart et al., 2017*; *Cadart et al., 2018*), we found that when cells deform as they spread on a flat adhesive surface, the degree of volume changes depends on the speed of spreading. To explain this observation, we propose an extension of the classical pump and leak model (PLM; *Cadart et al., 2019*), including a mechano-osmotic coupling activated upon cell deformation occurring faster than the membrane tension/actin cortex relaxation timescale. We further probe our model assumptions and predictions experimentally by characterizing the cell volume response during osmotic shocks and during ultra-fast (ms) mechanical cell deformation, as well as by performing tether pulling experiments on spreading cells to assess their membrane tension. We believe that our observations, together with this novel physical model, constitute strong evidence for the existence of a mechano-osmotic coupling constantly at work in animal cells and modulating their volume as they deform.

## Results

### Cell volume is not correlated to the final steady-state spreading area but significantly decreases during cell spreading

We first asked whether in a population growing and dividing at steady state, cells display a correlation between their spreading area and their volume. We used HeLa cells expressing hGeminin-mCherry, which accumulates in the nucleus during the S phase. Cell spreading area was measured using phase contrast and cell volume using FXm (*Cadart et al., 2017*, *Figure 1A* images). We did not find any strong correlation between spreading area and volume for HeLa hGeminin-mCherry, as well as for HeLa EMBL (Kyoto) and RPE-1 cells measuring their spreading area with reflection interference contrast microscopy (RICM) (*Rädler and Sackmann, 1993*; *Cuvelier et al., 2007*), larger cells in volume being also slightly more spread (*Figure 1A* graph and *Figure 1—figure supplement 1A and B*). A clearer positive correlation was observed for 3T3 fibroblasts, which were also generally more spread for a given volume (*Figure 1—figure supplement 1C*). Using the hGeminin cell cycle marker, we observed that S/G2 cells tend to be larger and more spread than G1 cells (*Figure 1A* graph), suggesting that the positive correlation is simply due to cell growth, with cells increasing their spreading area as they grow. Using live-cell recording of phase, volume, and hGeminin, we also considered cells at given windows of time following cell division, to examine the correlation between volume and surface area at a given cell cycle stage and thus independently of cell growth. Considering the same group of cells at various times after mitosis, or after the G1/S transition, we could not observe any correlation between cell volume and spreading area at any given cell cycle stage (*Figure 1A*). Finally, to extend the range of spreading areas considered, we used adhesive micro-patterns with areas smaller than

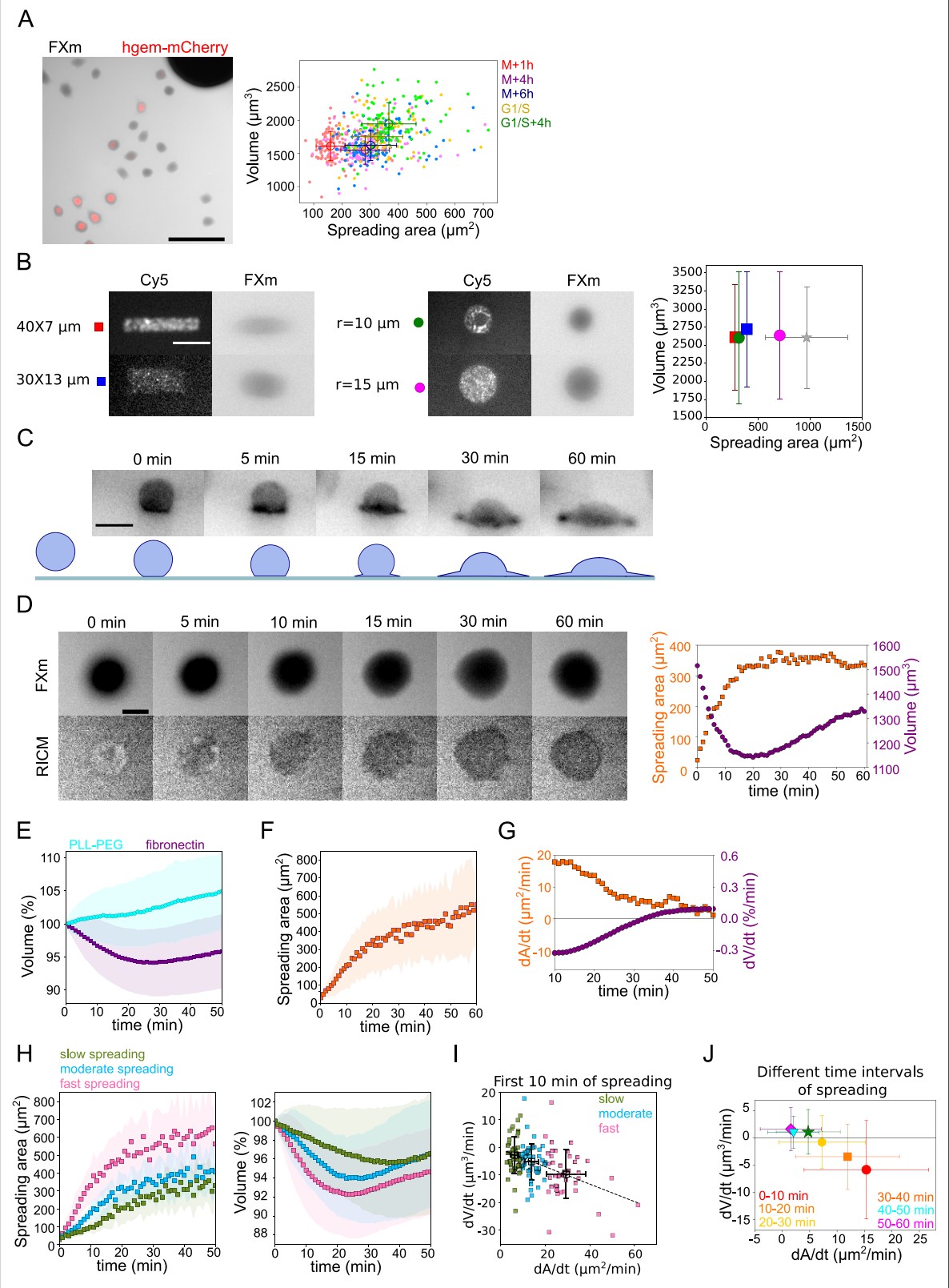

**Figure 1.** Cell volume in spreading cells. (**A**) *Left*: Composite of FXm in GFP channel and fluorescent image in mCherry channel of HeLa hgem-mCherry cells. Scale bar: 100 μm. *Right*: Relation between volume and spreading area of HeLa hgem-mCherry cells at the different cell cycle stages, N=3: M+1h (n=131) correlation coefficient R=0.11, M+4h (n=131) R=0.23, M+6h (n=131) R=0.26, G1/S (n=99) R=0.20, G1/S+4h (n=92) R=0.22. Error bars represent standard deviation. (**B**) *Left*: Typical images of micropatterns and typical images of cells plated on micropatterns. Scale bar: 10 μm. *Right*: Average

*Figure 1 continued on next page*

*Figure 1 continued*

volume of HeLa Kyoto cells plated on the patterns (measurements are done 4 hr after cell plating) of different shape and size in comparison with non-patterned cells. Blue: rectangle 30×13 µm$^2$ (n=131, N=2); red: rectangle 40×7 µm$^2$ (n=214, N=2); purple: circle, r=15 µm (n=338, N=4); green: circle, r=10 µm (n=242, N=3); gray: non-patterned cells (n=286, N=3). Error bars represent standard deviation. There is nos statistical difference between patterned cells and non-patterned cells: rectangle 30×13 p=0.15, rectangle 40×7 µm$^2$ p=0.96, r=15 µm p=0.63, r=10 µm p=0.97. (**C**) *Top*: Side view of a HeLa-Lifeact (black) cell spreading on fibronectin-coated glass. Scale bar: 20 µm. *Bottom*: Scheme of shape transition during cell spreading. (**D**) *Left*: FXm and RICM imaging of a HeLa Kyoto cell spreading on fibronectin-coated glass. Scale bar: 20 µm. *Right*: Volume (red) and spreading area (blue) of cell represented on the left panel. (**E**) Average normalized volume of control HeLa Kyoto cells (blue, n=127, N=3) spreading on fibronectin-coated glass, or plated on PLL-PEG-coated glass (cyan, n=493, N=5). Error bars represent standard deviation. (**F**) Average spreading area of control HeLa Kyoto cells (n=125, N=3), spreading on fibronectin-coated glass. Error bars represent standard deviation. (**G**) Linear derivatives dA/dt (blue) and dV/dt (red) for average spreading area and volume represented on (**F**) and (**E**) for sliding window 10 min. (**H**) Average normalized volume (*left*) and spreading area (*right*) of control HeLa cells divided in three categories based on their initial spreading speed, N=3: slow (n=42), moderate (n=43), fast (n=42). Error bars represent standard deviation. (**I**) Volume flux (dV/dt) of single control HeLa Kyoto cells (n=194, N=3) plotted versus their spreading speed (dA/dt) at the first 10 min of spreading. The data are fitted with linear regression y=−0.31x−0.71, R$^2$=0.19. Error bars represent standard deviation. Color code indicate three groups of cells represented on (**H**). (**J**) Median volume flux (dV/dt) of HeLa Kyoto cells plotted versus median spreading speed (dA/dt) at the different time intervals of spreading (n=194, N=3). Error bars represent standard deviation.

The online version of this article includes the following video, source data, and figure supplement(s) for figure 1:

**Source data 1.** Data tables related to quantifications in Figure 1.

**Figure supplement 1.** Cell volume in spreading cells.

**Figure supplement 1—source data 1.** Data tables related to quantifications in Figure 1 Supplement 1.

**Figure 1—video 1.** Side view of a HeLa-LifeAct cell spreading on fibronectin-coated glass.

https://elifesciences.org/articles/72381/figures#fig1video1

**Figure 1—video 2.** Spreading of HeLa EMBL cells on fibronectin-coated glass.

https://elifesciences.org/articles/72381/figures#fig1video2

the average spontaneous steady-state spreading area of HeLa EMBL (Kyoto) cells (*Figure 1B* images). We found that the distribution of volumes did not change when cells were plated on smaller adhesive patterns (*Figure 1B* graph). Overall, these experiments suggest that, as reported before (*Perez Gonzalez et al., 2018*), there is no strong correlation, at the cell population level, between spreading area and cell volume, independently of the cell cycle stage.

Previous studies also reported volume loss during cell spreading (*Guo et al., 2017*). When plated on a fibronectin-coated substrate, HeLa EMBL cells showed a transition from a sphere to a half-sphere in about 15 min, then continued spreading by extending lamellipodial protrusions (*Figure 1C* and *Figure 1—video 1*). We recorded spreading cells, combining FXm to measure volume and RICM to measure spreading area accurately (*Figure 1D* and *Figure 1—video 2*). RICM images showed an initial spreading phase of about 15±10 min until the radius of the contact region equaled that of the cell, corresponding to a hemispheric cap cell shape, which was followed by an extension of lamellipodial protrusions. Cell spreading was accompanied by a small (5% on average) but significant loss of volume, typically occurring during the first 20 min of spreading and followed by a volume increase of about 5%/hr, in the range of the expected cell growth for a doubling time of about 20 hr (*Figure 1E, F and G*). The same was observed for cells that had been synchronized by serum starvation, but with a smaller standard deviation (*Figure 1—figure supplement 1D*). The absolute precision of our measurements is within 10%, evaluated by comparing the initial average volume of cell populations in different chambers and on different days (*Figure 1—figure supplement 1E*). But the accuracy of the measurement when following the same individual cell is rather 1% (*Figure 1—figure supplement 1F*, evaluated by measuring the volume of the same cell multiple times at a 30 ms frame rate). Combining quantitative phase and volume measurement, we found that only cell volume decreased while dry mass remained constant over the few tens of minutes of initial cell spreading (*Figure 1—figure supplement 1G*), causing a transient density increase (*Figure 1—figure supplement 1H and G*). This suggests a loss of water (and probably small osmolites like ions) from the cell, similar to volume regulatory decrease following a hypo-osmotic shock (*Pedersen et al., 2013*). Cells plated on PLL-PEG, instead of fibronectin, did not spread and displayed an increase in the volume of about 7%/hr (*Figure 1E*). This result, together with our observation on steady-state spread cells, suggests that the spreading dynamics rather than the final spreading area might be coupled to the loss of volume.

## The loss of volume during spreading depends on the spreading speed

Taking advantage of the intrinsic variability in the cell spreading dynamics, we considered single cell volume and spreading trajectories. We observed that individual spreading cells could display a large range of volume loss (*Figure 1—figure supplement 1I*). Pooling cells together according to their spreading speed, we observed that faster-spreading cells were losing more volume, whereas slow-spreading cells lost less volume or did not lose volume at all (*Figure 1H*). To further validate this correlation, we measured the initial spreading speed for the first 10 min of spreading and plotted it against the rate of volume loss, for individual cells (*Figure 1I*). The graph clearly shows that faster spreading cells also lose volume faster in this initial spreading phase. Spreading speed and volume loss are both slowing down with time (*Figure 1J*), whereas absolute spreading area increases (*Figure 1—figure supplement 1J*). Conversely, the amount of volume lost was not correlated with the initial cell volume

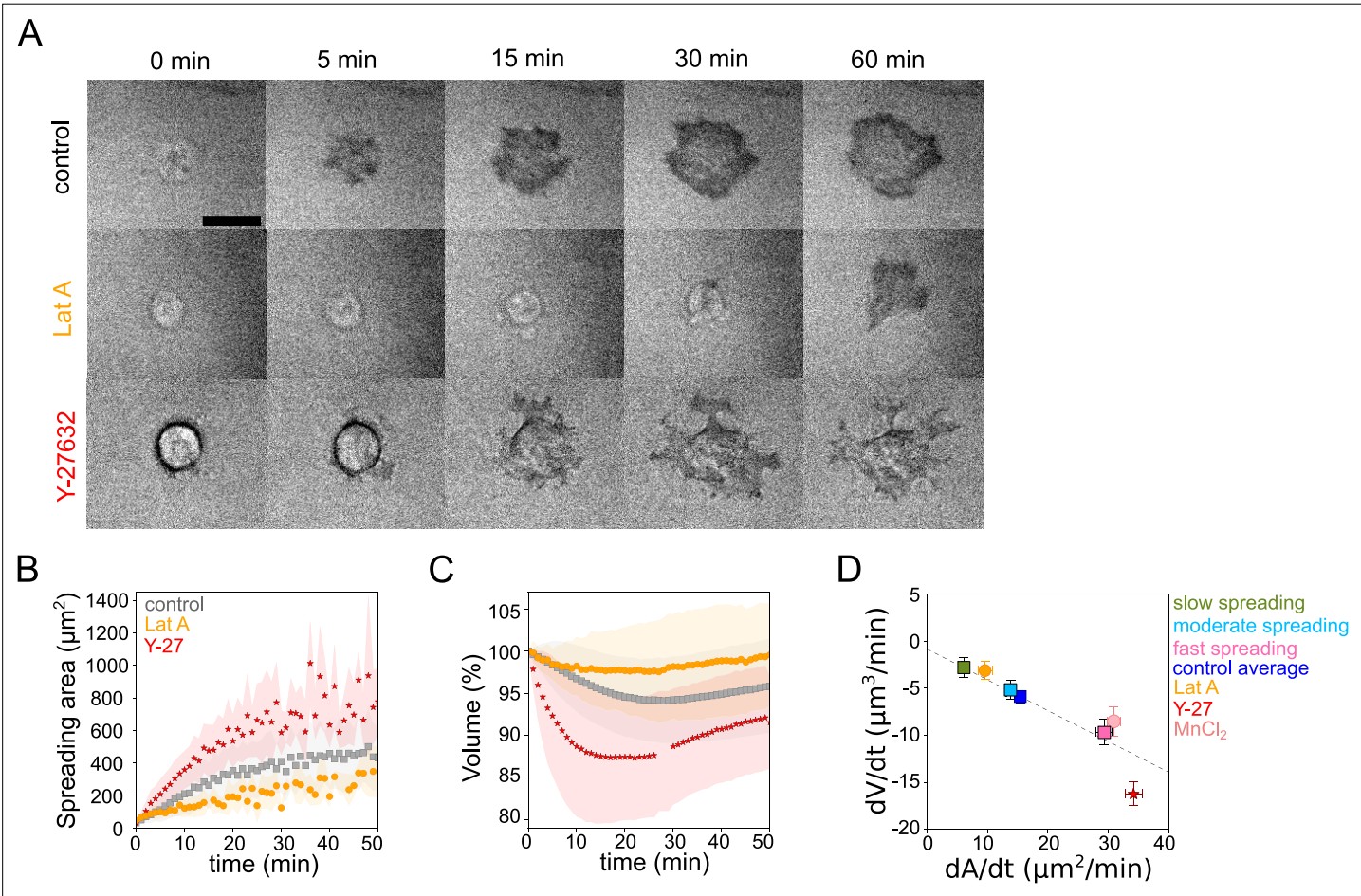

**Figure 2.** Cell volume depends on spreading speed. (**A**) RICM imaging of control HeLa Kyoto cell or cell treated with 100 nM Latrunculin A or with 100 µM Y-27632 spreading on fibronectin-coated glass. Scale bar: 20 µm. (**B**) Average spreading area of control HeLa Kyoto cells (gray, n=125, N=3), cells treated with 100 nM Latrunculin A (orange, n=30, N=2) or 100 µM Y-27632 (red, n=98, N=3). Error bars represent standard error. (**C**) Average normalized volume of control HeLa Kyoto cells (gray, n=125, N=3), cells treated with 100 nM Latrunculin A (orange, n=30, N=2) or 100 µM Y-27632 (red, n=98, N=3). Error bars represent standard error. (**D**) Median volume flux (dV/dt) of control (blue, n=194, N=3), 100 µM Y-27632 (red, n=121, N=4), 100 nM Latrunculin A (orange, n=41, N=3) or 1 mM MnCl$_2$ (N=3, n=57) treated HeLa Kyoto cells plotted versus their spreading speed (dA/dt) at the first 10 min of spreading. Average dV/dt(dA/dt) for three groups of control cells from *Figure 1I* are shown on the graph. The dashed line is a linear regression for control cells from panel 1I. Error bars represent standard error.

The online version of this article includes the following source data and figure supplement(s) for figure 2:

**Source data 1.** Data tables related to quantifications in Figure 2.

**Figure supplement 1.** Cell volume depends on spreading speed, additional cell lines.

**Figure supplement 1—source data 1.** Data tables related to quantifications in Figure 2 Supplement 1.

(*Figure 1—figure supplement 1K*). Overall, these data show that volume loss in spreading cells is a transient phenomenon correlated with the spreading kinetics and not the absolute spreading area.

Early spreading dynamics were shown to strongly depend on the properties of the actomyosin cortex (*Cuvelier et al., 2007*). Hence, we affected F-actin with a low dose (100 nM) of Latrunculin A (Lat A) which still allowed cell spreading, and myosin with the 100 µM ROCK inhibitor Y-27632 (Y-27, *Figure 2A*). As expected, we found that Lat A-treated cells spread slower, while Y-27-treated cells spread faster than control cells (*Figure 2B*). Accordingly, Lat A treated cells lost less volume (2–3%) than control cells, while Y-27 treated cells lost more (10–15%, *Figure 2C and D*, *Figure 2—figure supplement 1A*). Y-27 treated cells plated on PLL-PEG substrate on which they could not spread, increased their volume like control cells (*Figure 2—figure supplement 1B*), thereby showing that larger volume loss was not due to the drug treatment itself but was a result of the spreading kinetics in the presence of the drug. This coupling between spreading speed and volume loss was also found to be very similar for other cell types, RPE-1 and 3T3 (*Figure 2—figure supplement 1C-I*), although RPE-1 cells displayed an initial phase of volume increase before eventually losing volume (*Figure 2— figure supplement 1E, F, H, I*), a phenomenon that we have not investigated further in this article. This initial phase of volume increase was lost upon Y-27 treatment (*Figure 2—figure supplement 1F*, *Figure 2—figure supplement 1I*), suggesting that it was due to induction of contractility through mechanotransduction pathways (*Burridge et al., 2019*). In order to increase spreading speed without changing cell contractility, we added 1 mM $MnCl_2$ during the spreading experiment. This treatment increased spreading speed, as previously reported (*Edwards et al., 1988*) and also increased the volume loss (*Figure 2—figure supplement 1J* and *Figure 2D*). These data together suggest a general effect of spreading speed on volume modulation, with a loss of volume reaching up to 20% for fast-spreading cells.

## The classical pump and leak model describes properly the osmotic response of non-spreading cells but cannot explain the water loss during spreading

Because the observation of a coupling between volume modulation and spreading speed was not reported before, we asked whether classical volume modulation models are sufficient to explain it. Our results show that during fast spreading, cells lose more than 10% of their water content. Water loss exceeding 1% is considered to be dominated by osmotic volume regulation (see Appendix 1 and *Cadart et al., 2019*; *Yellin et al., 2018*). Volume set point and large volume modulation such as volume regulatory response following osmotic shocks can be accounted for by the general theoretical framework of the PLM (see Appendix 1 and *Cadart et al., 2019*; *Tosteson and Hoffman, 1960*; *Kay, 2017*). Briefly, the cell volume is determined by an osmotic balance involving the active pumping of specific ions (mainly sodium and potassium) to compensate for the pressure from impermeant solutes in the cell (*Figure 3A*). The PLM has been verified experimentally on several occasions, mostly with indirect methods for cell volume measurements *Kay and Blaustein, 2019*. We thus decided to check that we could reproduce these results with our cells. We performed series of osmotic shock experiments using PEG400 or distilled water, while recording cell volume by FXm (*Figure 3B* and *Figure 3— figure supplement 1A, B* and *Figure 3—video 1*). Cells showed the expected response to both hypo and hyperosmotic shocks, with a fast change in volume (less than a minute timescale) followed by a slower adaptation (timescale of minutes) (*Figure 3C* and *Figure 3—figure supplement 1C*). We also checked, using quantitative phase measure of dry mass, that these fast changes in volume were not accompanied by any change in dry mass and thus corresponded to water (and ion) fluxes, as expected (*Figure 3—figure supplement 1D*). Because of timescale separation between water flux in the seconds timescales and active ion transport, which takes minutes, upon an osmotic shock, cells first display a passive response corresponding to water fluxes, followed by a slower response due to ion exchanges. The Ponder relation (*Ponder and Saslow, 1931*), which relates the relative change in cell volume right after the shock (at timescale of seconds), to the relative difference of osmotic pressure imposed experimentally, corresponds to the passive cell response. Ponder's plot showed a very good agreement with previous reports (*Pritchard and Guilak, 2004*; *Zhou et al., 2009*), with a linear relation between the change in volume and the change in osmotic pressure, over a large range of imposed external osmolality (*Figure 3D*) and corresponds to about 30% of osmotically inactive volume (volume occupied by large molecules or solid components). As shown by others (*Pritchard*

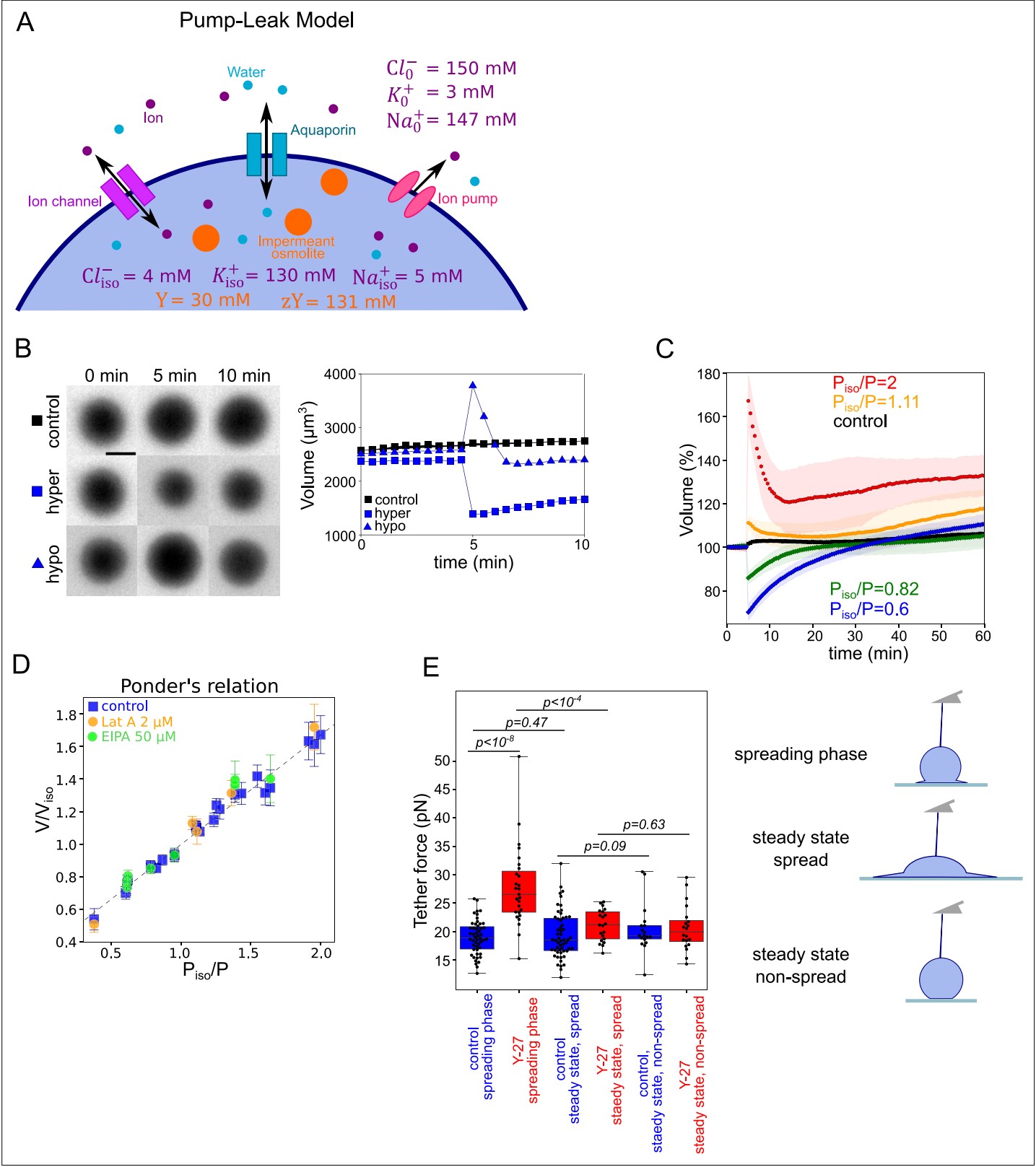

**Figure 3.** Verifying the Pump and Leak Model and its mecano-sensitive extension. (**A**) Schematic of "pump-leak" model (PLM). In brief, the plasma membrane let ions and water pass, with specific channels which increase their permeation coefficient (the 'leak'). Ions can also be pumped out of the cell (the 'pump'). Outside the cell, the concentration of ions is about 300 mM, while it is only of about 150 mM inside the cell. The quantity of ion species also differs, with more anions outside the cell (because proteins are on average negatively charged), more sodium outside the cell, and more potassium

*Figure 3 continued on next page*

*Figure 3 continued*

inside. To achieve the equilibrium, sodium and chloride need to be constantly pumped outside of the cell. The cell also contains nonpermeant osmolytes: charged osmolytes ('zY' at the scheme) and neutral osmolytes ('Y' at the scheme), for example, proteins, amino-acids, and sugars (which cannot pass the membrane at the same rate as water or ions). Considering these equilibria and the resulting osmotic balance, the PLM predicts the volume of the cell (see the model in Appendix 1 and *Cadart et al., 2019*). Numbers at the panels are taken from the model in Appendix 1. (**B**) *Left*: FXm images of HeLa Kyoto cells exposed to media exchange of same osmolarity, hypertonic and hypotonic. *Right*: Volume of the cells represented on the left panel. (**C**) Average normalized HeLa Kyoto cells volume response to osmotic shocks of different magnitudes acquired every 30 s. Number of cells in the experiments: control $P_{iso}/P=1$ (n=51, N=1), $P_{iso}/P=1.11$ (n=30, N=1), $P_{iso}/P=2$ (n=17, N=1), $P_{iso}/P=0.82$ (n=33, N=1), $P_{iso}/P=0.6$ (n=67, N=1). (**D**) Ponder's relation for control HeLa Kyoto cells (blue), treated with 2 µM Lat A (orange), or 50 µM EIPA (green). Each point represents average value of single experiment. Average number of cells in each experiment n~58. Error bars represent standard deviation. Dashed line represents linear regression fit for control cells y=0.67x+0.33, $R^2$=0.98. Coefficient 0.67 refers to the ratio of osmotically active volume to total volume named 'R' in the Appendix 1. (**E**) Tether force measurements of control HeLa Kyoto cells (blue) and treated with 100 µM Y-27632 (red) at the 'spreading phase' (measurements are performed within 30–90 min after cell plating), 'steady state, spread' (measurements are performed within 4–5 hr after plating) or 'steady state, non-spread' (measurements are performed within 4–5 hr after plating on 20 µm diameter micropatterns). For 'control, spreading phase' n=50, N=6; for 'Y-27, spreading phase' n=27, N=3; for 'control, steady state, spread' n=55, N=9; for 'Y-27, steady state' n=21, N=3, for 'control, steady state, non-spread' n=18, N=3, for 'Y-27, steady state, non-spread' n=20, N=3. Error bars represent standard deviation. The results of statistical tests are shown at the graph.

The online version of this article includes the following video, source data, and figure supplement(s) for figure 3:

**Source data 1.** Data tables related to quantifications in Figure 3.

**Figure supplement 1.** Osmotic response of HeLa cells and effect of contractility on cell volume.

**Figure supplement 1—source data 1.** Data tables related to quantifications in Figure 3 Supplement 1.

**Figure 3—video 1.** FXm imaging of HeLa EMBL cells attached on PLL-coated glass exposed to osmotic shock 20× LD.

https://elifesciences.org/articles/72381/figures#fig3video1

---

*and Guilak, 2004*), we find that the Ponder relation does not depend on the integrity of the actin cytoskeleton, as cells treated with 2 µM Lat A show the same relation. Ponder relation does not depend either on the inhibition of sodium/proton exchanger NHE1 by 50 µm EIPA (*Figure 3D*). These experiments also allowed us to estimate the bulk modulus of the cells defined as $B = \frac{V_{iso}\Delta P}{\Delta V}$, where $V_{iso}$ is the volume in isosmotic state, $\Delta V$ is the volume change, induced by the osmotic pressure difference $\Delta P$ (order of GPa, *Figure 3—figure supplement 1E*), which is in good agreement with previous measurements (*Guo et al., 2017*; *Monnier et al., 2016*). These results show both that our cell volume measurements are accurate, even for small volume changes, and that our cells display the expected response to osmotic shocks, explained by the classical PLM, in agreement with previously published results (*Ponder and Saslow, 1931*; *Pritchard and Guilak, 2004*; *Zhou et al., 2009*; *Roffay et al., 2021*).

The classical PLM does not account for the cell shape changes and mechanics. Several additional mechanisms have been proposed to account for the coupling between cell shape and cell volume. A recent model proposed a direct extension of the PLM to account for cell spreading, by including the assumption that channels and pumps are working differently on the adhered and the free surface of the cell (*Adar and Safran, 2020*). Nevertheless, such a model does not predict an effect of the spreading speed on volume, but rather an effect of the spreading area itself, while our data suggest that the opposite is true in our experiments. The correlation of volume loss with spreading speed suggests that the effect on volume could be due to a change in the mechanical state of the cell surface (membrane and/or cortex). A contribution of contractility to volume regulation has been proposed before (*Tao and Sun, 2015*). To test whether cell contractility could directly affect the cell volume, we recorded the volume of the same non-spreading cells, plated on PLL, before and after treatment with the contractility inhibitor (Rho-associated protein kinase ROCK inhibitor) Y-27632 (*Figure 3—figure supplement 1F*). We did not observe any significant difference between a change of medium with a control medium and with a medium containing the drug. This shows that Y-27632 has an effect on volume only during cell spreading. To confirm that Y-27632 treatment during cell spreading reduces traction forces (and thus cortical tension), we also performed traction force microscopy (*Sabass et al., 2008*) during cells spreading (*Figure 3—figure supplement 1G*). We found that, in control cells, the total traction energy, and thus the contractility, increased during spreading as expected, but that it was not the case in Y-27632 treated cells in which it remained very low throughout spreading (similarly to cells treated for the Arp2/3 inhibitor CK-666 which are spreading slower). Our results (*Figure 2*) show that the inhibition of contractility during spreading increased the volume loss, while if

contractility had a direct effect on volume via the force balance at the cell surface, its inhibition should lead to a larger volume.

These observations are consistent with order-of-magnitude estimates. The osmolarity of the cell is the sum of contribution of impermeant osmolytes and different species of permeable ions. Well accepted orders of magnitudes for the ion concentrations inside the cells (about 200 mM) and outside (about 300 mM) suggest that the concentration of the impermeant osmolytes should be about 100 mM, corresponding to an osmotic pressure of about $10^5$ Pa. Taking the cortical tension to be of order of 1 mN/m and cell radius to be about 10 μm we get the hydrostatic pressure difference (Laplace pressure) produced by the cell cortex to be about 100 Pa. It means that cortical tension would be able to impose a volume change that would 'concentrate' the impermeant osmolytes by a maximum of only about 1%. For small variation around this steady state, a 10% decrease in cell volume will increase the osmotic pressure of the trapped osmolytes by $10^4$ Pa. For this osmotic pressure to be balanced by an increase in the hydrostatic pressure, the cortical tension would need to increase by a factor of 100, which we consider to be too large to be realistic. This reasoning leads to the generally accepted result that even a large increase of contractility (e.g., by a factor of 10) would only change the volume by 1%. Overall, these considerations demonstrate that the current versions of the PLM or its extensions cannot explain our observations.

These results also confirm our interpretation that decreasing contractility leads to a larger loss of volume during spreading mostly indirectly, because it increases the speed of cell spreading.

## A mechano-sensitive PLM including a mechano-osmotic coupling predicts the observed relation between spreading speed and volume loss

We thus engaged in proposing a modified model (see the full model in Appendix 1), to combine PLM with cell mechanics and shape. To account for the observation of an increase in volume loss for faster spreading cells, we made the assumption, like the other models discussed above, that an element coupling cell mechanics (which is directly affected by the spreading speed) to the ion fluxes need to be added to the classical PLM. Ion channels and pumps can be affected by membrane tension, as demonstrated multiple times by others (*Cox et al., 2019*), we thus chose to implement this mechano-osmotic coupling, similarly to the model discussed above (*Xie et al., 2018*; *McEvoy et al., 2020*; *Jiang and Sun, 2013*; *Tao and Sun, 2015*), but with a full PLM, including permeant and impermeant solutes (see also *Yellin et al., 2018*; *Li et al., 2021*). We also chose to implement cell growth in the model, using the experimentally measured rate (about 5% volume increase per hour), because it plays a significant role at timescales of a few tens of minutes, thus overlapping with the timescale of the latest part of the spreading process.

An important assumption of such a model is that faster spreading cells display a higher membrane tension specifically during the phase of fast spreading. To test this hypothesis, we pulled membrane tubes with an atomic force spectrometer tip following a well-established protocol (see Materials and methods for further details; *Diz-Muñoz et al., 2016*). This allows to measure the tether force, which varies with the square root of the membrane tension in the absence of a cell cortex. It is a common readout of an apparent membrane tension, even if it is not a direct measure, because it also depends on the membrane interaction with the cell cortex (*Sitarska and Diz-Muñoz, 2020*). Because tether force measures take a significant time and require adhered cells, we started measures 30 min after cell plating and performed them for the following hour. These measures correspond to the spreading phase, which is comparable to the phase of volume loss in the single-cell spreading experiments (in which time zero is taken as the time when each single cell starts to spread, and not the initial seeding time, which gives on average a delay of about 10 min). This early time point designated as 'spreading phase' thus overlaps with the phase in which cells are still spreading and losing volume. We then performed a second measure within 4–5 hr after seeding, which is long enough after the spreading to consider that cells have recovered their steady-state growing phase and the effect of their initial spreading on volume is lost (see more details in Materials and methods). We found that cells treated with Y-27, which spread faster than control cells, displayed a higher tether force during the spreading phase, while the force was similar to control cells at steady state (*Figure 3E*), showing that the increase was not due to the drug treatment itself. Importantly, we also showed that cells that stayed rounded because they had been plated on small (20 μm in diameter) micropatterns, measured 4–5 hr after

seeding, had a low membrane tension, treated or not with Y-27632. This experiment suggests an effect of spreading speed on membrane tension. This is consistent with the hypothesis that membrane tension might modulate cell volume upon fast cell shape changes.

Another important requirement to propose that an extended PLM could explain our observations on spreading cells is that the timescales involved in the osmotic and in the spreading phenomena match. To estimate the typical timescales of water and ion fluxes, we performed a detailed characterization of the cell response to osmotic shocks. We first made high time resolution recordings of cell swelling and shrinking upon a change in the external osmolarity (*Figure 4A* and *Figure 4— figure supplement 1A, B* and *Figure 4—video 1*). The change of volume occurred in a timescale of seconds, as expected. These experiments provided the rate of cell water entry and exit as a function of the difference in osmotic pressure (*Figure 4B*). This allowed us to estimate the permeability (*Figure 4—figure supplement 1*), which appeared smaller for hyper-osmotic shocks than for hypo-osmotic shocks, as reported previously (*Chara et al., 2005*; *Peckys et al., 2011*), although the reason for this difference is not understood. We next characterized the longer, minutes timescale of volume adaptation (*Figure 4C*). It showed that volume adapted faster for larger shocks. At the level of individual cells, the response was quite homogenous for the recovery from hypertonic shocks, while there was a higher cell-cell variability during recovery from hypotonic shocks (*Figure 4—figure supplement 1D*), with cells showing only partial recovery, especially for large hypo-osmotic shocks. Despite these complex single-cell behaviors, these experiments provide clear evidence, as well known from decades of studies of this phenomenon, of an active volume regulation mechanism on the timescale of minutes, setting the typical timescale for ion fluxes. These two timescales, seconds for water flows through the membrane and minutes for ion fluxes, are basic assumptions of the PLM model verified by our experiments. Importantly, the rate of volume change observed for small shocks is similar to the rate of volume change during cell spreading experiments (about 10 μm³/min). This justifies the use of a mechanosensitive PLM to explain the cell spreading data.

In brief, in this new model (see details in Appendix 1), the dependence of the volume on membrane tension is through the mechanosensitivity of the ion channels and pumps. In the linear regime, we assume that small changes in tension lead to a small change in ion transport rates so that the volume change is proportional to the change in tension. The equation for change in volume reads

$$\frac{d\delta V}{V_{iso}} = \alpha \frac{d\gamma}{\gamma_{iso}},$$

(1)

where $\delta V = V(t) - V_{iso}(1 + r_{\text{growth}}t)$, with $r_{\text{growth}}$ and being the growth rate of the cell and time, respectively. If the mechanosensitivity parameter is negative, then the volume will decrease upon an increase in the tension. We note that *Equation 1* is valid for small changes in volume corresponding to small changes in tension. The other assumption we make in writing *Equation 1* is that over the timescale of cell-spreading the ion and water transport have equilibrated. Since we are dealing with linearized equations, *Equation 1* includes the complex properties of the ion transport like voltage and concentration dependence of ion channels and pumps. We explicitly evaluate the mechanosensitivity parameter, by analyzing a model of ion transport with three ion species—sodium, potassium, and chloride. We find that the sign and magnitude depend on the mechanosensitivity of the potassium and sodium channels, and on the ion concentrations before spreading. For physiological values of parameters found in the literature (see values in Appendix 1), we expect to be negative. Note that other effects not accounted for in our simple estimate, such as the voltage and concentration dependence of ion channels and the existence of co-transporters, could affect the value and sign of. To relate tension variations to the rate of cell spreading, we model surface tension using a Maxwell fluid model, with a relaxation timescale and elastic modulus, driven by the rate change of total surface area. The tension dynamics reads

$$\left(1 + \tau \frac{d}{dt}\right) \frac{\delta\gamma}{\gamma_{iso}} = \frac{k\tau}{\gamma_{iso}} \frac{dA_{tot}}{dt}$$

(2)

The elastic modulus characterizes the short-time elastic response, while the relaxation timescale accounts for the existence of tension homeostasis mechanisms that have a longer response time. During cell spreading, the total surface area will increase leading to a spreading rate-dependent increase in tension, which will relax back to the homeostatic value, in agreement with the tether pulling experiments reported in *Figure 3E*. To estimate the total surface area, we take the cell shape

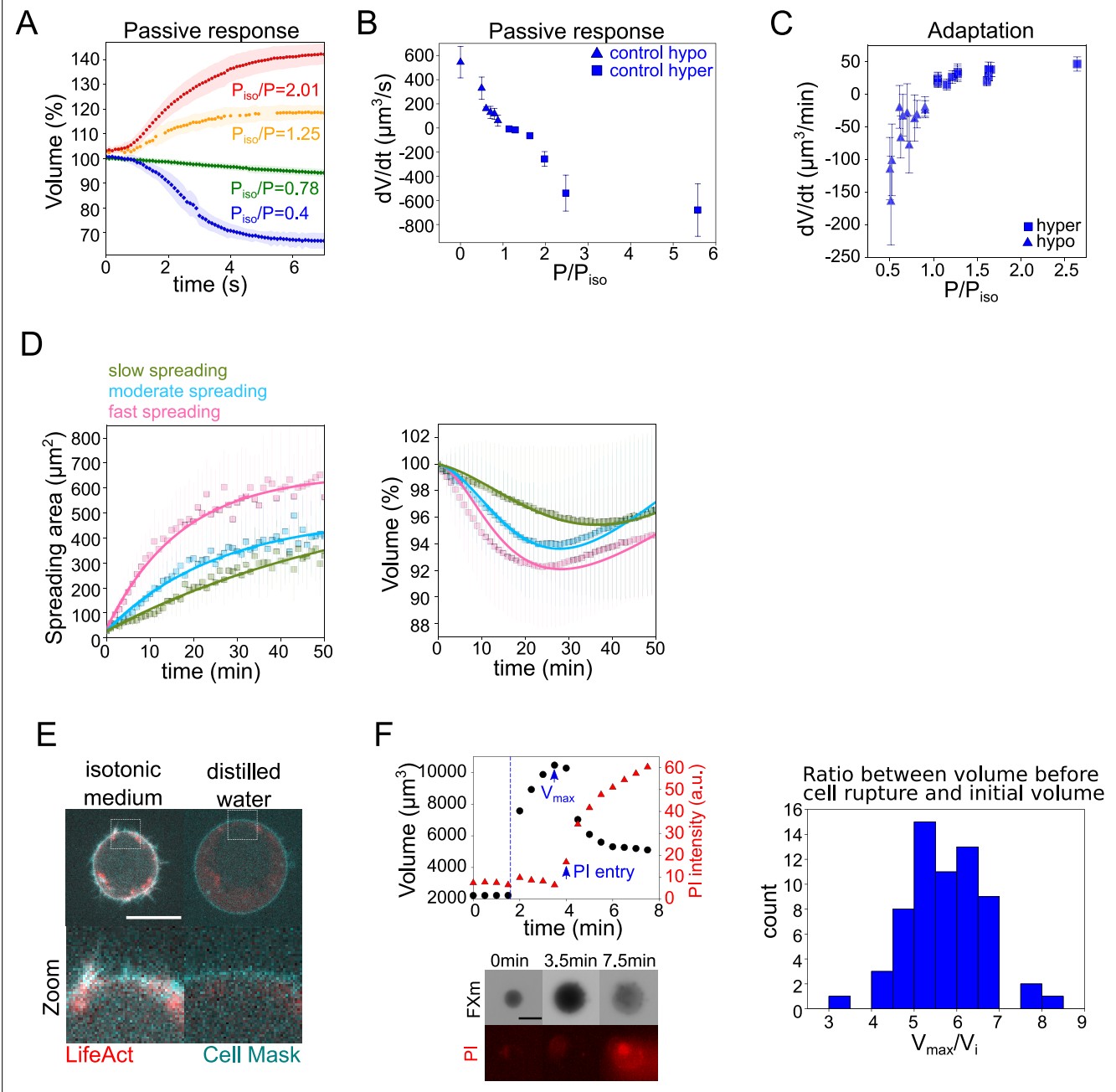

**Figure 4.** The mecano-sensitive PLM explains the volume loss in spreading cells. (**A**) Average normalized volume of HeLa Kyoto cells during initial response to osmotic shocks of different magnitudes measured with high time resolution, 100 ms. Number of cells in the experiments: control $P_{iso}/P=1.25$ (n=13, N=1), $P_{iso}/P=2.01$ (n=19, N=1), $P_{iso}/P=0.78$ (n=15, N=1), $P_{iso}/P=0.4$ (n=17, N=1). Error bars represent standard deviation. (**B**) Average volume flux in HeLa Kyoto cells during initial response to osmotic shocks of different magnitudes. Each point represents average value of single experiment, average number of cells in each experiment n~12. Error bars represent standard deviation. (**C**) Average volume flux in HeLa Kyoto cells during regulatory volume adaptation. Each point represents average value of single experiment, average number of cells in each experiment n~48. Error bars represent standard deviation. (**D**) Fits of the spreading data from the model using best fit parameters on average normalized volume (*right*) and spreading area (*left*) of control cells divided into three categories represented in *Figure 1H*. (**E**) Z-plane of HeLa LifeAct-mcherry (red) cell before and after addition of distilled water, cell membrane is stained with CellMask Green (cyan). Scale bar 10 μm. (**F**) *Left top*: Volume (black) and propidium iodide (PI) intensity of single HeLa Kyoto cell exposed to distilled water. Dashed line indicates the time of distilled water addition. Reaching of maximum cell volume is followed by cell membrane rupture, volume decrease, and PI entry into the cell. *Left bottom*: Corresponding FXm images and PI staining. *Right*: Distribution of ratio between maximum volume cells reach before bursting induced by exposure of distilled water and their initial volume (n=63, N=3).

The online version of this article includes the following video, source data, and figure supplement(s) for figure 4:

*Figure 4 continued on next page*

*Figure 4 continued*

**Source data 1.** Data tables related to quantifications in Figure 4.

**Figure supplement 1.** Experimental estimation of parameters for the PLM.

**Figure supplement 1—source data 1.** Data tables related to quantifications in Figure 4 Supplement 1.

**Figure 4—video 1.** FXm imaging of HeLa EMBL cells attached on PLL-coated glass exposed to osmotic shock recorded with high frame rate.
https://elifesciences.org/articles/72381/figures#fig4video1

**Figure 4—video 2.** 3D-shape reconstruction by FXm of HeLa EMBL cells spread for 20 min at fibronectin-coated glass.
https://elifesciences.org/articles/72381/figures#fig4video2

---

to be that of a spherical cap (*Figure 4—video 2*; we discuss in more details the possible shape approximations and their relation to the measured spreading area in the Appendix 1). Combining the tension dynamics in *Equation 2* with tension/volume coupling in *Equation 1* leads to the following effective viscoelastic model for volume dynamics driven by a change in the total area,

$$\left(1 + \tau \frac{d}{dt}\right) \frac{\delta V}{V_{iso}} = -\xi\tau \frac{1}{A_{tot}} \frac{dA_{tot}}{dt},$$ (3)

where the effective elastic modulus $\xi = -\left(A_{tot}k\alpha\right)/\gamma_{iso}$ is proportional to the effective elasticity of the membrane and to the magnitude of the mechanosensitivity parameter relating volume loss to the tension increase and is also inversely proportional to the surface tension. The total area itself depends on the volume, we can write *Equation 3* as

$$\left(1 + \tau_{eff} \frac{d}{dt}\right) \frac{V}{V_{iso}} = 1 + r_{\text{growth}}\left(t + \tau\right) - \xi\tau f_2\left(V, A_c\right) \frac{dA_c}{dt},$$ (4)

where $\tau_{eff} = \tau\left(1 + \xi f_1\left(V, A_c\right) V_{iso}\right)$, $f_1\left(V, A_c\right)$, and $f_2\left(V, A_c\right)$ are functions that are given by the geometry of the cell, which relate the change in total area to the change in volume, and change in contact area respectively. The model parameter $\tau_{eff}$ is an effective relaxation timescale for volume that depends on $\tau$, volume, and contact area. The difference between $\tau$ and $\tau_{eff}$ stems from the complicated geometrical relationship between contact area and total area in a spreading cell. The contact area dynamics is fitted to an exponentially saturating function. The two input parameters for the models are $\tau_a$: the timescale of cell spreading and $A_0$: the saturating value, obtained by fitting the cell spreading curve. Fitting the volume dynamics yields the two model parameters $\xi$ and $\tau$ for the mechanosensitivity of ion transport and membrane mechanics.

These parameters allowed us to fit the various experimental data and their values are discussed in more details below. Importantly, this simple extension of the PLM predicts the observed proportionality between volume loss and the speed of spreading, and no dependency on the absolute cell spreading area (*Equation 3*, *Figures 1I, 2C and D*). We conclude that this new model is able to explain our main observation and constitutes a robust implementation of a membrane tension homeostasis mechanism within the PLM framework. We propose to call it the mechanosensitive PLM.

## Fitting the spreading and volume data with the mechano-sensitive PLM

To further test the capacity of the model to explain our observations, we performed a fit of our experimental data using the mechanosensitive PLM model and analyzed the parameters obtained. We used the three groups of control cells defined in *Figure 1H*, sorted based on spreading speed during the first 10 min. The spreading parameters were extracted from the experimental spreading data, and the model allowed a satisfactory fit of the experimental volume data (*Figure 4D*).

Because the mechanosensitive PLM assumes a coupling between membrane tension and ion fluxes, how much volume is lost by a cell during spreading depends on whether the cell deforms in a rather elastic or viscous regime. The transition between these regimes is defined by the relative values of the spreading rate and the effective tension relaxation time—if the spreading rate is faster than the relaxation time, the cell deforms in a rather elastic regime, and as a result, the membrane gets tensed and the cell loses volume. The effective relaxation time depends on the two main fitting parameters, the bare tension relaxation timescale (which varies in the minutes to tens of minutes timescale) and the stiffness (which varies around one). When fitting the three classes of fast, intermediate, and slow-spreading cells, we found that the values of the fitting parameters (*Appendix 1—table 3* in

Appendix 1), do vary significantly for the three classes. However, this variation could not explain the difference in volume loss (see Appendix 1), which must therefore be attributed to the difference in spreading speed. We conclude that our mechanosensitive PLM not only captures properly the coupling of spreading kinetics on volume modulation but that the parameter fitting suggests that the key ingredient of the model, the finite response time of the mechano-osmotic feedback, might be the cause of the volume loss in fast-spreading cells.

## Volume loss upon fast cell deformation depends on branched actin and on changes in ion fluxes

We then asked what could be the origin of the increase in surface tension during fast cell spreading. We first evaluated the total amount of cell membrane available. We exposed cells to distilled water and first imaged actin and membrane staining. It showed a rapid full unfolding of membrane reservoirs (*Figure 4E*) before the cell exploded. We then used propidium iodide (PI) to identify the timing of plasma membrane rupture (*Figure 4F*). We found that on average, the plasma membrane ruptured when cells reached 5.7 times their initial volume, which corresponds to an excess of membrane surface area of about 3.3 times, in accordance with previous measures (*Ting-Beall et al., 1993*; *Guilak et al., 2002*).

It means that cells have a very large excess of membrane surface area and that membrane tension could not arise from a limitation in the total amount of membrane. Nevertheless, the plasma membrane being bound to the underlying cytoskeleton, its restricted unfolding could generate an increase in tension depending on the rate of cell deformation. This would explain why the volume loss depends on the spreading speed. We thus further tested the role of the actin cytoskeleton in volume loss during cell spreading.

Because branched actin was shown to more specifically interact with the plasma membrane (*Diz-Muñoz et al., 2016*; *Lieber et al., 2013*) and modulate membrane tension, we used HeLa EMBL cells treated with the Arp2/3 complex inhibitor CK-666, and combined the treatment with Y-27 to induce fast spreading. We found that CK-666 treatment alone induced both a slower spreading and lower volume loss (2–3%, *Figure 5A and B*), similar to the low Lat A treatment (*Figure 2B and C*), which was well fitted by the mechanosensitive PLM (fits on *Figure 5A and B*). Treatment with Y-27 increased the spreading speed of CK-666 treated cells, but the volume did not decrease in this fast-spreading condition (*Figure 5C and D*). The mechanosensitive PLM could fit these data by adjusting the parameter coupling surface tension to the change of activity of ion pumps (*Appendix 1—table 4* in Appendix 1). To directly test the role of ion fluxes in the volume loss, we targeted two main players: first, stretch-activated calcium channels (including Piezo), using gadolinium chloride (GdCl$_3$) and second NHE1, the sodium/proton exchanger, using EIPA. Neither of them have a direct role in volume regulation (*Figure 5—figure supplement 1A*), because there are too few calcium ions in the cell (for Piezo) or protons (for NHE1). Affecting direct volume regulation by targeting the transport of sodium or potassium ions would be too detrimental for the cell and change dramatically the initial cell volume. By contrast, inhibition of these two channels did not affect the initial average cell volume (*Figure 5—figure supplement 1* A). Piezo was chosen because it is known to be mechanosensitive and because calcium acts upstream of many fast cell response pathways, including regulating cell contractility (*Clapham, 2007*) and it was proposed before to play a role in cell volume regulation (*Hua et al., 2010*). We also chose to target NHE1 because its inhibition was previously shown to affect fast cell volume changes, at mitotic entry (*Byun et al., 2015*; *Miettinen et al., 2021*) and during cell spreading (*Xie et al., 2018*), as we confirmed in our experiments (*Figure 5—figure supplement 1B*). Treatment with GdCl$_3$ led to an increase in volume loss (from 5% to 8%), which could be fully accounted for by the increase in cell spreading speed (*Figure 5E and F*) and well fitted by the mechanosensitive PLM. Because the effect of GdCl$_3$ can be purely accounted for by the increase in spreading speed and thus does not disrupt the mechanosensitive PLM, we speculate that it could be explained by a reduction of cell contractility, since preventing calcium entry might reduce actomyosin motors activation (*Clapham, 2007*), and thus increase the spreading speed (*Wakatsuki et al., 2003*). On the other hand, HeLa EMBL cells treated with EIPA, while spreading slightly faster than control cells, lost less volume (*Figure 5G and H*). Combining EIPA with Y-27 showed that, despite a fast spreading speed comparable to Y-27 treated cells, NHE1 inhibition fully prevented volume loss (*Figure 5G and H*), an effect that we also observed for RPE-1 cells (*Figure 5—figure supplement*

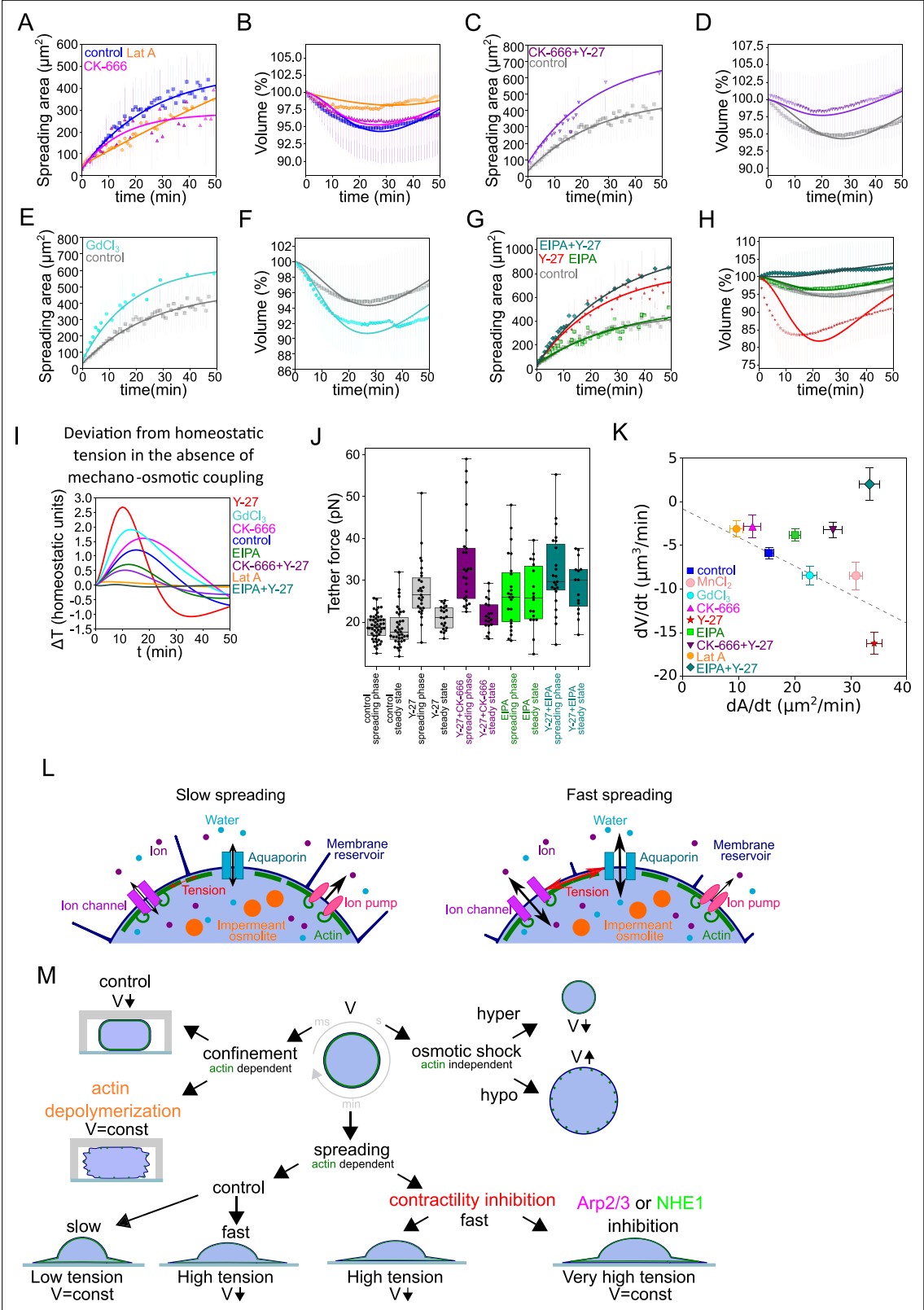

**Figure 5.** The mecano-sensitive PLM depends on branched actin and modulation of ion fluxes and constitutes a surface tension homeostasis mecanism. (**A**) Two parameter fits for the spreading kinetics using the exponential saturation anzatz (see text) on average area of control cells (blue, n=73, N=1), 100 nM Latrunculin A (orange, n=30, N=2) or 100 μM CK-666 (magenta, n=37, N=2) treated. Error bars represent standard deviation. (**B**) Fits from the model using best fit parameters on average normalized volume of control cells (blue, n=73, N=1), 100 nM Latrunculin A (orange, n=30, N=2) or 100 μM

*Figure 5 continued on next page*

*Figure 5 continued*

CK-666 (magenta, n=37, N=2) treated. Error bars represent standard deviation. (**C**) Two parameter fits for the spreading kinetics using the exponential saturation anzatz (see text) on average area of control cells (gray, n=73, N=1) or combination of 100 μM CK-666 and 100 μM Y-27632 (violet, n=24, N=1) treated. Error bars represent standard deviation. (**D**) Fits from the model using best fit parameters on average normalized volume of control cells (gray, n=73, N=1) or combination of 100 μM CK-666 and 100 μM Y-27632 (violet, n=24, N=1) treated. Error bars represent standard deviation. (**E**) Two parameter fits for the spreading kinetics using the exponential saturation anzatz (see text) on average area of control cells (gray, n=73, N=1) or 100 μM GdCl$_3$ (cyan, n=30, N=2) treated. Error bars represent standard deviation. (**F**) Fits from the model using best fit parameters on average normalized volume of control cells (gray, n=73, N=1) or 100 μM GdCl$_3$ (cyan, n=30, N=2) treated. Error bars represent standard deviation. (**G**) Two parameter fits for the spreading kinetics using the exponential saturation anzatz (see text) on average area of control cells (gray, n=73, N=1), 100 μM Y-27632 (red, n=21, N=1), 50 μM EIPA (green, n=73, N=1), or combination of 50 μM EIPA and 100 μM Y-27632 (dark cyan, n=30, N=2) treated. Error bars represent standard deviation. (**H**) Fits from the model using best fit parameters on average normalized volume of control cells (gray, n=73, N=1), 100 μM Y-27632 (red, n=21, N=1), 50 μM EIPA (green, n=73, N=1), or combination of 50 μM EIPA and 100 μM Y-27632 (dark cyan, n=30, N=2) treated. Error bars represent standard deviation. (**I**) Predicted by model, plots for difference between tension without mechano-osmotic coupling (for $\alpha = 0$ and $\frac{k}{\gamma_{iso}} = 100$) and tension with mechano-osmotic coupling (for fitted and $\frac{k}{\gamma_{iso}} = 100$). (**J**) Tether force measurements of control HeLa Kyoto cells (gray, for 'spreading phase' n=50, N=6; for 'steady state' n=55, N=9), treated with Y-27632 (gray, for 'spreading phase' n=27, N=3; for 'steady state' n=21, N=3), CK-666+Y-27 (purple, for 'spreading phase' n=25, N=3; for 'steady state' n=19, N=3), EIPA (green, for 'spreading phase' n=23, N=3; for 'steady state' n=18, N=3), EIPA+Y-27 (dark cyan, for 'spreading phase' n=23, N=3; for 'steady state' n=15, N=3) during the first 30–90 min after plating or 4–5 hr after plating. Error bars represent standard deviation. The results of statistical tests are shown at the graph. (**K**) Volume flux (dV/dt) of single control HeLa Kyoto cells (n=194, N=3), treated with Lat A (n=41, N=3), CK-666 (n=54, N=3), Y-27 (n=121, N=4), EIPA (n=117, N=3), GdCl$_3$ (n=53, N=3), CK-666+Y-27 (n=74, N=3), EIPA+Y-27 (n=50, N=3), MnCl$_2$ (N=3, n=57) plotted versus their spreading speed (dA/dt) at the first 10 min of spreading. Error bars represent standard error. (**L**) Scheme of mechanosensitive "pump-leak" model. (**M**) Scheme representing cell volume regulation in response to deformations.

The online version of this article includes the following source data and figure supplement(s) for figure 5:

**Source data 1.** Data tables related to quantifications in Figure 5.

**Figure supplement 1.** Effect of drug treatments on initial cell volume, adaptation to osmotic shocks and volume loss in RPE1 cells.

**Figure supplement 1—source data 1.** Data tables related to quantifications in Figure 5—figure supplement 1.

*1C*). Inhibition of NHE1, which is known to affect ion transport, is thus fully preventing volume loss during fast spreading. Importantly none of the drug treatments performed significantly affected the initial volume of cells prior to spreading (*Figure 5—figure supplement 1A*). This is consistent with the importance of changes in ion fluxes in the mechanosensitive PLM.

## The mechano-osmotic coupling moderates the membrane tension increase in fast-spreading cells, acting as a membrane tension homeostasis mechanism

The mechanosensitive PLM predicts that inhibition of the mechano-osmotic coupling in fast-spreading cells, would prevent the associated volume loss, and lead to an increase in the membrane tension during the spreading phase. To give a qualitative prediction of this effect, the membrane tension value was extracted from the model, for the various conditions tested experimentally, using the fit on the experimental data, and the tension values were compared for the case of a model with or without a mechano-osmotic coupling (the difference between the two values is given in *Figure 5I*). This model prediction shows that the largest increase in tension, in case the mechano-osmotic coupling was absent, is expected in the case of the fastest spreading cells (Y-27 or GdCl3 treatments). On the other hand, in case the mechano-osmotic coupling is already disrupted by the treatment (e.g., in the case of EIPA treatment), or if cells spread very slowly (e.g., Lat A treatment), the predicted difference is small. To test this prediction experimentally, we performed tether pulling experiments (*Figure 5J*). These experiments showed that, as predicted by the model, disrupting the mechano-osmotic coupling in fast-spreading cells (Y-27 plus CK-666 or Y-27 plus EIPA) leads to the highest tether force values in the spreading phase, while the steady-state values did not change. Similarly, combined EIPA and Y-27 treated cells showed higher tension than Y-27 or EIPA alone. Tension was highest during early spreading compared to steady-state spread cells, suggesting that the increase was due to spreading and not to the drug treatments alone, even though EIPA alone also add an effect on steady-state tension. This shows that, in these cells, the coupling between membrane tension and volume regulation is lost, and that fast spreading in the absence of volume loss induces higher tension increase (*Figure 5J and K*), as predicted by the model (*Figure 5I*). Taken together, these experiments confirm the validity of our mechanosensitive PLM. They also support the existence of a membrane tension

homeostasis mechanism that reduces the extent of changes in membrane tension upon fast cell shape changes by modulating the relative contribution of surface expansion and volume loss.

## Discussion

### A mechano-osmotic coupling leads to volume loss in fast-spreading cells

Our detailed characterization of cell volume during cell spreading revealed that, while cell volume is not related to the steady-state shape of the cell, it is modulated by the rate of cell shape change. We propose that this is due to a coupling between cell membrane tension and rates of ion fluxes (*Figure 5L*). An extension of the classical PLM including this coupling can account for our observations of cell volume during cell spreading. Measures of membrane tension during spreading and at steady state under a variety of conditions confirmed that fast spreading is associated with a transient increase in membrane tension and that preventing volume modulation leads to even higher membrane tension, as predicted by the model. Taken together, these experiments and this model suggest the existence of a mechano-osmotic coupling at the level of the cell membrane, which acts as a membrane tension homeostasis mechanism by reducing membrane tension changes upon fast cell deformation (*Figure 5M*).

### The role of membrane binding to the actin cortex in inducing membrane tension and volume loss in fast-spreading cells

A central hypothesis in the model is that the physical coupling between the actin cortex and the cell membrane leads to an increase in membrane tension when the rate of deformation is faster than the relaxation time of the actin cortex and membrane ensemble. To verify this hypothesis, we performed membrane tether experiments in various conditions, during spreading and at steady state (*Figure 5J*). Membrane-to-cortex attachment is at least partly mediated by proteins of the ERM family *Chugh and Paluch, 2018*. Thus, we performed an additional experiment using an inhibitor of Ezrin, membrane-cytoskeleton linker (20 µM NSC668394) and monitored cell volume during spreading. We found that while spreading was similar or even slightly faster during the initial phase, treated cells lost less volume than control cells (*Figure 6A*), consistent with a role of cortex/membrane coupling in mediating the effect of spreading kinetics on volume loss. To further investigate the ultrastructure of the cell cortex during spreading, we unroofed (*Vassilopoulos et al., 2019*) Hela cells after 30 min spreading on a fibronectin-coated substrate. Following electron microscopy imaging confirmed the different extent of spreading in the various conditions assayed, and the perturbation of branched actin in the CK-666 treated cells (*Figure 6B*). Membrane folds and structures such as clathrin-coated pits and caveolae were present in all conditions. Although their number and degree of curvature did not change significantly, the different populations of caveolae appeared hard to quantify and compare on spreading cells without underestimating the amount of flat caveolae. We conclude that, while our biophysical measures and fluorescence imaging gave a clear indication of changes in membrane tension and folding state during large cell deformations, further investigations are needed to precisely describe the change in the state of the membrane and its association to the actin cortex in this context. In particular, branched actin is known to be important for endo- and exocytosis (*Toshima et al., 2005*; *Tran et al., 2015*) and, on the timescale of several minutes to tens of minutes, it could regulate membrane tension, especially during cell spreading (*Gauthier et al., 2011*) and thus contribute to cell volume modulation.

### The sign of the volume change upon fast cell deformation

Although the effect of mechanosensitive ion channels on volume has been discussed before for simplified systems considering one or two solutes (*Guo et al., 2017*; *Xie et al., 2018*; *Adar and Safran, 2020*; *Jiang and Sun, 2013*), the relation between mechanosensitivity of ion channels and pumps and volume change is far from obvious. The sign of the volume change upon an increase in tension depends on whether the contribution of ions to the osmotic pressure increases or decreases. Since the cell is always osmotically balanced, if the concentration of the ions inside the cell decreases, the concentration of the trapped molecules should increase by decreasing the volume such that the cell osmotic pressure stays constant and vice-versa. For instance, an increase in sodium channel

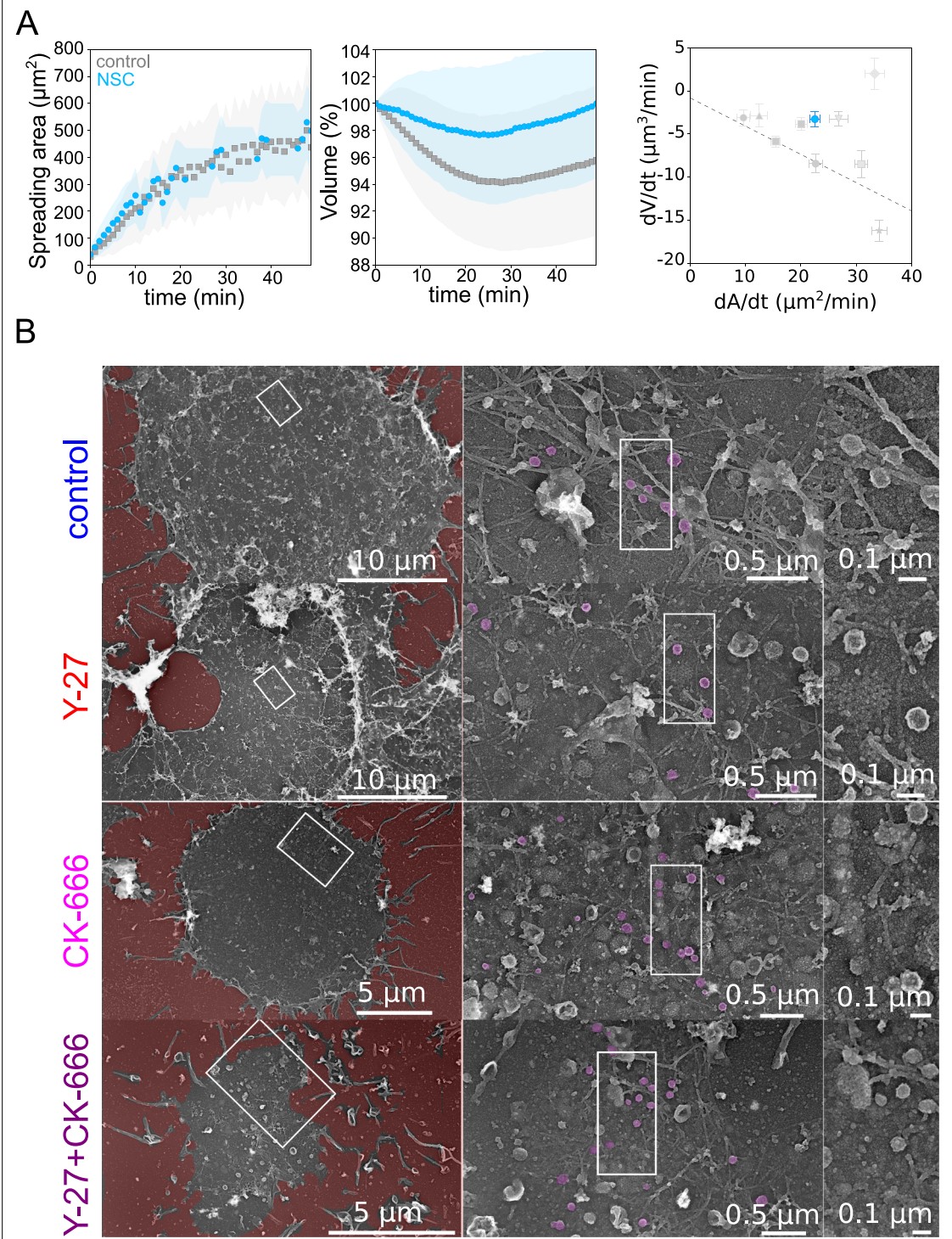

**Figure 6.** Role of membrane to actin attachment on volume loss and state of the plasma membrane in spreading cells. (**A**) *Left*: Average spreading area of control HeLa Kyoto cells (gray, n=125, N=1) or 20 μM NSC (light blue, n=101, N=3). *Middle*: Average normalized volume of control HeLa Kyoto cells (gray, n=125, N=1), or 20 μM NSC (light blue, n=101, N=3). Error bars represent standard deviations. *Right*: Volume flux (dV/dt) plotted versus their spreading speed (dA/dt) of single control HeLa Kyoto cells and treated with various drugs and represented in *Figure 5K* and treated with 20 μM NSC (light blue, n=101, N=3). Error bars represent standard error. (**B**) Platinum replica electron microscopy survey views of the cytoplasmic surface in control, Y-27632, CK-666, or CK-666+Y-27632-treated unroofed Hela cells spread on glass coverslips for 30 min. Extracellular substrate is pseudo-colored in red. For each panel, high magnification views corresponding to the boxed regions are shown on the right.

The online version of this article includes the following source data for figure 6:

**Source data 1.** Data tables related to quantifications in Figure 6.

conductance upon an increase in tension leads to an increase in volume, whereas an increase in potassium conductance upon an increase in tension leads to a decrease in volume. Using a detailed model of ion transport, we show that for physiological values of parameters, as observed in experiments, the volume is indeed expected to decrease upon an increase in tension.

## Values of fitting parameters for the mechanosensitive PLM suggest a role for branched actin in modulating ion fluxes

This new mechanosensitive PLM gives a Maxwell viscoelastic model for the volume with two fitting parameters, the effective stiffness, and the bare relaxation timescale. In most cases, we get a good quantitative fit for cells treated with different drugs that perturb the cytoskeleton and the ion channels (Appendix 1). For the Y-27 and Lat A treated cells, the fits only qualitatively capture the temporal dynamics of the volume. One of the reasons for an imperfect fit for these two drugs could be the failure of the spherical cap approximation used to estimate the surface area (see more discussion on the shape estimates and the parameters used for cell surface area in the model, Appendix 1). Control cells, and cells treated with Y-27, EIPA, and CK-666 show less than 30% variation in the value of $\xi$, implying that most of the volume loss is explained by differences in spreading speed. Cells treated with GdCl$_3$ show a larger decrease of 60% but stay in the same range of parameters (and they are close to the same line in the dV/dt versus dA/dt summary graph shown in *Figure 5K*). However, for the cells treated with Y-27+EIPA and Y-27+CK-666, $\xi$ decreases by an order of magnitude, leading to low volume loss even though the cells are spreading fast (*Appendix 1—table 4* in Appendix 1). This decrease of $\xi$ could be either due to a decrease in the elasticity of the membrane or due to the decrease in the value of the mechanosensitivity parameter. Spreading experiments show that, for both Y-27+CK-666 and Y-27+EIPA, membrane tension reaches the highest values. This means that, in both cases, spreading is still inducing an increase in membrane tension, and the absence of volume loss reinforces the effect on membrane tension. It suggests that the elasticity parameter is not affected but rather the volume-tension electromechanical coupling. This could mean that, unexpectedly, branched actin networks are specifically required for this coupling. This could be due to a direct association of branched actin with ion channels and pumps (*Mazzochi et al., 2006*; *Shaw and Koleske, 2021*).

## Volume loss in ultra-fast deforming cells

The PML and our mechanosensitive extension are meant to explain volume changes at the minutes timescale, which correspond to the time needed for large enough ion fluxes to take place. To test the limit of validity of the model, we imposed fast (less than a second timescale) deformation on the cell, we used our previously developed cell confiner (*Le Berre et al., 2014*). This device can impose a precise height on cells and thus gives access to a large range of deformations (*Figure 7A* and *Figure 7—figure supplement 1A* and *Figure 7—video 1*). RICM measure of the cell contact area showed a range of spreading similar to what was observed during spontaneous cell spreading (*Figure 7B* and *Figure 7—figure supplement 1B*). In addition, imaging of the plasma membrane showed that confinement below 10 µm induced a clear loss of membrane folds and reservoirs (*Figure 7C* images and *Figure 7—video 2*), while treatment with Lat A induced the formation of large membrane blebs and less extension of the cell diameter upon confinement (*Figure 7C* graph and *Figure 5—figure supplement 1C* and *Figure 7—video 2*). This suggests that cell confinement, like hypo-osmotic shocks, induces membrane reservoir unfolding, and that Lat A treatment, by reducing the membrane anchorage and causing bleb formation, reduces the surface expansion following confinement. FXm volume measurement combined with confinement showed a strong loss of volume of confined HeLa EMBL control cells, while Lat A treated cells kept a constant volume (*Figure 7D* and *Figure 7—figure supplement 1D* showing for both treatments the decrease in FXm background intensity corresponding to the confiner height; and *Figure 7—video 3*). In control cells, stronger confinement led to larger volume loss, while Lat A treated cells showed no significant volume loss except for the lowest confinement height (*Figure 7E*). Conversely, treatment with Y-27632 to reduce contractility had no effect on volume loss upon confinement (*Figure 7E*), confirming that contractility has no direct effect on cell volume and that the effect of Y-27632 on cell volume during cell spreading is mostly via an increase of the spreading speed. The loss of volume in control cells corresponded to a deformation at an almost constant surface area (*Figure 7F*, calculated from the volume, see Appendix 1). Below 5 µm height, the cell surface significantly increased, which also corresponded to

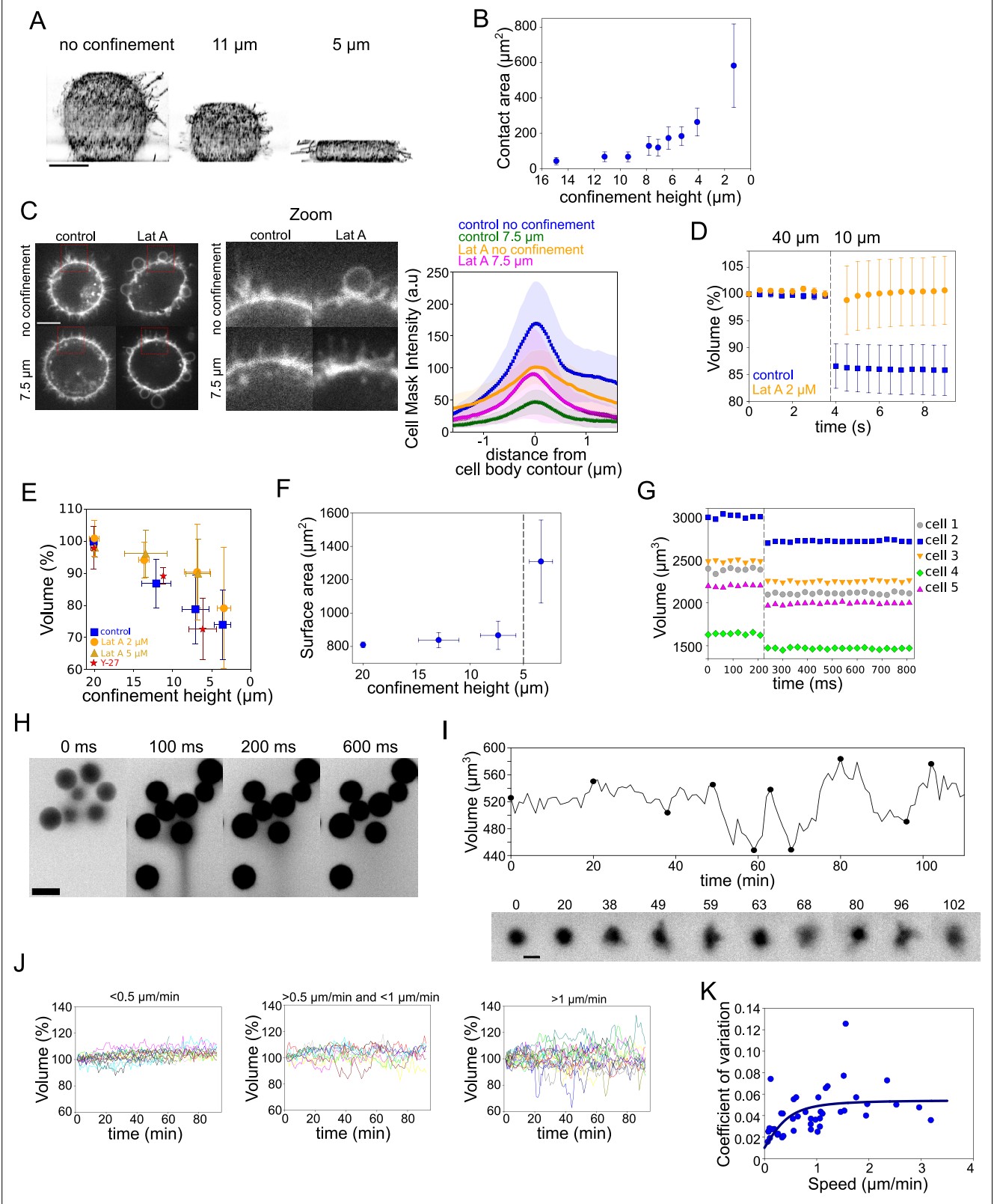

**Figure 7.** Volume modulation during ultra-fast cell flattening and during cell migration through collagen matrices. (**A**) 3D-membrane reconstruction of HeLa expressing MyrPalm-GFP (black) cells cell shape under different confinement heights, side view. Scale bar: 10 μm. (**B**) Contact area with bottom glass substrate of Hela Kyoto cells under different confinement heights. Average number of cells in each experiment n~79, for each condition N=1. Error bars represent standard deviation. (**C**) *Left*: Z-plane of control and 2 μM Lat A treated HeLa cells under 20 μm and 7.6 μm confinement heights. Cell

*Figure 7 continued*

membrane is stained with CellMask Far Red (white). Scale bar: 10 µm. *Right*: Average CellMask intensity plotted versus distance from cell body contour on the middle Z-plane of HeLa-MYH9-GFP-LifeAct-mcherry cells. Number of cells in each condition n=10, N=1 for each condition. Error bars represent standard deviation. (**D**) Average normalized volume of control (blue, n=48, N=1) and 2 µM Lat A treated (orange, n=32, N=1) HeLa Kyoto cells during dynamic confinement experiment. Dashed line indicates the moment of confinement. Error bars represent standard deviation. (**E**) Average normalized volume of HeLa Kyoto cells (blue) and cells treated with Lat A 2 µM (orange) or 5 µM (yellow) or 100 µM Y-27 (red) under different confinement heights. Each data point represents an average of N~10 experiments; each experiment contains n~160 individual cells. Error bars represent standard deviation. There is statistically significant difference between control and Lat A for the heights ~12 µm and ~7 µm (p=0.02 and p<10⁻⁸). There is no statistically significant difference between control and Y-27 for the heights ~12 µm and ~7 µm (p=0.4 and p=0.3). (**F**) Projected surface (computed from volume represented in panel *Figure 5G*) of HeLa Kyoto cells under different confinement heights. Dashed line indicates the confinement height that corresponds to blebs appearance. Error bars represent standard deviation. (**G**) Volume of single HeLa Kyoto cells during dynamic confinement experiment. Dashed line indicates the moment of confinement. (**H**) FXm images of HeLa Kyoto cells during dynamic confinement experiment taken with high NA objective. (**I**) *Top*: Volume of single DC migrating in collagen. *Bottom*: Corresponding FXm images. (**J**) Volume of single DCs migrating in collagen with the different speeds, N=1. *Left*: <0.5 µm/min (n=14), *middle*: >0.5 µm/min (n=10) and <1 µm/min, *right*: >1 µm/min (n=19). (**K**) Coefficient of variation of volume flux dV/dt computed for 10 min intervals during single DCs migration in collagen plotted versus their average speed (n=43, N=1). The line is the fit for coefficient of variation of DCs volume using best-fit parameters (see Appendix 1).

The online version of this article includes the following video, source data, and figure supplement(s) for figure 7:

**Figure supplement 1.** Characterisation of cell protrusions and dry mass in ultra-fast HeLa cell flattening and volume modulation in ultra-fast flattening of RPE1 and HEK-293 cells.

**Figure supplement 1—source data 1.** Data tables related to quantifications in Figure 7—figure supplement 1.

**Figure 7—video 1.** 3D-membrane reconstruction of HeLa expressing MyrPalm-GFP (black) cells cell shape under different confinement heights 63×. https://elifesciences.org/articles/72381/figures#fig7video1

**Figure 7—video 2.** Z-planes of control and 2 µM Lat A treated HeLa-MYH9-GFP-LifeAct-mcherry, cells under 20 µm and 7.6 µm confinement heights. https://elifesciences.org/articles/72381/figures#fig7video2

**Figure 7—video 3.** FXm imaging of HeLa EMBL cells during dynamic confinement recorded with 20× LD. https://elifesciences.org/articles/72381/figures#fig7video3

**Figure 7—video 4.** FXm imaging of HeLa EMBL cells during dynamic confinement recorded with 20× PA. https://elifesciences.org/articles/72381/figures#fig7video4

**Figure 7—video 5.** FXm imaging of DCs migrating in collagen gel with 20× LD. https://elifesciences.org/articles/72381/figures#fig7video5

the formation of large blebs (*Figure 7—figure supplement 1E*). This loss of volume induced by fast confinement was also found in other cell types, RPE-1 and HEK-293 (*Figure 7—figure supplement 1F, G*) and was also previously observed in confined *Dictyostelium* cells (*Srivastava et al., 2020*). Overall, these experiments show that fast imposed cell deformation induces an actin-dependent loss of volume (up to 30%), at almost constant surface area.

To better estimate the speed of deformation imposed by the confiner, we imaged at high frame rate during the confinement process. It showed that, even with a time lapse of 30 ms, the volume loss happened between two consecutive frames (*Figure 7G* a *Figure 7—figure supplement 1H, I* and *Figure 7—video 4*). Only volume is lost and not dry mass (*Figure 7—figure supplement 1J*), which suggests that only water and probably small solutes are lost. Nevertheless, the speed of volume change is not compatible with our mechanosensitive PLM, as in this model, volume loss occurs in the minutes timescale due to changes in ion transport rates. Fast imaging of the fluorescent medium surrounding the cells used for FXm indeed showed a transient appearance of streams of darker fluid (non labeled, thus coming from the cells) emanating from confined groups of cells (*Figure 7H* and *Figure 7—video 4*), likely corresponding to the expelled water and osmolites. Overall, these confinement experiments suggest that, although at this timescale of milliseconds, the mechanism of volume loss very likely differs from the context of spontaneous cell spreading, it is also induced by an increase in membrane tension, and requires the presence of the actin cytoskeleton.

Within the PLM framework, for a given osmolarity of an external medium, the cell volume may change either due to a change in hydrostatic or osmotic pressure. Fast compression can increase the cortical tension, which can cause an increase in hydrostatic pressure of the cell. However, the maximum hydrostatic pressure in the cell before the membrane detaches from the cortex is of the order of $10^2$ Pa, thus producing no direct effect on the cell volume, as discussed before (*Cadart et al., 2019*; *Salbreux et al., 2012*). Hence, the observed volume loss of 10–15% can only be

due to a change in the osmolarity of the cell, and not to a change in hydrostatic pressure. For ion transport to take place at timescales of milliseconds, the transport rates of channels and pumps would need to increase by 4 orders of magnitude. Such an increase can be easily attained if the high tension upon compression leads to transient formation of pores in the plasma membrane (observed in spreading GUVs; *Karatekin et al., 2003*). If these pores are small enough to allow for free ion transport but do not let the larger molecules trapped in the cell pass through (which should be the case since the dry mass was found to remain constant), the cell volume will increase rather than decrease (a consequence of the Donnan effect; *Sperelakis, 2012*). The formation of pores thus cannot explain our observations. Another mechanism that may lead to volume decrease upon compression without losing the trapped osmolites requires a selective increase of the ion conductance upon compression, but by orders of magnitude. Whether the ion conductance can increase by 4 orders of magnitude by mechanical stretching requires further investigation. Finally, it is also possible that due to its poroelastic nature (*Moeendarbary et al., 2013*; *Charras et al., 2009*), the cytoplasm behaves as a gel-like structure, and that water and osmolites are pressed out of the cell upon confinement, without changing the osmotic balance nor the dry mass (*Sachs and Sivaselvan, 2015*). In conclusion, confinement experiments confirm that fast deformation is associated with volume loss in an actin-dependent way, also suggesting a coupling between cell mechanics and volume regulation. However, they are hard to fully interpret in physical terms. This means that such a simple experiment as squeezing a cell cannot yet be understood with the current general knowledge on cell biophysics, pointing to a need for further investigations of the physics of large cell deformations. Such deformations are likely to occur in physiological contexts such as circulation of white blood cells and circulating cancer cells through small capillaries and may lead to volume change as was shown in vitro (*Liu et al., 2020*).

## Volume fluctuations in fast migrating immune cells can be explained by the mechanosensitive PLM

While our mechanosensitive PLM might be limited in the interpretation of cell deformations occurring below the second timescale, it captures well the larger timescales, based on a modulation of ion fluxes by membrane tension. Such timescales correspond to deformations that cells experience, for example, as they migrate through dense tissues. This implies that migrating cells might display volume fluctuations. To test this prediction, we used a classical cell migration assay with fast-moving bone marrow-derived dendritic cells (DCs) from mice embedded in a collagen gel (*Vargas et al., 2016*). The collagen gel mixed with fluorescent dextran was assembled inside a cell volume measurement chamber (*Figure 7—figure supplement 1K* and *Figure 7—video 5*). Because of the low fraction of collagen in the solution and the homogeneity of the fluorescent background, regular FXm measurements could be performed. We observed that the cell volume changed by a few percent as single cells moved through the collagen gel (*Figure 7I*), with periods of cell protrusion corresponding to a decrease in cell volume. To assess whether these fluctuations in volume were related to the migration of cells, we split individual cells into three groups according to their average speed and plotted their volume (in %) as a function of time (*Figure 7J*). This clearly showed that faster moving cells displayed larger volume fluctuations. Finally, to get a more quantitative assessment of the correlation, we plotted the coefficient of variation of the volume against the speed (*Figure 7K*), for single cells shown in (*Figure 7J*). Faster cells displayed more volume fluctuations. Interestingly, this relation was well fitted by an extension of the model to cells moving through a meshwork (fit in *Figure 7K*, and see Appendix 1 for the model extension). This experiment suggests that the mechano-osmotic coupling that we describe in our study is at work in migrating cells, inducing larger volume fluctuations (and thus larger density changes) in faster migrating cells. These volume and density fluctuations could thus be present in a large range of cells in physiological conditions, with yet unknown consequences on cell physiology and behavior.

Beyond the potential functional significance of volume and density fluctuations associated with cell shape changes, our observations and our model demonstrate that a membrane tension homeostasis mechanism is constantly at work in mammalian cells. This mechanism is most likely due to crosstalk between mechanical, osmotic, and electrical properties of the cell pointing to the importance of

taking into account complex coupling between various physical parameters to understand cellular physics and physiology.

# Materials and methods

## Key resources table

| Reagent type (species) or resource | Designation | Source or reference | Identifiers | Additional information |
|---|---|---|---|---|
| Cell line (*Homo sapiens*) | HeLa EMBL (Kyoto) | Gift from Valérie Doye | | |
| Cell line (*H. sapiens*) | RPE1 | ATCC | | |
| Cell line (*H. sapiens*) | 3T3 | ATCC – from Alba Diez-Munoz Lab, EMBL, Heidelberg, Germany | | |
| Cell line (*H. sapiens*) | HEK-293 | Gift from Liam Holt lab, NYU, NewYork | | |
| Chemical compound, drug | Fetal bovine serum | PAN-Biotech | P30-193306 | Use at 10% |
| Chemical compound, drug | Dextran, Alexa Fluor 647; 10,000 MW, Anionic, Fixable | Sigma-Aldrich | D22914 | Stock at 10 mg/ml in PBS |
| Chemical compound, drug | Fluorescein isothiocyanate-dextran; 10,000 MW | Sigma-Aldrich | FD10S | Stock at 50 mg/ml in PBS |
| Chemical compound, drug | Fibronectin | Sigma-Aldrich | F1141-1MG | 50 µg/ml in PBS |
| Chemical compound, drug | Poly-L-lysine | Sigma-Aldrich | P8920 | Use at 0.01% |
| Chemical compound, drug | PLL-PEG | SuSoS | | 0.1 mg/ml solution in HEPES |
| Software, algorithm | Software for FXm image analysis and volume calculation | Available upon request to the authors | RRID:SCR_001622 | |

## Cell culture and drug treatment

HeLa EMBL (Kyoto), and derived cell lines HeLa LifeAct, HeLa Myrpalm-GFP-LiFeact mCherry, Hela hgem-mCherry, RPE-1, 3T3-ATCC, and HEK-293 cells were maintained in Dulbecco's modified Eagle medium with Glutamax (DMEM/Glutamax; Gibco) supplemented with 10% fetal bovine serum (FBS; Biowest or PAN-Biotech) and 1% penicillin-streptomycin solution Thermo Fisher Scientific, and stored at 37°C and 5% $CO_2$. All cell lines were regularly tested for mycoplasma contamination.

Bone marrow-derived DCs were obtained by differentiation of bone marrow precursors for 10 days in DCs medium (IMDM-Glutamax, FCS 10%, pen-strep 100 U/ml, and 2-ME 50 µM) supplemented with granulocyte-macrophage colony-stimulating factor (GM-CSF)-containing supernatant (50 ng/ml) obtained from transfected J558 cell line, as previously described (*Barbier et al., 2019*).

Latrunculin A (Sigma-Aldrich, used in the final concentrations: 100 nM, 2 or 5 µm), CK-666 (Sigma-Aldrich, used in the final concentrations: 100 µM), EIPA (Tocris Bioscience, used in the final concentrations: 50 µM), NSC668394 (Sigma-Aldrich, used in the final concentrations: 20 µM) dissolved in DMSO (Sigma-Aldrich), Y-27632 (Tocris Bioscience, used in the final concentrations: 100 µM), and GdCl3 (Sigma-Aldrich, used in the final concentrations: 100 µM) dissolved in $H_2O$. Manganese(II) chloride solution (Sigma-Aldrich) was used in the final concertation of 1 mM. Incubations with drugs were done for suspended cells 30 min prior to experiments.

For volume measurements, 10 kDa dextran conjugated with different fluorophores were used in the final concentration of 1 mg/ml: fluorescein isothiocyanate-dextran (Sigma-Aldrich) or Alexa Fluor 647 (Thermo Fisher Scientific).

For serum starvation experiments, plated cells were incubated overnight in DMEM without FBS. Prior to the experiments, cells were detached with EDTA and resuspended in the DMEM without FBS collected from cells or in the fresh DMEM supplemented with 10% FBS and incubated for 30 min in suspension.

## Cell cycle stage detection

The cell cycle state of the cells is indicated by the expression of h-Geminin protein which is expressed by cells from the start of S phase until mitosis (*Sakaue-Sawano et al., 2013*) in HeLa hgem-mCherry cell line. To quantify the fluorescence of geminin in the nucleus, first, a background subtraction is

performed on the images using the ImageJ software. An ROI is used to define an area containing the background fluorescence in the image. An average value of the ROI is then subtracted from all the frames. Subsequently, an ROI is drawn to drawn as close to the cell, as possible, and then the mean gray value is measured across all the frames.

## Monitoring of cell volume and contact area while spreading

PDMS chambers were prepared as described in *Cadart et al., 2017*. The typical height of PDMS chambers for volume measurements was 20 μm. PDMS chambers were incubated with 50 μg/ml fibronectin (Sigma-Aldrich) in phosphate-buffered saline (PBS) for 1 hr, washed and incubated overnight with culture medium. Cells were detached with warm Versen (Gibco) and resuspended in medium collected from cells to facilitate spreading.

In case of measurements of non-adherent cells, we used chambers incubated with PLL-PEG coating (0.1 mg/ml solution in HEPES, SuSoS), washed and incubated overnight with culture medium without FBS. Cells were detached with Trypsin and resuspended in a fresh culture medium.

The cell volume measurement explained in details in *Cadart et al., 2017* and used in were coupled with spreading area measurement performed by RICM (*Rädler and Sackmann, 1993*; *Cuvelier et al., 2007*). Microscopy was performed at 37°C with 5% $CO_2$ atmosphere. Imaging was started immediately after cell injection into the chamber with 1-min time interval. Imaging was performed using a ZEISS Z1 Observer epi-fluorescence microscope equipped with an Orca-Flash 4 Camera (Hamamatsu), 20× Plan-Apochromat objective, NA0.8 and the software Metamorph (Molecular Devices).

The volume extraction was performed with a MatLab software as described in *Cadart et al., 2017*.

The analysis of spreading and contact area was performed manually using the ImageJ software. The borders of the cell were delimited manually and then the area, and different shape descriptors were extracted. For the volume and spreading area data, first experimental point was taken not as experimental time point t=0, but at the first point where contact area was detectable by RICM and not exceeding 100 μm². For average volume and spreading area during spreading values of different cells were averaged at every time point. As the measurements of spreading area and volume were done not at every time point, start time point differs from cell to cell, and because of the different duration of the experiments, averaging leads to the appearance of outliers that were deleted manually. For better visualization of the experiments, we used continuous standard deviation at the graphs instead of error bars by using matplotlib.axes.axes.fill_between. For better visualization to avoid the gaps in the continuous standard deviations, we used the average of closest points in case a value was missed by the reasons explained above.

For HeLa and 3T3 cells, initial speed of spreading $\frac{dA}{dt}$ and volume flux $\frac{dV}{dt}$ was calculated as linear slope in the first 10 min after measurable cell to substrate contact. For RPE-1 cells, initial $\frac{dA}{dt}$ and $\frac{dV}{dt}$ were calculated as linear slopes in the 10 min prior to the time point when spreading area is equal to cross-section area of cell in initial non-spread state, and in the first 10 min after that time point.

Measurements of cell volume and spreading area at the level of population (and for micropatterns experiments) were done 4 hr after cell seeding.

## Micropatterning

Cells were patterned using the existed technique (*Azioune et al., 2011*) or PRIMO (Alveole) in case of tether pulling experiments.

## Side-view microscopy

Glass slide was attached to glass bottom dish by UV-glue, the position of glass was slightly tilted from perpendicular to the dish bottom. Glass was coated with fibronectin and washed with medium. Cells were detached with Versen and resuspended in warm medium collected from cells and incubated for 30 min. Then drop of cell was added to the dish, close to the angle between dish bottom and attached glass. Dish was placed to the incubator for 2 min to allow cell initial attachment to the tilted glass. Then 2 ml of medium collected from cells was added to the dish and microscopy started with a time frame of 1 min. Imaging was performed using a ZEISS Z1 Observer epi-fluorescence microscope 20× NA0.4.

## Monitoring of cell volume during cell migration in the collagen

Collagen mix was prepared on ice to delay polymerization: 25 µl 10× PBS + 25 µl culture medium + 55 µl collagen + 140 µl culture medium with DCs (2*10⁶ /ml) + 5 µl FITC-dextran + 1.3 µl NaOH.

Immediately after mixing, suspension was added into PDMS chamber for volume measurements with height of 12 µm. Microscopy was started ~10 min after injection. Imaging was performed using a ZEISS Z1 Observer epi-fluorescence microscope equipped with 20× NA0.8.

Cell velocity during migration in collagen gel was calculated for 10 min intervals. Cell position was defined as a center of mass of a binary mask applied on FXm images of cells.

## Monitoring of cell volume during osmotic shock

PDMS chambers were coated with 0.01% PLL (Sigma-Aldrich) to prevent cell detachment during changing medium and maintaining cell round shape during experiment, then washed and incubated overnight with culture medium without FBS. Cells were detached with Trypsin. Isoosmotic medium was exchanged to the medium with known osmolarity typically for 2.5 min after beginning of acquisition. Full medium exchange in the chamber takes less than 1 s. Imaging was performed using a ZEISS Z1 Observer epi-fluorescence microscope equipped with 20× NA0.4, and 20× NA0.8 in case of the stream movies. Hypoosmotic solutions were made by water addition to culture medium, hyperosmotic by addition PEG400. Osmolarity of working solutions was measured by osmometer Type 15 M (Löser Messtechnik).

Cell rupture in response to distilled water exposure was monitored by PI (1 µg/ml) (Sigma-Aldrich) intensity inside the cell.

Volume flux for passive response to osmotic shock was defined as a linear slope at the linear region of volume curves defined manually.

Adaptation speed for osmotic shock recovery was calculated as a linear slope starting from the minimum or maximum volume value achieved during passive response (for hyper or hypoosmotic shock) at 5-min interval.

## Monitoring of cell volume under confinement

Cells were detached with Trypsin and resuspended in fresh culture medium. Both static six-well confiner and dynamic confiner were used according to experimental procedure described in *Le Berre et al., 2014*. Imaging was performed using a ZEISS Z1 equipped with 20× long-distance objective NA0.4.

For volume measurements performed with dynamic confiner experiments, bottom glass was coated with 0.01% PLL, that prevented cell escape from the field of view and allowed following the same cells before and after confinement.

Calculation of surface area of non-confined cell was done with the assumption of spherical cell shape, and of confined cells with the assumption of cylindrical cell shape, based on measured cell volume.

## Spinning disk microscopy

Qualitative imaging for osmotic shock and confinement experiments was performed with spinning disk set-up (Leica DMi8). 63× and 100× oil objectives were used. CellMask (Invitrogen) staining was performed in warm PBS solution (1 µl of dye to 1000 µl PBS).

Filopodia were manually segmented. Filopodia density is plotted as number of filopodia per µm of cell body diameter. Bleb was manually segmented from middle plane images. For membrane density measurements on cell contour, cells were background substrated, and resliced by their contour, where most of the membrane marker accumulates. An average projection was plotted for 3 µm around the cell edge.

## Dry mass measurements

Mass measurement was performed by quantitative phase microscopy using Phasics camera (*Aknoun et al., 2015*). Images were acquired by Phasics camera every 15 min for 35 hr during the duration of the experiment. To get the reference image, 32 empty fields were acquired on the PDMS chips and a median image was calculated. Custom MATLAB scripts were written by Quantacell for analysis of interferograms (images acquired by phasics). The interferograms were associated with reference

images to measure the optical path difference and then separated into phase, intensity and phase cleaned images (background set to 1000 and field is cropped to remove edges). Background was then cleaned using gridfit method and a watershed algorithm was used to separate cells that touch each other. Mass was then calculated by integrating the intensity of the whole cell.

## Tether pulling

For apparent membrane tension measurements, tether force was measured with single-cell atomic force spectroscopy by extruding tethers from the plasma membrane on top of the nucleus of HeLa EMBL cells. Cellview glass bottom dishes (Greiner) were coated for 1 hr with fibronectin (50 µg/ml; Sigma-Aldrich). Cells were incubated for 30 min with in the presence of drugs or vehicle, then plated, and probed either during spreading (from 30 to 90 min after plating) or at steady state (fully spread; from 4 to 5 hr after plating). To perform experiments on non-spread cells, fibronectin-coated circles (Ø 20 um) were micropatterned onto Cellview glass bottom dishes (Greiner) using PRIMO (Alveole) following the manufacturer's recommendations.

Tether extrusion was performed on a CellHesion 200 BioAFM (Bruker) integrated into an Eclipse Ti inverted light microscope (Nikon). OBL-10 Cantilevers (spring constant ~60 pN/nm; Bruker) were mounted on the spectrometer, calibrated using the thermal noise method (reviewed in *Houk et al., 2012*) and coated for 1 hr at 37°C with 2.5 mg/ml Concanavalin A (Sigma-Aldrich), which binds poly-saccharides expressed on the surface of the cell (*Goldstein and So, 1965*). Before the measurements, cantilevers were rinsed in PBS and cells were washed and probed in DMEM/Glutamax (Gibco) supplemented with 2% FBS (Life Technologies) and 1% penicillin-streptomycin solution (Life Technologies). Measurements were run at 37°C with 5% $CO_2$ and samples were used no longer than 1 hr for data acquisition.

Tether force was measured at 0 velocity, which is linearly proportional to apparent membrane tension, assuming constant membrane bending rigidity (*Hochmuth et al., 1996*). In brief, approach velocity was set to 0.5 µm/s while contact force and contact time ranged between 100 and 200 pN and 100 ms to 10 s, respectively. The latter two parameters were experimentally tuned before every tether pulling attempt, aiming to reach a tradeoff between the maximization of the probability to extrude single tethers, and the reduction of experimental stress on the cells. The larger the contact time and force, the higher is the probability of formation of bonds between the molecules of Concanavalin A on the surface of the cantilever and the polysaccharides on the surface of the cell. On the other side, the lower those two parameters, the lower is the stress experienced by the cell during the contact with the cantilever. As a general trend, contact force and time must be increased over the course of the experiment, owing to the depletion of Concanavalin A from the cantilever (*Krieg et al., 2008*).

To ensure tether force measurement at 0 velocity, after contacting the cell surface, the cantilever was retracted for 10 µm at a velocity of 10 µm/s. The position was then kept constant for 30 s and tether force was recorded at the moment of tether breakage at a sampling rate of 2000 Hz. Each tether extrusion attempt lasted about 3 min and each cell was probed until three single tethers were successfully extruded or for a maximum of 10 min.

Force-time curves resulting from successful tether extrusions were analyzed using the JPK Data Processing Software. Tether force values from tethers extracted from the cell were then averaged, and each cell was accounted as a single data point.

## Traction force measurements

Force measurements were conducted directly after seeding the cells on the sample and spreading was observed for 90 min on an inverted microscope (Nikon Ti-E2) with a Orca Flash 4.0 sCMOS camera (Hamamatsu) and a temperature control system set at 37°C. To avoid shaking the cells during stage movement, a POC-R2 sample holder in closed perfusion configuration was used and cells were seeded with a syringe right before image acquisition. The medium was supplemented with 20 mM of HEPES in order to buffer the pH during the experiment. Force measurements were performed using a method described previously (*Tseng et al., 2011*). In short, fluorescent beads were embedded in a polyacryl-amide substrate with 20 kPa rigidity and images of those beads were taken during cell spreading. The first frame, before cells started attaching to the substrate, served as unstressed reference image. The displacement field analysis was done using a homemade algorithm based on the combination of particle image velocimetry and single-particle tracking. After correcting for experimental drift, bead

images were divided into smaller subimages of 20.7 µm width. By cross correlating the subimages of the stressed and the unstressed state, mean displacement of the subimage can be measured. After correcting for this displacement, the window size is divided by 2 and the procedure is repeated twice. On the final subimages, single-particle tracking was performed to obtain a subpixel resolution displacement measurement. From the bead displacement measurements, a displacement field was then interpolated on a regular grid with 1.3 µm spacing. Cellular traction forces were calculated using Fourier transform traction cytometry with zero-order regularization (*Sabass et al., 2008*; *Milloud et al., 2017*), under the assumption that the substrate is a linear elastic half-space and considering only displacement and stress tangential to the substrate. To calculate the strain energy stored in the substrate, stress and displacement field were multiplied with each other and with the grid pixel area and then summed up over the whole cell. All calculations and image processing were performed with MATLAB.

## Electron microscopy

Hela cells plated on fibronectin-coated glass coverslips for 30 min were disrupted by scanning the coverslip with rapid sonicator pulses in KHMgE buffer (70 mM KCl, 30 mM HEPES, 5 mM MgCl2, 3 mM EGTA, and pH 7.2). Paraformaldehyde 2%/glutaraldehyde 2%-fixed cells were further sequentially treated with 0.5% OsO4, 1% tannic acid, and 1% uranyl acetate prior to graded ethanol dehydration and Hexamethyldisilazane substitution (HMDS, Sigma-Aldrich). Dried samples were then rotary-shadowed with 2 nm of platinum and 5–8 nm of carbon using an ACE600 high vacuum metal coater (Leica Microsystems). Platinum replicas were floated off the glass with 5% hydrofluoric acid, washed several times by floatation on distilled water, and picked up on 200 mesh formvar/carbon-coated EM grids. The grids were mounted in a eucentric side-entry goniometer stage of a transmission electron microscope operated at 80 kV (Philips, model CM120) and images were recorded with a Morada digital camera (Olympus). Images were processed in Adobe Photoshop to adjust brightness and contrast and presented in inverted contrast.

## Statistical analysis

Error bars represent standard deviation and in some cases standard error, which is specified in the legend. When applicable Shapiro-Wilk test was used to test for normality of data. Student's t-test was chosen for statistical testing of normal distributed data, while Mann-Whitney U-test was performed on non-normal distributed data.

# Acknowledgements

The authors thank Jian Shi, Rafaële Attia, and Li Wang for help with photolithography; Pierre Recho and Romain Rollin for fruitful discussions; Pablo Vargas for help with DCs experiment; Olivier Thouvenin and Matthieu Maurin for assistance with optical details; Vincent Frasier for help with 3D-reconstruction of confined cells. This study was supported by a French Agence Nationale de la Recherche (ANR) grant to MP (ANR-19-CE13-0030). This study has also received the support of Institut Pierre-Gilles de Gennes-IPGG (Equipement d'Excellence, 'Investissements d'avenir,' program ANR-10-EQPX-34) and laboratoire d'excellence, 'Investissements d'avenir' program ANR-10-IDEX-0001-02 PSL and ANR-10-LABX-31. The authors also thank IPGG technical platform for providing equipment and technical assistance. LV has received funding from the European Union's Horizon 2020 Research and Innovation Program under the Marie Sklodowska-Curie Grant agreement no. 641639, and Fondation pour la Recherche Médicale (FDT201805005592). NS acknowledges support from the Human Frontier Science Program (LT000305/2018 L). ASV acknowledges the support of ANR under reference ANR-17-CE13-0020-02.

# Additional information

## Competing interests

Pierre Sens: Reviewing editor, *eLife*. The other authors declare that no competing interests exist.

## Funding

| Funder | Grant reference number | Author |
|---|---|---|
| Agence Nationale de la Recherche | ANR-19-CE13-0030 | Matthieu Piel |
| Agence Nationale de la Recherche | ANR-10-EQPX-34 | Matthieu Piel |
| Agence Nationale de la Recherche | ANR-10-IDEX-0001-02 PSL | Matthieu Piel |
| Agence Nationale de la Recherche | ANR-10-LABX-31 | Matthieu Piel |
| Fondation pour la Recherche Médicale | FDT201805005592 | Larisa Venkova |
| Human Frontier Science Program | LT000305/2018-L | Nishit Srivastava |
| Agence Nationale de la Recherche | ANR-17-CE13-0020-02 | Amit Singh Vishen |
| European Union's Horizon 2020 research and innovation programme | Marie Sklodowska-Curie grant agreement no. 641639 | Larisa Venkova |
| Human Frontier Science Program | | Nishit Srivastava |

The funders had no role in study design, data collection and interpretation, or the decision to submit the work for publication.

## Author contributions

Larisa Venkova, Conceptualization, Formal analysis, Investigation, LV performed the imaging, volume measurements, analysis for cell spreading experiments with help of DC, BD, NS and AD. LV performed the imaging, volume measurements, analysis for cell confinement and osmotic shock experiments. LV. and MP. conceived the study and designed experiments. LV and MP wrote the manuscript., Methodology, Software, Visualization, Writing – original draft, Writing – review and editing; Amit Singh Vishen, ASV developed the mechano-sensitive PLM model under the supervision of J-FJ and PS and fitted the model predictions to the experimental data, Conceptualization, Formal analysis, Investigation, Methodology, Software, Supervision, Visualization, Writing – original draft, Writing – review and editing, Lead author for the physical model; Sergio Lembo, Formal analysis, Investigation, Methodology, SL performed the tether force measurements under the supervision of AD-M, Supervision, Visualization, Writing – review and editing; Nishit Srivastava, Formal analysis, Investigation, Methodology, NS performed mass measurements and cell cycle experiments; Baptiste Duchamp, BD performed imaging, volume measurements and analysis of cell plated on micropatterns under the supervision of DC, Formal analysis, Investigation, Methodology, Supervision, Writing – review and editing; Artur Ruppel, AR performed the traction force experiments under the supervision of MB, Conceptualization, Formal analysis, Investigation, Methodology, Supervision, Visualization, Writing – review and editing; Alice Williart, AW and LV performed the experiments with the FLIPPER probe, and AW performed the image analysis, Formal analysis, Investigation; Stéphane Vassilopoulos, Conceptualization, Formal analysis, Investigation, Methodology, SV performed electron microscopy experiments, Visualization; Alexandre Deslys, Formal analysis, Investigation, Methodology; Juan Manuel Garcia Arcos, Conceptualization, Formal analysis, Investigation, JMGA performed the analysis of cell membranes under confinement, Methodology, Supervision, Validation, Writing – review and editing; Alba Diz-Muñoz, ADM supervised the tether pulling experiments, Conceptualization, Supervision, Validation, Writing – review and editing; Martial Balland, Conceptualization, MB supervised the traction force microscopy experiments, Methodology, Supervision, Validation, Writing – review and editing; Jean-François Joanny, Conceptualization, Formal analysis, Investigation, JFJ participated in establishing and supervising the physical modeling, Methodology, Supervision, Validation, Writing – review and editing; Damien Cuvelier, Conceptualization, DC supervised the micro-patterning experiments, Formal analysis, Investigation, Methodology, Validation, Writing – original

draft, Writing – review and editing; Pierre Sens, Conceptualization, Investigation, Methodology, PS supervised and coordinated the physical modeling, Supervision, Validation, Writing – original draft, Writing – review and editing; Matthieu Piel, Conceptualization, MP supervised and coordinated the study, Methodology, Project administration, Supervision, Validation, Writing – original draft, Writing – review and editing

**Author ORCIDs**
Larisa Venkova http://orcid.org/0000-0001-5721-7962
Amit Singh Vishen http://orcid.org/0000-0001-8938-0852
Sergio Lembo http://orcid.org/0000-0002-2253-8771
Nishit Srivastava http://orcid.org/0000-0003-4177-6123
Stéphane Vassilopoulos http://orcid.org/0000-0003-0172-330X
Alba Diz-Muñoz http://orcid.org/0000-0001-6864-8901
Jean-François Joanny http://orcid.org/0000-0001-6966-3222
Pierre Sens http://orcid.org/0000-0003-4523-3791
Matthieu Piel http://orcid.org/0000-0002-2848-177X

**Decision letter and Author response**
Decision letter https://doi.org/10.7554/eLife.72381.sa1
Author response https://doi.org/10.7554/eLife.72381.sa2

## Additional files

**Supplementary files**
• Transparent reporting form

**Data availability**
All data generated or analysed during this study are included in the manuscript and supporting file; all the raw analysed data shown in the figure panels in the article are available in the accompanying Source Data files.

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

## Appendix 1

### Extended PLM for cell volume

The volume of the cell changes due to water flux driven by the difference of the osmotic and hydrostatic pressure. The volume dynamics is given by

$$\frac{dV}{dt} = -L_p A \left( \left( p\left(V\right) - p_{\text{ext}} \right) - \left( P\left(V\right) - P_{\text{ext}} \right) \right) \tag{A1}$$

where $L_p$ is the cell's hydraulic conductivity, is the total surface area, $p_{\text{ext}}$ and $P_{\text{ext}}$ are the hydrostatic and osmotic pressure of the external medium, respectively. The volume dependence of the hydrostatic $p\left(V\right)$ and osmotic pressure $P\left(V\right)$ of the cell is explicitly shown. The volume dependence of hydrostatic pressure is through the force balance at the cell surface (*Cadart et al., 2019*) in which both tension and curvature are functions of cell size. In the dilute limit, the osmotic pressure is $P\left(V\right) = k_B T \left( \sum_{i=1}^{N} X_i + \sum_{i=1}^{M} Y_j \right)$, where $T$ is the temperature, $X_i$ is the concentration of solute species that are either actively or passively transported across the cell, and $Y_j$ is the concentration of impermeant solutes. The concentration of the trapped solute is $Y_j = y_j / \left( V - V_{\text{solid}} \right)$ where $y_j$ is the total number of molecules of type $j$ in the cell, and $V_{\text{solid}}$ is the solid volume of the cell that is inaccessible to the solute molecules. The solid volume is essentially the sum of the volume taken up by the proteins and the DNA. The osmotic pressure of the external medium is $P_{\text{ext}} = k_B T \left( \sum_{i=1}^{N} X_{0i} + \sum_{j=1}^{M} Y_{0j} \right)$.

In *Equation A1*, the contribution of the hydrostatic pressure to the volume (or the volume change) is negligible in comparison to the osmotic pressure, and this even for small variation near the steady-state volume. Here is how we reach this conclusion: the concentration of the trapped osmolytes and its counterions is estimated to be about 100 mM, the osmotic pressure corresponding to this is about $10^5$ Pa. Taking the cortical tension to be of order of 1 mN/m and cell size to be about 10 µm, we get the hydrostatic pressure difference of about 100 Pa. A 10% decrease in cell volume will increase the osmotic pressure of the trapped osmolytes by $10^4$ Pa. For this osmotic pressure to be balanced by an increase in the hydrostatic pressure, the cortical tension would need to increase by a factor of 100, which we consider to be too large for this term to be relevant. Therefore, we find it reasonable to ignore the contribution of the hydrostatic pressure difference in the water flux equation. In this limit the volume dynamics is given by

$$\frac{dV}{dt} = L_p A k_B T \left( \sum_{i=1}^{N} \left( X_i - X_{0i} \right) + \sum_{j=1}^{M} \left( Y_j - Y_{0j} \right) \right). \tag{A2}$$

The main solutes that are transported across the cell membrane are sodium, potassium, and chloride (*Kay, 2017*). The sodium and potassium are actively transported through sodium-potassium ($Na^+/K^+$) pumps. The pump transports two potassium ions into the cell in exchange for three sodium ions transported out of the cell. This leads to the enrichment of potassium inside the cell and sodium outside the cell. Along with the $Na^+/K^+$ pump, there are various cotransporters and channels that passively transport the other ions - chloride, hydrogen ion, carbonates, and so on. A general expression of the ion transport including different cotransporters reads

$$\frac{d\left[ \left( V - V_{\text{solid}} \right) X_i \right]}{dt} = -\sum_j \Lambda_{ij} \left( k_B T \log \frac{X_j}{X_{0j}} + z_j \Delta\Phi \right) + S_i, \tag{A3}$$

where the first term is a passive flux of ion $i$ due to the electrochemical potential difference of ion $j$ (summed over all $j$), $\Lambda_{ij}$ is element of a symmetric matrix that couples the flux of the ions $i$ to the chemical potential difference of ion $j$, $\Delta\Phi$ is the electric potential energy difference between the inside and the outside of the cell, $z_j$ is the electric charge of species $j$. The second term, $S_i$ is the source term due to active pumping. The non-diagonal terms of the transport matrix $\Lambda$ are due to the co-transporters (like NKCC, NHE, KCl, etc.) that couples the flux of ion $j$ to that of $i$. In general, the permeability $\Lambda_{ij}$ and $S_i$ depend on the concentration of ions, electric potential difference, and membrane tension. Their volume dependence may be due to the transporters' mechanosensitivity or due to feedback from signaling molecules that are sensitive to volume change.

The membrane potential is determined using the electroneutrality condition in the cell: $\sum_{i=1}^{N} z_i X_i + \sum_{j=1}^{M} z_j Y_j = 0$. Using the electroneutrality condition and *Equation A3*, the potential is given by

$$\Delta\Phi = -\frac{\sum_{i=1}^{N} z_i \left(\sum_{k=1}^{N} \Lambda_{ik} \log\left(\frac{X_k}{X_{0k}}\right) + S_i\right)}{\sum_{i=1}^{N}\sum_{k=1}^{N} \Lambda_{ik} z_i z_k}. \tag{A4}$$

Grouping the positive and negative ions and taking the trapped solute to be negatively charged, we get

$$\sum_{i=1}^{N^-} |z_i| X_i^- + \sum_{j=1}^{M} |z_j| Y_j = \sum_{k=1}^{N^+} |z_k| X_k^+, \tag{A5}$$

where $N^-$ is the number of negative ionic species and $N^+$ is the number of positive ion species. For monovalent ions, substituting *Equation A5* in *Equation A2* we get the steady-state volume as

$$V(t) - V_{\text{solid}} = \frac{\sum_{j=1}^{M} (|z_j|+1) y_j(t)}{\left(\beta P_{\text{ext}} - 2\sum_{i=1}^{N^-} X_i^-(t)\right)}, \tag{A6}$$

where $P_{\text{ext}} = k_B T \left(\sum_{i=1}^{N} X_{0i} + \sum_{j=1}^{M} Y_{0j}\right)$, and $\beta = 1/k_B T$.

## Fast timescales: cell as an osmometer

At the isotonic steady-state condition, the cell volume as given by *Equation A6* is

$$V_{\text{iso}} - V_{\text{solid}} = \frac{\sum_{j=1}^{M} (z_j+1) y_j}{\left(\beta P_{\text{iso}} - 2\sum_{i=1}^{N^-} X_i^-\right)}, \tag{A7}$$

where $P_{\text{ext}} = P_{\text{iso}}$ is the osmotic pressure of the external medium at the isotonic condition. For hypotonic shock the medium is diluted leading to osmotic pressure $P_{\text{ext}} = r_{\text{hypo}} P_{\text{iso}}$ where $r_{\text{hypo}}$ is the dilution factor, and for hyperosmotic shock, PEG400 is added to the external medium leading to osmotic pressure $P_{\text{ext}} = r_{\text{hyper}} P_{\text{iso}}$, where $r_{\text{hyper}} = (P_{\text{iso}} + P_{\text{PEG}})/P_{\text{iso}}$. After the osmotic shock, the cell volume changes in response to the new extracellular osmolarity. There is a clear separation in the timescale between water transport and ion transport (*Cadart et al., 2019*). The ion concentration changes over minutes whereas the volume change due to water flux is on the timescale of seconds. This timescale separation divides the volume dynamics into 'fast' passive response, in which the water flows in and out of the cell with constant number of ions within the cell, and 'slow' response, in which the ions are transported across the cell. Over the timescale of seconds, the number of ions inside the cells is constant, that is, $X_i (V - V_{\text{solid}}) = X_{i(\text{iso})} (V_{\text{iso}} - V_{\text{solid}})$. Substituting this in *Equation A2*, we get

$$\frac{dV}{dt} = P_{\text{iso}} L_p A \left(\frac{R V_{\text{iso}}}{V - (1-R) V_{\text{iso}}} - \frac{P_{\text{ext}}}{P_{\text{iso}}}\right), \tag{A8}$$

where $R = (1 - V_{\text{solid}}/V_{\text{iso}})$. Thus, we see that the volume dynamics is well approximated by the Van't-Hoff relation with a fixed number of solutes in the cell. This equation at steady state gives the maximum (minimum) volume $(V_m)$ after the fast hypoosmotic (hyperosmotic) shock. At steady state, we get the Ponder's relation $V_m/V_{\text{iso}} = R P_{\text{iso}}/P_{\text{ext}} + (1 - R)$. From *Equation A8*, we get the rate of volume change just after the shock as

$$\frac{dV}{dt}\big|_{t\to 0} = P_{\text{iso}} L_p A_{\text{iso}} \left(1 - \frac{P_{\text{ext}}}{P_{\text{iso}}}\right). \tag{A9}$$

Comparing *Equation (A9)* with the experimentally measured rate of volume increase just after the shock, we calculated $L_p$ (*Figure 4—figure supplement 1C*). The cell volume over the minute's timescale can be changed from this osmotic shock value by tuning the ion channels and pumps. We first consider the case when the ion transport does not change before and after the osmotic shock. The hypo-osmotic shock in the experiments is attained by dilution, we can see from *Equation A4* that the membrane potential is constant, and from *Equation A3*, we see that the right hand side remains constant. This implies that after the fast increase in volume, the ions reach a new concentration, which is their steady state value, hence, there is no further ion flow at long timescale. However, in the experiments, it has been observed that the cell goes through a volume decrease. This necessarily implies a feedback mechanism for regulatory volume decrease.

In hyperosmotic shock condition, the membrane potential does change; hence, at long times, the volume of the cell increases to a value larger than the maximum decrease. This value is still less than the isotonic volume. The experiments show almost perfect adaptation, implying a regulatory volume increase.

## Mechano-osmotic mechanism for cell volume regulation

There is a clear timescale separation between fluid flow, which is of the order of seconds, and spreading kinetics, which is of the order of minutes. Hence, over the timescale of spreading the cell is in osmotic balance with the external medium. The volume of the cell changes quasi-statically with the change in ion concentration according to *Equation (A6)*. The change in volume can be due to change in the number of impermeant ions, or due to a change in the concentration of negatively charged ions. The rate change of the volume, obtained by taking the time derivative of *Equation A6* and linearizing is

$$\frac{1}{V_{\text{iso}}}\frac{dV}{dt} = \frac{1}{V_{\text{iso}}}\frac{dV_{\text{solid}}}{dt} + \frac{R}{\sum_{j=1}^{M}(z_j+1)y_{j(\text{iso})}}\sum_{j=1}^{M}(z_j+1)\frac{dy_j}{dt} + \frac{2R}{\left(\beta P_{\text{iso}}-2\sum_{i=1}^{N^-}X_{i(\text{iso})}^-\right)}\sum_{i=1}^{N^-}\frac{dX_i^-}{dt} \qquad \text{(A10)}$$

where the first and second term on the right are due to a change in the number of impermeable molecules in the cell due to growth and the third term is due to the change in ion concentration due to change of ion transport rates. The volume-dependent feedback could affect either of the terms. We assume that the change in volume due to feedback is on the ion transport parameters, and the change in impermeable ions is only due to growth. We take the growth rate to be a $r_{\text{growth}}$ for all the trapped molecules, the growth in the number of trapped molecules also contributes to the growth in the volume of the solid fraction. For simplicity we take

$$\frac{dy_j}{dt} = r_{\text{growth}}\, y_j \quad and \quad \frac{dV_{\text{solid}}}{dt} = r_{\text{growth}} V_{\text{solid}}\,. \qquad \text{(A11)}$$

In the following we take $r_{\text{growth}}$ = 0.05/hr, which gives a doubling time of 20 hr.

Our hypothesis is that the cell shape changes upon spreading generate a transient increase in membrane tension; this increase in tension leads to the activation of mechanosensitive ion channels and induces ion flux leading to volume change.

Slow spreading would then induce a lower transient tension increase that would lead to a smaller volume loss. We use a simplified model that includes the mechanosensitivity of ion transport. Assuming that the ion transport parameters vary quasi-statically, we use the following phenomenological expression for the change in ion concentration:

$$\frac{2R}{\left(\beta P_{\text{iso}}-2\sum_{i=1}^{N^-}X_{i(\text{iso})}^-\right)}\sum_{i=1}^{N}\frac{dX_i^-}{dt} = \alpha\frac{1}{\gamma_{\text{iso}}}\frac{d\gamma_m}{dt}, \qquad \text{(A12)}$$

where the term on the right accounts for the change in ion concentration due to the feedback from membrane tension $\gamma_m$ on mechanosensitive ion transporters. This defines the mechanosensitivity parameter $\alpha$. Substituting *Equation A11* and *Equation A12* into *Equation A10* we get

$$\frac{d\delta V}{V_{\text{iso}}} = \alpha\frac{d\delta\gamma_m}{\gamma_{\text{iso}}}. \qquad \text{(A13)}$$

where $\delta V = V - V_{\text{iso}}\left(1 + r_{\text{growth}}t\right)$ and $\delta\gamma_m = \gamma - \gamma_{\text{iso}}$. Thus, we see that the change in volume is proportional to change in tension. For volume to decrease upon an increase of tension, the coefficient $\alpha$ should be negative. For a simplified model of transport of three ions—chloride, sodium, and potassium, we later show that $\alpha$ , in general, can take both positive and negative values. For the physiological value of the parameters, we find that $\alpha$ is negative if the increase in potassium permeability is much larger than that of sodium.

## Effect of mechano-osmotic coupling of volume recovery upon osmotic shock

We can ask if a purely mechano-osmotic coupling could be sufficient to give a volume recovery upon osmotic shock. The answer to that question is that the mechano-osmotic coupling will affect the volume dynamics after the shock, but it is not sufficient to give a volume recovery at long timescales. This is because at long times the membrane tension goes back to its homeostatic value ($\gamma_{\text{iso}}$) and

hence the ion transport parameters $\Lambda_{ij}$ and $S_i$ in **Equation A3** also recover their pre-shock values. However, since the osmolarity of the external medium is different even at long time the volume will not go back to the value before the shock. The mechano-osmotic coupling will thus affect the volume recovery dynamics, but not the asymptotic value of the volume.

## Sign of tension and volume coupling

We now solve for volume, electric potential, and $\alpha$, for a minimal model which only accounts for three permeant ions: sodium, potassium, and chloride. The external medium only includes these ions and the cytoplasm also includes an impermeant negatively charged species. Following references **Adar and Safran, 2020**; **Tosteson and Hoffman, 1960**; **Kay, 2017**; **Kay and Blaustein, 2019**, we consider a simplified model of ion transport, that is, we ignore the contribution of co-transporters to the ion flux and take the matrix $\Lambda$ in **Equation A3** to be diagonal. The dynamics of ion transport through the cell membrane as given by **Equation A3** now reads

$$\frac{d\left[\left(V - V_{\text{solid}}\right)Na^+\right]}{dt} = -\Lambda_{Na}\left(k_B T \log \frac{Na^+}{Na_0^+} + \Delta\Phi\right) - 3S_p,$$ 
(A14)

$$\frac{d\left[\left(V - V_{\text{solid}}\right)K^+\right]}{dt} = -\Lambda_K\left(k_B T \log \frac{K^+}{K_0^+} + \Delta\Phi\right) + 2S_p,$$ 
(A15)

$$\frac{d\left[\left(V - V_{\text{solid}}\right)Cl^-\right]}{dt} = -\Lambda_{Cl}\left(k_B T \log \frac{Cl^-}{Cl_0^-} - \Delta\Phi\right),$$ 
(A16)

where $\Delta\Phi = \Phi - \Phi_0$ is the electric potential energy difference between inside and outside of the cell, $S_p$ is the effective permeability of the membrane to ion X due to ion channels, and $S_p$ is the activity of the sodium-potassium pump – Na$^+$/K$^+$ ATPase, the prefactors—minus three and two—are due to the fact that the pump exchanges two potassium for three sodium. Cells pump sodium out of the cell, implying a positive value of $S_p$. Since the chloride ions are not actively transported there is no active pump contribution to the chloride flux. At steady state, **Equation A14** to **Equation A16** gives

$$Na^+ = Na_0^+ e^{-\beta(\Delta\Phi - 3S_p/\Lambda_{Na})}, \ K^+ = K_0^+ e^{-\beta(\Delta\Phi - 2S_p/\Lambda_K)}, \text{ and } Cl^- = Cl_0^- e^{\beta\Delta\Phi}$$ 
(A17)

Note that if $\Lambda_{Na}$, $\Lambda_K$ and $S_p$ depend on the ion concentration and the electric potential difference, then **Equation A17** is a set of implicit equations that need to be solved self-consistently.

In the following, we consider a single impermeant species (M=1 in **Equation A2**) of number in the cell Y, and effective charge z, which represents the proteins and the small impermeant charged molecules like phosphate ions. At steady state the osmotic balance reads

$$Na^+ + K^+ + Cl^- + \frac{y}{V - V_{\text{solid}}} = Na_0^+ + K_0^+ + Cl_0^-.$$ 
(A18)

The electroneutrality condition inside the cell and in the external medium is given by $Na^+ + K^+ = Cl^- \mp zy/\left(V - V_{\text{solid}}\right)$ and $Cl_0^- = Na_0^+ + K_0^+$, respectively. From the electroneutrality condition, **Equation A17**, and **Equation A18** we get the potential difference across the cell membrane, and the cell volume. The volume thus obtained reads

$$V = V_{\text{solid}} + \frac{(z+1)y}{2Cl_0^-\left(1 - e^{\beta\Delta\Phi}\right)}.$$ 
(A19)

For the volume to be finite we need $\Delta\Phi < 0$, consistent with different experimental measurements. Substituting **Equation A19** in the electroneutrality condition inside the cell, we obtain a quadratic equation for $e^{\beta\Delta\Phi}$. However, only one of the two roots leads to $\Delta\Phi < 0$. The electric potential difference thus obtained reads

$$e^{\beta\Delta\Phi} = \frac{z - \sqrt{z^2 - (z^2 - 1)\left((1-\delta)e^{-3\beta S_p/\Lambda_{Na}} + \delta\ e^{2\beta S_p/\Lambda_K}\right)}}{(z-1)},$$ 
(A20)

where $\delta = K_0^+/Cl_0^-$. We can now evaluate the change in concentration of chloride ions when tension is changed. Identifying $Xi^-$ with $Cl^-$ the l.h.s of **Equation A12** reads

$$\frac{1}{\left(Cl_0^- - Cl_{iso}^-\right)} \frac{dCl^-}{dt} = \frac{Cl_0^-}{\left(Cl_0^- - Cl_{iso}^-\right)} \frac{de^{\beta\Delta\Phi}}{d\gamma_m} \frac{d\gamma_m}{dt}, \tag{A21}$$

where we have used $P_{iso} = 2Cl_0^-$. The electric potential difference depends on tension due to mechanosensitivity of the ion channels and pumps. Varying *Equation A20* we get

$$de^{\beta\Delta\Phi} = \frac{(z+1)Na_{iso+}}{2Cl_0^-\left(ze^{-\beta\Delta\Phi} - (z-1)\right)} \frac{3\beta S_p}{\Lambda_{Na}} \left(\left(\frac{d\Lambda_{Na}}{\Lambda_{Na}} - \frac{dS_p}{S_p}\right) - \frac{K_{iso+}}{Na_{iso+}} \frac{2\Lambda_{Na}}{3\Lambda_K}\left(\frac{d\Lambda_K}{\Lambda_K} - \frac{dS_p}{S_p}\right)\right). \tag{A22}$$

For a small change in tension the channels and pumps change by a small value given by the relation

$$\frac{d\Lambda_{Na}}{\Lambda_{Na}} = \alpha_{Na} \frac{d\gamma_m}{\gamma_{iso}}, \frac{d\Lambda_K}{\Lambda_K} = \alpha_K \frac{d\gamma_m}{\gamma_{iso}}, \text{and } \frac{dS_p}{S_p} = -\alpha_S \frac{d\gamma_m}{\gamma_{iso}}. \tag{A23}$$

We expect an increase in the channel values and decrease in the pump values due to an increase in tension, implying that the proportionality factors should be positive. Substituting this we get

$$\alpha = \frac{R\ (z+1)Na_{iso+}\ (\alpha_{Na}+\alpha_S)}{2\left(Cl_0^- - Cl_{iso^-}\right)\left(ze^{-\beta\Delta\Phi} - z+1\right)} \frac{3\beta S_p}{\Lambda_{Na}} \left(1 - \frac{2\Lambda_{Na}}{3\Lambda_K} \frac{K_{iso+}}{Na_{iso+}} \frac{\alpha_K+\alpha_S}{\alpha_{Na}+\alpha_S}\right). \tag{A24}$$

Substituting the parameter values from *Appendix 1—table 5* into *Equation A24* we get

$$\alpha \sim -0.0084\left(1.7\alpha_K + 0.7\alpha_S - \alpha_{Na}\right). \tag{A25}$$

Thus we see that we expect to get $\alpha < 0$ if the values of $\alpha_K, \alpha_{Na}$, and $\alpha_S$ are comparable. For slightly different parameters values, it is indeed possible that $\alpha > 0$. Fitting the experimentally measured cell volume and the spreading data, we get $\xi = A_0 k\alpha/\gamma \sim 1$, taking $\alpha \sim 10^{-3}$, this gives $A_0 k/\gamma \sim 10$. In other words, a 10% increase of contact area leads to a doubling of tension in the elastic regime.

## Membrane tension as a function of rate change of contact area

We model membrane as a Maxwell viscoelastic element (elastic at short time and viscous at long time), taking the change in membrane tension $\gamma_m$ to be proportional to the contact area $A_c$, that is,

$$\left(1 + \tau\frac{d}{dt}\right)\left(\frac{\delta\gamma_m}{\gamma_{iso}}\right) = \frac{k\tau}{\gamma_{iso}} \frac{dA_c}{dt}, \tag{A26}$$

where $k$ is the elastic modulus and $\tau$ is the tension relaxation timescale that depends on various factors related to cortex organization as well on the membrane turnover. Note that $k$ is not the elastic response of the lipid bilayer, it is an effective parameter that is related to the cells ability to access its membrane reservoirs upon stretching. Substituting *Equation A26* in *Equation A13*, we get

$$\frac{1}{V_{iso}}\tau\frac{d\delta V}{dt} + \frac{1}{V_{iso}}\delta V = -\xi\tau\frac{1}{A_0}\frac{dA_c}{dt}, \tag{A27}$$

where $\xi = -A_0 k\alpha/\gamma_0$. For a time much less than the volume relaxation timescale $\tau$ the rate of volume change is proportional to the rate of spreading

$$\frac{1}{V_{iso}}\frac{d\delta V}{dt} = -\frac{\xi}{A_0}\frac{dA_c}{dt}. \tag{A28}$$

As observed in the experiments, faster the spreading rate larger the initial rate of volume loss. We take the time-series of the contact area as input in *Equation A27*. We first fit the contact area times series to the following equation $A_c(t) = A_0\left(1 - \exp\left(-t/\tau_a\right)\right)$, and thus obtain the best fit values of $A_0$ and $\tau_a$. Solving *Equation A27* with this expression for $A_c(t)$, we get

$$\frac{V}{V_{iso}} = 1 + r_{growth}t + \xi\tau\frac{e^{-t/\tau_a} - e^{-t/\tau}}{\tau - \tau_a}. \tag{A29}$$

We then obtain the best fit value of $\tau$ and $\xi$ by numerically fitting the volume dynamics to the measured volume time series using the 'NonlinearModelFit' function in Mathematica.

## Fast and slow-spreading control cells

As seen from the estimates the drug treatment affects multiple parameters. To confirm the relation between the spreading speed and volume loss we sorted the control cells (n=127) into three equal groups based on their average spreading speed in the first 10 min. We find that, indeed, the fast-spreading cells lose more volume than the slow-spreading cell. We fit the three groups—slow, intermediate, and fast-spreading cells—to the model and obtain the best-fit parameters. We find that the parameters characterizing the mechano-osmotic feedback ($\xi$ and $\tau$) are similar for the three classes. Compared to the cells spreading at intermediate speed, the rate of volume loss of the slowest spreading cell was 50% less than that of the fast-spreading cell was 50% more. The average of all the control cells was close to the group of cells with intermediate spreading speed and the fast-spreading cells behaved similarly to the GdCl$_3$ treated cells. The value of the parameters is listed in *Appendix 1—table 1*.

**Appendix 1—table 1.** List of fitted parameter values for the control cells are grouped based on spreading speed when the tension depends on change in contact area.

| Condition/parameters | $A_0$ (µm$^2$) | $\tau a$ (min) | $\tau$ (min) | $\xi$ |
|---|---|---|---|---|
| Slow spreading | 620 | 63.3 | 98.5 | 0.21 |
| Moderate spreading | 488 | 25.6 | 64.2 | 0.16 |
| Fast spreading | 656 | 17.2 | 99.4 | 0.16 |

## For different drug treatments

The parameter $\tau$ varies over a wide range of values, whereas $\xi$ is about one tenth for all cases except Y-27 and about half for the Y-27 treated cells. For fast-spreading cells treated with Y-27 and GdCl$_3$, the spreading rate $A_0/\tau_a$ is about 1.8 times that of control. However, the initial rate of volume loss from the model is about four times for Y-27 treated cells and about 1.5 times for the GdCl$_3$ treated cells. The parameter $\xi/A_0$ increases by a factor of 1.5 for the Y27 treated cells and decreases by a factor of 0.75 for the GdCl$_3$ treated cells. For EIPA treated cells, the spreading rate is the same as control but the parameter $\xi$ decreases by 20% as seen by a smaller volume loss. The initial rate of volume loss and rate of spreading is similar for the Lat A and CK-666 treated cells.

The volume recovery timescale is quite variable for the different drug treatments. Over the measured timescale of an hour, the volume recovery of the GdCl$_3$ and CK-666 treated cells is mainly due to growth. The value of the parameters is listed in *Appendix 1—table 2*.

**Appendix 1—table 2.** List of fitted parameter values for different drug treatment when the tension depends on change in contact area.

| Condition/parameters | $A_0$ (µm$^2$) | $\tau a$ (min) | $\tau$ (min) | $\xi$ |
|---|---|---|---|---|
| control | 479 | 25.9 | 53.5 | 0.16 |
| Y-27 | 833 | 24.4 | 16.4 | 0.67 |
| EIPA | 549 | 33.7 | 30.7 | 0.18 |
| Lat A | 1522 | 212 | 13.8 | 0.88 |
| GdCl$_3$ | 628 | 18.3 | 171.2 | 0.17 |
| CK-666 | 285 | 14.9 | $\infty$ | 0.09 |
| CK-666+Y-27 | 738 | 25.6 | 36 | 0.1 |
| EIPA+Y-27 | 1035 | 29 | $\infty$ | 0.02 |

## Membrane tension as a function of total area

The membrane tension dynamics is given by following Maxwell viscoelastic model:

$$\left(1 + \tau \frac{d}{dt}\right) \frac{\delta \gamma_m}{\gamma_{\text{iso}}} = \frac{k}{\gamma_{\text{iso}}} \frac{\tau}{A_t} \frac{dA_t}{dt},$$

(A30)

where $\gamma_{\text{iso}}$ is the homeostatic value of membrane tension, $\tau$ is the relaxation timescale, $k$ is the membrane elasticity, and $A_t$ is the total cell area. Substituting *Equation A13* in *Equation A30* the equation for volume dynamics reads

$$\left(1 + \tau\frac{d}{dt}\right)\frac{\delta V}{V_{\text{iso}}} = -\frac{\xi\tau}{A_t}\frac{dA_t}{dt}, \tag{A31}$$

where $\xi = -k\alpha/\gamma_{\text{iso}}$. To compute the total cell area we need to make assumptions about the shape of the cell. At early times, the shape can be well approximated by a spherical cap. However, at a later stage of spreading it is not a reasonable assumption. Since we do not know more about the three dimensional shape of the cell, for simplicity, we model the cell as a spherical cap through the duration of spreading. With this assumption, we can calculate the total area in terms of the instantaneous volume and the contact area. The volume and the total surface area of the spherical cap are given by

$$V = \tfrac{1}{6}h\left(3A_c + \pi h^2\right) \text{ and } A_t = \left(2A_c + \pi h^2\right), \tag{A32}$$

where $h$ is the height of the cell and $A_c$ is the contact area. Eliminating the height from *Equation A32*, we get the total area in terms of the volume and contact area, which reads

$$A_t = \left(2A_c + \pi\left(\frac{-A_c + \left(3\sqrt{\pi}V + \sqrt{A_c^3 + 9\pi V^2}\right)^{2/3}}{\sqrt{\pi}\left(3\sqrt{\pi}V + \sqrt{A_c^3 + 9\pi V^2}\right)^{1/3}}\right)^2\right). \tag{A33}$$

Taking the time derivative of *Equation A33* and using *Equation A32*, we get the following expression for rate change of the total area as function of rate change of the volume and the contact area:

$$\dot{A}_t = \frac{24\pi V\dot{V}}{(A_t + A_c)(A_t - A_c)} + \frac{2A_c\dot{A}_c}{A_t - A_c}. \tag{A34}$$

Substituting *Equation A33* into *Equation A31*, we get

$$\left(1 + \tau\frac{d}{dt}\right)\frac{V}{V_{\text{iso}}} = 1 + (\tau + t)g - \xi\tau\left(f_1\left(V, A_c\right)\frac{dV}{dt} + f_2\left(V, A_c\right)\frac{dA_c}{dt}\right), \tag{A35}$$

where $f_1\left(V, A_c\right) = 24\pi V/\left(A_t\left(A_t^2 - A_c^2\right)\right)$ and $f_2\left(V, A_c\right) = 2A_c/\left(A_t\left(A_t - A_c\right)\right)$. Rearranging the terms we get

$$\left(1 + \tau_{\text{eff}}\frac{d}{dt}\right)\frac{V}{V_{\text{iso}}} = 1 + (\tau + t)g - \xi\tau f_2\left(V, A_c\right)\frac{dA_c}{dt}, \tag{A36}$$

where $\tau_{\text{eff}} \equiv \tau\left(1 + \xi f_1\left(V, A_c\right)V_{\text{iso}}\right)$. Thus, we see that the volume relaxation timescale is normalized by the volume dependent term in total area, and this effective timescale is larger than the bare tension relaxation timescale.

The effective tension dynamics obtained by using *Equation A13* in *Equation A36* is

$$\left(1 + \tau_{\text{eff}}\frac{d}{dt}\right)\frac{\delta\gamma}{\gamma_{\text{iso}}} = \frac{k\tau}{\tau_{\text{eff}}\gamma_{\text{iso}}}\tau_{\text{eff}}\left(-r_{\text{growth}} + f_2\left(V, A_c\right)\frac{dA_c}{dt}\right), \tag{A37}$$

We now fit *Equation A36* with the experimentally measured volume to obtain the best-fit parameters. For this, we fit the cell spreading data with an exponentially saturating function of the form

$$A_c\left(t\right) = A_0\left(1 - \frac{A_0 - A_c(0)}{A_0}e^{-t/t_a}\right), \tag{A38}$$

where $A_c\left(0\right)$ is equal to the initial contact area which we obtain from the data. We solve *Equation A36* numerically, using the NDSolve function in Mathematica, for a set of parameter values $\xi$ and $\tau$. We then select the parameters which minimize the error between the numerically calculated volume and the experimentally measured volume using $L^2$ norm.

The best-fit parameter values for control cells grouped in three groups based on spreading speeds are listed in *Appendix 1—table 3* and that for cells treated with different drugs are listed in *Appendix 1—table 4*. We see that the value of $\tau$ is much smaller compared to the case in which we use contact area as a proxy for total area. This is due to the fact that the volume relaxation time is normalized by the dependence of the total area of the spherical cap on the volume. We also find that parameter $\xi$ is an order of magnitude larger compared to the fit using contact area. Although the parameters for the cells in three control groups (*Appendix 1—table 3*) vary significantly, this variation cannot explain the difference in volume loss. If we use the same spreading speed for the three sets of parameters, the resulting volume loss is actually maximum for the parameter corresponding to the slowest spreading cell. This implies that the observed difference in volume loss must be attributed to the difference in spreading speed itself (*Appendix 1—figure 1*).

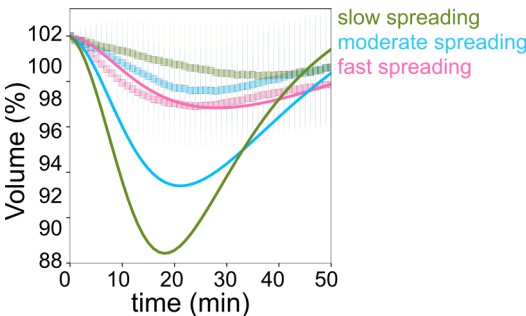

**Appendix 1—figure 1.** Model *Figure 1*. Fits for volume taking the contact area of the fast-spreading cells for the three sets of parameters. Model predictions for the cell volume, based on fits from data, for slow (green), moderate (cyan), and fast (pink) spreading.

We now compare the change in tension for two cases, one with finite volume tension coupling, that is, $\alpha \neq 0$, and compare it with the case $\alpha = 0$, which implies $\xi = 0$. For $\xi = 0$, the volume is given by $V = V_{iso} \left( 1 + r_{growth} t \right)$.

Substituting this volume in *Equation (A30)*, we can compute the difference between the two tensions. We find that during the initial spreading the change in tension when there is volume loss due to spreading is always lower than the case when there is not volume loss. In the later part of the spreading, this is not the case. However, since the spherical cap model is more reasonable at the start of the spreading rather than later. This supports the hypothesis that the functional role of the volume loss may be to prevent rapid increase in tension due to fast spreading.

## Fast confinement

From *Equation A1*, we see that, within the PLM framework, for a fixed external medium, the cell volume can change either due to change in hydrostatic or osmotic pressure. The Membrane rupture tension is $\sim 20$mN/m, for radius of curvature of about 5 this gives $\Delta P \sim 4 \times 10^3$Pa. The change in volume due to this pressure increase as given by *Equation A1* for constant ion concentration is

$$\frac{V_{iso}}{V} = \frac{\Delta P}{P_{iso}} + 1. \tag{A39}$$

For external osmolarity of 300mM, $_{iso} \sim 7 \times 10^5$Pa, which gives $\Delta P/P_{iso} \sim 0.5 \times 10^{-2}$. Thus, we see that even at the rupture tension the hydrostatic pressure can change the volume only by about 1%. Hence, the volume loss upon fast confinement that is of the order of 10% cannot be explained by just the increase of the hydrostatic pressure due to compression.

Within the framework of PLM, the volume change of the order of 10% can only be due to change in the osmolarity of the cell, which requires transport of ions. For ions transport to take place at timescales of milliseconds, the rates need to increase by 4 orders of magnitude. Such increase can be easily attained due to pore formation. However, formation of small pores that allow the ions to leak through but does not discriminate between the different ions will lead to an increase rather than a decrease in volume. This is due to the fact that the concentration of ions outside the cell is larger than that inside, hence once the pores open the ion flux is into the cell and the water flux follows the ions flux.

## Volume fluctuation during cell migration

The cell area fluctuates as the cell passes through the collagen matrix.

We take the total area to be $A_t = A_0 \sin(\omega t)$, where $\omega \sim v_{cell}/l_{mesh}$. As before assuming a viscoelastic model for tension driven by the area $A_t$, we can compute the volume fluctuation due to mechano-osmotic coupling with membrane tension. The volume dynamics is given by

$$\left(1 + \tau \frac{d}{dt}\right) \frac{V}{V_{iso}} = \frac{\xi\tau}{A_{iso}} \frac{dA_t}{dt} \tag{A40}$$

From *Equation A40*, the standard deviation of the volume is

$$\sqrt{\left\langle \frac{\tilde{V}^2}{V_{iso}} \right\rangle} = \frac{\xi\tau\omega}{\sqrt{1+\omega^2\tau^2}} \frac{A_0}{A_{iso}} \tag{A41}$$

If the distance cell moves in time $\tau$ is much smaller than the mesh size $v_{cell}\tau \ll l_{mesh}$ then the standard deviation of volume increases linearly with the cell velocity. As the speed increases, standard deviation of the volume saturates to the value $\xi A_0/A_{iso}$. We fit *Equation A41* to the experimentally measured values (*Figure 7K*). Taking the standard deviation of the volume at zero velocity to be 0.01 gives the best fit parameter values to be $\tau/l_{mesh} = 1.8 \text{min}/m$ and $\xi A_0/A_{iso} \sim 0.04$. For a mesh size of about 5 µm and area change of the order of few percents we get $\tau \sim 10\text{min}$, and $\xi \sim 1$, which is in the same range as that obtained when fitting the volume change upon spreading (see *Appendix 1—table 4*).

**Appendix 1—table 3.** List of fitted parameter values for the control cells are grouped based on spreading speed when the tension depends on change in total surface area.

| Condition/parameters | $A_0$ (µm²) | $\tau a$ (min) | $\tau$ (min) | $\xi$ |
|---|---|---|---|---|
| Slow spreading | 620 | 63.3 | 5 | 2.5 |
| Moderate spreading | 488 | 25.6 | 13 | 0.9 |
| Fast spreading | 656 | 17.2 | 51 | 0.3 |

**Appendix 1—table 4.** List of fitted parameter values for different drug treatment when the tension depends on change in total surface area.

| Condition/parameters | $A_0$ (µm²) | $\tau a$ (min) | $\tau$ (min) | $\xi$ |
|---|---|---|---|---|
| Control | 479 | 25.9 | 12 | 1 |
| Y-27 | 833 | 24.4 | 8.5 | 1.3 |
| EIPA | 549 | 33.7 | 8.5 | 0.8 |
| Lat A | 1522 | 212 | 2 | 3.4 |
| GdCl₃ | 628 | 18.3 | 34.5 | 0.4 |
| CK-666 | 285 | 14.9 | 29.5 | 0.9 |
| CK-666+Y-27 | 738 | 25.6 | 18 | 0.2 |
| EIPA+Y-27 | 1,035 | 29 | 4.5 | 0.1 |

**Appendix 1—table 5.** List of parameter values.

| Parameter | Definition | Estimate |
|---|---|---|
| | Concentration of chloride ions in the medium | 150 mM (*Kay, 2017*) |
| $K_0^+$ | Concentration of potassium ions in the medium | 3 mM (*Kay, 2017*) |
| $Na_0^+$ | Concentration of sodium ions in the medium | 147 mM (*Kay, 2017*) |

*Appendix 1—table 5 Continued on next page*

*Appendix 1—table 5 Continued*

| Parameter | Definition | Estimate |
|---|---|---|
| $Cl^-_{iso}$ | Concentration of chloride ions in the cell | 4 mM (*Equation (A17)*) |
| $K^+_{iso}$ | Concentration of potassium ions in the cell | 130 mM (*Equation (A17)*) |
| $Na^+_{iso}$ | Concentration of sodium ions in the cell | 5 mM (*Equation (A17)*) |
| $Y$ | Concentration of trapped particles in the cell | 30 mM (osmotic balance) |
| $zY$ | Concentration of trapped charges in the cell | 131 mM (electro-neutrality) |
| $\Delta\Phi$ | Potential difference across the plasma membrane | –90 mV (*Equation (A5)*) |
| $k_BT$ | Temperature (25°C) | $4.1*10^{-21}$ J |
| $\Lambda_{Na}$ | Permeability of $Na^+$ | $2*10^{-8}$ moles/(m²*s) (*Kay, 2017*) |
| $\Lambda_K$ | Permeability of $K^+$ | $6*10^{-7}$ moles/(m²*s) (*Kay, 2017*) |
| $\Lambda_{Cl}$ | Permeability of $Cl^-$ | $4*10^{-7}$ moles/(m²*s) (*Kay, 2017*) |
| $\beta S_p$ | NaK Atpase pumping rate | $1.4*10^{-7}$ moles/(m²*s) |
| $L_p$ | Hydraulic conductivity | $10^{-(12-13)}$ m/(Pa*s) (*Equation (A9)*) |
| R | Ratio of osmotically active volume to total volume | 0.7 |

