## [Editor Report]

The paper by Venkova et al. is a comprehensive study of mammalian cell volume dynamics during the common cellular process of adhesion and spreading on a flat substrate, osmotic changes, and mechanical confinement. The paper reveals a complex interplay between cell water/ion regulation, cytoskeletal processes, and mechanical deformation of the cell. The topic is important in cell physiology and should be of considerable interest to cell biologists, mechanobiologists and biophysicists.

---

## [Decision Letter]

**Decision letter after peer review:**

Thank you for submitting your article "A mechano-osmotic feedback couples cell volume to the rate of cell deformation" for consideration by *eLife*. Your article has been reviewed by 3 peer reviewers, and the evaluation has been overseen by a Reviewing Editor and Anna Akhmanova as the Senior Editor. The following individual involved in review of your submission has agreed to reveal their identity: Sean X. Sun (Reviewer #1).

Essential revisions:

- Discuss the accuracy/validity of the FXm method (Rev. 1, point 1).

- Is there an effect on acto-myosin contraction? Include the effect of hydraulic pressure in the model or further discuss why it can be neglected (Rev. 1, point 2 and recommendation 2,3). Related to this, consider also tje effect of myosin II mechanosensitive activation (Rev. 2 point 5).

- Similarly, the constant term in the model for the active pumping needs to be either reformulated or the further justified (Rev. 1, point 3).

- How is membrane tension at early times after seeding (e.g. 10min condition)? (Rev. 2, point 1).

- Somewhat related, it is not clear the role membrane tension plays in volume regulation, in particular with respect to the comparison between drug treatments and the theoretical model (Figure 6I,J) (Rev. 3 main comment). Would it be possible to monitor tension dynamics as suggested by Rev. 3?

- Add details and discuss further on the pulling experiments (Rev. 2, point 2 + Rev. 1 recommendation 4).

- Improve on the statistical analyses of the results (Rev. 2 point 9).

- I agree this is a very complex paper with lots of data. Probably the authors should consider re-writing some parts of the paper more clearly and expose better the different parts of the paper (Rev. 1, point 4 and recommendation 1). Although splitting the paper in two is of course a possibility, I think I would suggest that the distinction between the two scenarios (osmotic shock vs. spreading) is made more clear by re-sectioning and maybe adapting the model for each case, following the reviewer's suggestions (rev. 1 point 4 and also comment 3 of rev. 2).

- Discuss on the role (if any) of the initial cell volume in the observed effects (e.g. volume regulated by spreading speed) (Rev. 2, point 2).

- Direct measurements on calcium-mediated activation of myosin II (Rev. 2 point 6).

- Related to NHE1, is it also involved in volume regulation under osmotic stress? (Rev. 2 point 7).

- Discuss on alternative Arp2/3 functions that can lead to volume regulation (Rev. 2 point 8).

*Reviewer #1 (Recommendations for the authors):*

1) It has to be said that the paper is quite complex to read. There is almost too much data. 3 different scenarios, spreading, osmotic shock, and confinement, are considered. While I appreciate that these phenomena are inter-related, as it is it's difficult to read. Each scenario can be explored in more detail, with models specifically addressing the scenarios. I understand this would require extensive revision, so I leave it up to author's discretion.

Consider splitting the paper and concentrate on 1 or 2 phenomena only. Personally, I think the dynamics during cell spreading and osmotic shock are significantly different. Osmotic shock involves a very large change in ion concentration, therefore might be more appropriately described by the model. During spreading and confinement, there is a small volume change, and I think the correct approach is to consider active contraction and ion dynamics together.

2) It could be more appropriate to model spreading and confinement using a 1-solute phenomenological model, but include myosin contraction.

3) Some comments could be more rigorous. For example, the tether force is not just a function of membrane tension, but also active contraction. Comments about 1-solute model, etc.

4) What is the significance of measuring tether force 4hrs vs. 30min?

*Reviewer #2 (Recommendations for the authors):*

The authors find a correlation between dV/dt and dA/dt in spreading cells and argue that an increase in tension is associated with cell volume loss. It is however not fully clear how a fast increase in tension is build up during the initial phases of cell spreading to explain the highest change of dV/dt at short time scales.

Tension values were also measured via AFM at 30 min after spreading when dV/dt and dA/dt are zero. The authors should clarify how this time point is informative to argue about tension driving volume change. In this context, it remains to be tested how tension evolves at earlier time points during spreading. Given also the higher tension in cells for specific treatments at 30 min (e.g. Y-27632), why would this not lead to a longer period of cell volume decrease compared to control cells considering the authors' argument of tension homeostasis in cells? In this context, the authors could also extend their comments on the results of panel 6I in the main text and how far they match with the measured apparent membrane tension.

The presentation of AFM measurements and data would benefit from a more detailed description of the methodology. Tether pulling experiments were indicated at 30min/4h but given the time it takes to perform a single experiment and probe multiple cells the authors should comment on the specific time duration of experiments to facilitate the interpretation of results. In addition, parameters as tether pulling positions (cell body versus protrusions) and tether pulling statistics would enable to assess the presented data more rigorously. Can the authors further comment on the highly variable contact times used for AFM tether pulling experiments, is it associated with specific experimental conditions? Were drug treatments further performed 30min prior to measurements or was the incubation performed for 4h, to clarify long-term effects of drugs and potential cell adaptations.

Authors define categories of fast/intermediate and slow spreading cells. Can the authors clarify if initial cell volumes is relevant to consider and different in these categories? Does initial cell volume also change with drug treatments (e.g. Y-27632 treatment, a condition in which the cells spread faster)? The authors could further comment in the main text on potential parameters regulating spreading rate variability in cells.

The authors use confinement and osmotic shocks to test model assumptions and to study cell volume changes at short time scales. However, the authors so far do not consider applying the modified PLM to volume adaption in osmotic shock conditions or potential long-term volume changes in confinement. Can the mechano-sensitive PLM also capture volume adaption in osmotic shock conditions or what are the potential limitations? In addition, do cells in confinement show a volume adaptation over time that would follow similar principles as in spreading cells or osmotic shock experiments?

The authors argue for a role of myosin II in counteracting volume loss in spreading cells. Would this effect also be relevant in confinement as the authors previously showed a mechanosensitive activation of myosin II in confined cells? Does volume loss occur similarly when myosin II activity is inhibited? Do the authors expect a similar role of myosin II in regulating volume change in confinement and volume adaptation in osmotic shock conditions and spreading cells?

The authors explain the Gadolinium effect by referring to the fact that calcium increase might activate myosin II. Direct measurements such as MLCK treatment or different calcium interference approaches would allow to support this argument directly.

Does NHE1 play a similar role during adaptation to osmotic stress conditions as observed in spreading cells? It would further be interesting to show whether the function of NHE1 for mechano-sensitive volume regulation is observed in the other cell types tested in this study.

Arp2/3 is identified as a molecular regulator of cell volume in spreading cells and the authors speculate on its role in regulating ion channel/pump activity. The authors could further include a discussion on alternative functions of Arp2/3 such as regulating endo/exocytosis relevant for cell volume regulation.

The discussion could further benefit from adding further arguments on the physiological relevance of cell volume regulation, considering the observed range of volume change and the focus on the specific mechanical stress conditions studied in this work.

Statistical testing, the authors present N=1 in multiple panels of the presented data and should re-check the statistical significance of these results. Including p-values directly in the graphs would ease the reading of figure panels.

Propidium iodide staining could be used to monitor plasma membrane rupture in confinement, considering the high leakage/volume loss in confined cells. It also appears that the authors did not comment on the high increase in surface area in Figure 5H in the main text.

LatA treatments are often used at concentrations which only partially depolymerize the cell cortex but concentrations vary and could be more clearly commented in the main text. Would higher LatA concentrations increase cell spreading in confinement and alter the cell volume change?

Related to Figure 1G, the authors state "causing a transient density increase". Panel S1G however indicates an increased density with a stable plateau, can the authors clarify this statement?

The authors state that "with a loss of volume reaching up to 20% for fast-spreading cells" and "actin-dependent loss of volume (up to 30%)". The max volume loss of cells is not clearly visible in the current graphs and appears exaggerated in the text. The authors could include individual data points or change to a different data illustration to visualize the range of cell volume loss better.

The authors could list the specific cell types tested in the study in various statements in the main text that appear rather generic otherwise.

How does tau_a versus tau in the table relate to tau_eff and tau in the main text? Providing a clearer introduction and overview of model parameters would support the readability.

Comments on figure panels:

Figure 1A, cells are not well visible in this figure panel and a higher magnification or zoom in could be used.

Figure 1B, indicate channels in legend and check for consistency in figure panel.

Figure 1C, for the micropattern data indicate spreading time relevant for volume calculation. The authors also argue that they use micropatterns to reduce spreading area to a defined size compared to control conditions. Comparing Figure 1B and 1A, the cell areas are however similar and control cell spreading area appears to be higher in panel 1B versus 1A?

Figure 1I, J, indicate in the legend at which time point dV/dt was measured. How was the scale bar in panel 1I calculated?

Figure 2C, there appears to be a gap in the data of Y-27632?

Figure 3A, the explanation of the schematic of the PLM could be explained in more detail to ease the reading.

Figure 3D vs 4B, why is the Ponders equation in panel 3D independent of actin but not in panel 4B? LatA data in panel 4B appear to be not commented in main text.

Figure 4B, check sign of dV/dt compared to other figure panels as it appears neg/pos for hypo/hyper conditions when cells swell/shrink.

Figure 4C, over which duration was adaptation measured?

Figure 5E, the legend indicates that a double expression line of myosin II/actin was used but only one channel is shown (membrane?). The visualization of myosin II and actin would be informative to include.

Figure 6K, indicating measured tension values in Figure 6K or matching the color code between panel 6J and 6K would ease the reading of figures.

Figure S2G Why is the correlation before and after lamellopodia changing slope, appears to be not commented and the figure is not cited in the main text.

Figure S2F, what do the lines indicate?

*Reviewer #3 (Recommendations for the authors):*

It would be good to insert figure S4H in main figure, as it clearly how speed of deformation affects the volume change.

The absence of correlation between volume and spreading area at a given spreading rate is interesting and new. It is probably worth discuss the authors' results in light of Ref 11, that conclude on a weak correlation between spread area and volume, and ref 9 that observe a strong correlation between area/volume.

Fig1F

Could the authors consider including the spreading area change in time on PLL-PEG?

Fig1D-H

Color code is not consistent and make the figure harder to read.

Page 5, first paragraph

Please explain which criteria you used to pool the cells according to their spreading speed: do you have 3 clear populations, well separated by spreading speed, or you had to apply cutoffs sort cells in different populations?

Could the authors indicate the three corresponding populations ('slow spreading', 'moderate spreading', 'fast spreading') on Figure 1I? Consider clarifying the color code.

Can the authors plot Figure 1J for each population ('slow spreading', 'moderate spreading', 'fast spreading')? Is the same tendency still observed?

Page 5, second paragraph

In RPE-1, dynamics of spreading area (Figure S2D) and volume (Figure S2E right) for control cell and under Y-27 are very similar, while it is not the case in HeLa cells. Can the authors comment on it?

Figure 2B-C

Axes do not have the same minima/maxima so complicated to read and to compare with Figure 1H.

Figure 2D

Axis do not have the same minima/maxima. It would be easier to read and compare with Figure 1I if they had the same range. Are the linear fits in fig2D and fig1I giving the same results? They seemed to be different.

Page 5, third paragraph

The results of this part are very similar to the paper cited above (https://doi.org/10.1101/2021.01.22.427801), probably worth mentioning. In the main text, authors should specify that you used PEG400 to apply hyperosmotic shocks. Do the authors find differences if they apply osmotic shocks with salts or adding non-metabolized sugars such as sucrose or sorbitol?

Page 6, second paragraph

Under Y27 treatment, cell spread faster and their membrane tension measured using AFM is 30% higher than control 30 min after seeding. As LatA treated cells spread slower, wouldn't the authors expect a smaller membrane tension? Dai and Sheetz-1999 looked at the effect of cytochalasin D and indeed observed a reduced membrane tension.

Page 8

What are the differences, beside the timescale, between Figure 3C and Figure 4A? It would be worth mentioning the way these different were acquired in the figure legend.

Page 9 last paragraph + page 10 first paragraph

Is intensity of CellMask a proper tool to extrapolate on membrane folds and reservoirs (Figure 5E)? The spread around the contour line seems larger for 7.5um than for no confinement for control cells, and the reverse for Lat A treated cells. I am not sure how these intensity plot can be quantitatively linked to the size of the membrane reservoir.

Also, in the before to last paragraph in page 9, I kindly disagree with the statement that "membrane tension could not arise from a limitation in membrane availability". Isn't it that unfolding the membrane partially, not reaching the full size of the membrane reservoir, would still change membrane tension, because of opposing cytoskeletal and adhesive forces trying to fold the membrane?

Figure 5F

LatA treated cell keep a constant volume after confinement while control cell decrease volume. Since you have a condition (Y27 treated cell) that have opposing effect of LatA on spreading area and volume, I wonder what would be the effect of confinement on Y27 treated cells. Would cells 'overreact?'

Figure 6 A-H

Y-axis scale is different for each graph, would be better to fit one range and keep it so that the reader can compare from one to another.

Figure 6J

Do you affect absolute cell volume when adding those drugs?

[Editors' note: further revisions were suggested prior to acceptance, as described below.]

Thank you for submitting your article "A mechano-osmotic feedback couples cell volume to the rate of cell deformation" for consideration by *eLife*. Your article has been reviewed by 3 peer reviewers, and the evaluation has been overseen by a Reviewing Editor and Anna Akhmanova as the Senior Editor. The following individual involved in the review of your submission has agreed to reveal their identity: Sean X. Sun (Reviewer #1).

Essential revisions:

We all agreed that the revised version of the manuscript is much improved and clearly presents a very interesting and compelling story, well done!

Before the formal acceptance of the manuscript, one of the reviewers (rev. #1) made two final comments regarding (1) the contractile stress's role in volume control; and (2) the relative importance of membrane tension and voltage for the effects presented here. I think it might be a good idea to consider these comments and decide whether you think you can adapt the discussion part of the text so your points are clearly laid down.

*Reviewer #1 (Recommendations for the authors):*

The authors have addressed my comments and the manuscript is much improved. I can recommend publication. I think there is generally no disagreement in our thinking of contractility and cell volume regulation. It is true that contractile force, or pressure, alone cannot really change the volume very much. This was already shown in Yellin et al., Biophys. J. 2018, and Li and Sun, Frontiers Cell Dev. Bio., 2022, and I agree with the authors' general thinking. However, the discussion in Pg. 7, I think, can be improved. Right now, it sounds like contractile stress is being compared with the total osmotic pressure of impermeable solutes inside the cell. But in reality, the contractile stress should be compared with the total osmotic pressure difference between inside and outside of the cell, which is much smaller than 100mM. During 10% volume change during cell spreading, I think since it's happening slowly (order of minutes), the ion flux and contractile stress should change simultaneously, possibly with contractile stress changing slowly, while ion flux happening very fast. But, the contractile stress must change, since at minimum, the cell is changing shape, and therefore cortical stress must change to maintain relative force balance. I'm still not completely sure whether is ok to neglect it, since the steady-state result of the proposed model is zero osmotic pressure difference between inside and outside, but in reality, there is about 1kPa of hydraulic pressure difference and 1kPa osmotic pressure (~0.5mM) difference. This difference may be small enough to be negligible in the current context. But it may not be small in other contexts, such as during cell motility.

The other issue I hope the authors can clarify in the discussion is the equal importance of membrane voltage and membrane tension in this story. Ion exchangers such as NHE and NaK are affected by both. The current dependence of exchanger flux with respect to voltage is based on just passing an ion through a constant electric field. This is generally not true for ion exchangers. I think most studies of ion exchangers show a voltage gating effect. This, after all, is the basis of action potentials. Therefore, these important facts should be clearly stated.

*Reviewer #2 (Recommendations for the authors):*

The authors have made an effort to provide a detailed response to the reviewer's comments and the manuscript is of high quality to be published in Elife. The work presents a relevant step forward to understand the coupling of cell mechanics during cell spreading and volume regulation and opens up interesting questions to be addressed in future studies.

*Reviewer #3 (Recommendations for the authors):*

The authors have completely addressed all my points, and I enthusiastically recommend publication.

---

## [Author Response]

Essential revisions:- Discuss the accuracy/validity of the FXm method (Rev. 1, point 1).

This has been done. The result is that, from one experiment to another, the absolute measure of the average volume of a population of cells varies by +/- 10% as mentioned by the referee. But in this article, most of the measures are temporal measures of the same single cell. In this case, the accuracy of the measure is +/- 1%, as we now show experimentally. See more details in the answer to Referee 1, point 1 below.

- Is there an effect on acto-myosin contraction? Include the effect of hydraulic pressure in the model or further discuss why it can be neglected (Rev. 1, point 2 and recommendation 2,3). Related to this, consider also tje effect of myosin II mechanosensitive activation (Rev. 2 point 5).

This has been done. The experimental results show that acto-myosin contraction has no direct effect on cell volume. This is also what the classical theory predicts (and thus others before us, see ^1^). The reason comes from the fact that while ions represent the most abundant osmolytes in the cell (about 200 mM), the amount of nonionic osmolytes (about 100 mM) cannot be neglected. The force needed to ‘concentrate’ these nonionic osmolytes even by a few percent is much larger than what the acto-myosin cortex can produce. The only way to change the volume of the cell by a few % or more is by changing the ion content, as predicted by the classical pump and leak model (or change the nonionic content, but that is even much slower and is rather on the timescale of cell growth). This means that the effect of myosin contractility on the volume of spreading cells is only indirect, via the modulation of the spreading speed. See also the more detailed answers to the referees below.

- Similarly, the constant term in the model for the active pumping needs to be either reformulated or the further justified (Rev. 1, point 3).

This has been addressed. See the detailed discussion on this modeling aspect in the answer to the referee.

- How is membrane tension at early times after seeding (e.g. 10min condition)? (Rev. 2, point 1).

This point has been addressed. It is not possible to measure membrane tension at early times for two reasons: the measure takes time, and the cells need to be adhered enough (they should not move). We thus instead provide a measure of the membrane tension for cells which are adhered but not spread, with a spreading area corresponding to the early times after seeding, using small adhesive micropatterns. See more details in the answer to the referee.

- Somewhat related, it is not clear the role membrane tension plays in volume regulation, in particular with respect to the comparison between drug treatments and the theoretical model (Figure 6I,J) (Rev. 3 main comment). Would it be possible to monitor tension dynamics as suggested by Rev. 3?

This point was mostly due to a confusing labelling of the axis of Figure 6I, which made it hard to understand what the graph shows, despite details given in the figure legend. We changed the labelling. What this graph shows is the difference in tension, as predicted by the model, comparing the case with or without the mecano-osmotic coupling effect, for each of the condition, using the fits from the experimental data. The largest difference is for the Y-27 treatment, because in this case the cells are spreading fastest and not having the mechano-osmotic feedback would lead to a large increase in tension. On the other hand, for conditions in which the mechano-osmotic feedback is already altered (EIPA treatment for example), the difference of removing it in the model is small. Having that in mind, it is then clear that the predictions of the model from Figure 6I actually correspond perfectly to the experimental observations for the membrane tension – affecting the mechano-osmotic coupling in Y-27 treated cells (for example the Y-27 plus EIPA or the 7-27 plus CK 666 treatments) leads to a further increase in the membrane tension assessed from tether pulling, mostly during the fast spreading phase. We understand that it is not a very easy point to understand, but we hope that it is now more clear. We also modified the text describing these results (p. 12).

As for the suggestion of Referee 3 to use Flipper probes, we have tried and provide the results that we obtained, but we failed to get good enough data to lead to significant results. We also discussed with the lab which developed these probes (Aurélien Roux) and Aurélien confirmed that such experiments are hard to do (live cell imaging with the probe) and would require us to perform a series of experiments in his lab, which was not possible to organize given the current situation and the short time for the revisions. Moreover, comparing strictly the results from tether pulling and the Flipper probe is a project per se. These experiments will be performed in the future, because they are indeed very interesting, but it is out of the scope of this article.

- Add details and discuss further on the pulling experiments (Rev. 2, point 2 + Rev. 1 recommendation 4).

This point has been addressed. See the details in the answer to the referees below.

- Improve on the statistical analyses of the results (Rev. 2 point 9).

This point has been addressed.

- I agree this is a very complex paper with lots of data. Probably the authors should consider re-writing some parts of the paper more clearly and expose better the different parts of the paper (Rev. 1, point 4 and recommendation 1). Although splitting the paper in two is of course a possibility, I think I would suggest that the distinction between the two scenarios (osmotic shock vs. spreading) is made more clear by re-sectioning and maybe adapting the model for each case, following the reviewer's suggestions (rev. 1 point 4 and also comment 3 of rev. 2).

This has been addressed be rewriting extensively the introduction and part of the discussion of the article. We feel that the main issue was that we did not make it clear enough that our article is solely centered on understanding the volume modulation during cell spreading and that the central result is that volume modulation depends on spreading speed. The other aspects, osmotic shocks and confinement, were introduced to test some aspects of the model, and to characterize the (almost) purely osmotic and (almost) purely mechanical volume responses of the cell, while spreading corresponds to a regime where these two aspects overlap due to the modulation of ion fluxes by cell membrane tension.

- Discuss on the role (if any) of the initial cell volume in the observed effects (e.g. volume regulated by spreading speed) (Rev. 2, point 2).

This has been addressed by showing that there is no effect of the initial volume of the cells, and that the various drug treatments do not affect the initial volume distributions. See more details in the answer to the referee (in fact Referee 2 point 3, as well as Referee 3 point 21).

- Direct measurements on calcium-mediated activation of myosin II (Rev. 2 point 6).

This point has not been addressed, because we think it is out of the scope of the study. We have modified the text to avoid speculation on that point. We also performed a few experiments but decided not to include them because they were not conclusive. See more details in the answer to the referee.

- Related to NHE1, is it also involved in volume regulation under osmotic stress? (Rev. 2 point 7).

This point has been addressed. The answer is that NHE1 inhibition has in general an effect on the speed of cell volume increase, so it affects the adaptation to osmotic shocks, but, as expected, it has not effect on the immediate response to the shock (the Ponder’s relation).

- Discuss on alternative Arp2/3 functions that can lead to volume regulation (Rev. 2 point 8).

This has been implemented.

Reviewer #1 (Recommendations for the authors):1) It has to be said that the paper is quite complex to read. There is almost too much data. 3 different scenarios, spreading, osmotic shock, and confinement, are considered. While I appreciate that these phenomena are inter-related, as it is it's difficult to read. Each scenario can be explored in more detail, with models specifically addressing the scenarios. I understand this would require extensive revision, so I leave it up to author's discretion.Consider splitting the paper and concentrate on 1 or 2 phenomena only. Personally, I think the dynamics during cell spreading and osmotic shock are significantly different. Osmotic shock involves a very large change in ion concentration, therefore might be more appropriately described by the model. During spreading and confinement, there is a small volume change, and I think the correct approach is to consider active contraction and ion dynamics together.

We apologize for the confusion introduced by the way we wrote the article. In fact, we completely agree with the referee, and we do not think that the dynamics of volume during spreading and osmotic shock are the same at all. The goal of our article is really centered on understanding volume loss during cell spreading, and our central result is that it depends on the rate of spreading, which was not reported before and is the basis for the model we propose. We added experiments on osmotic shocks and on confinement to better characterize our cells in the context of a (almost) purely osmotic response (osmotic shocks) and an (almost) purely mechanical response (fast confinement), because we believe that in the context of cell spreading, the timescale is such that both osmotic and mechanical responses overlap and are coupled, leading to our observation of a deformation rate dependent volume loss.

We have tried to clarify this point by rewriting the end of the introduction and also largely rewriting the first part of the article and moving some elements to supplementary figures or to discussion. We hope that the focus of the paper is now more clear.

2) It could be more appropriate to model spreading and confinement using a 1-solute phenomenological model, but include myosin contraction.

We respectfully disagree with the referee on that point. Because the referee revealed his name, we know that we have a disagreement on this precise point, and we have tried in our answer to his main point 2 above to explain our view. We hope that we convinced him that our view of the question is correct. In brief, on the theoretical side, we do not believe that a one solute model is an appropriate model to use in our case, because the volume changes of several % to more than 10% cannot be accounted by changes in contraction (because of the 100 mM of nonionic osmolytes present inside the cell, which cannot be ‘concentrated’ by the force produced by the acto-myosin cortex). Experimentally, we now added results showing that modulating contractility has no direct effect on cell volume, which supports our interpretation that the effect we observe of decreasing contractility during cell spreading is indirectly due to a modulation of the cell spreading speed. We agree that it could also have other indirect effects, which we now mentioned. But in any case, what we observed is that reducing contraction leads to larger volume loss during cell spreading, which goes in the opposite way of a direct effect of contraction on volume. If forces produced by contraction were large enough to have an effect on volume, myosin inhibition should lead to an increase in cell volume and not a reduction as we observed.

3) Some comments could be more rigorous. For example, the tether force is not just a function of membrane tension, but also active contraction. Comments about 1-solute model, etc.

We apologize for these approximations, and we have modified the text accordingly (e.g. top of p 8 for the tether force).

4) What is the significance of measuring tether force 4hrs vs. 30min?

This is an important point, and we apologize if it was not clearly explained in the text. These timings are the time at which the tether force is measured after cells have been plated to spread. 30 min after plating correspond to the spreading phase and the phase of volume loss, while 4 hours after platting, cells have finished spreading for several hours and have returned to steady-state growth. Accordingly, inducing faster spreading with Y27632 leads to an increase in tether force only during the spreading phase (30 min) and not at steady-state (4h). In general, for all treatments, when there is an effect leading to a higher tether force, it is observed mostly at 30 min and goes down at 4h, meaning that it is a transient increase in tether force related to the spreading phase.

We have explained that more clearly in the text when we present these results (p 8).

Reviewer #2 (Recommendations for the authors):The authors find a correlation between dV/dt and dA/dt in spreading cells and argue that an increase in tension is associated with cell volume loss. It is however not fully clear how a fast increase in tension is build up during the initial phases of cell spreading to explain the highest change of dV/dt at short time scales.

We thank the reviewer for this point. We emphasize that one should make the distinction between the actual value of the tension and the rate at which the tension changes. In our model, the rate of volume change is directly proportional to the rate of tension change (Equation1). Although the tension needs some time to increase above its resting value (it needs the cell to spread), the rate at which the tension changes is (at short time, in the elastic regime) proportional to the rate at which the cell area increases upon spreading, which is typically fastest at the onset of spreading. This naturally leads to the conclusion that the rate of volume loss will be highest at the onset of spreading (as observed). Importantly, in our analysis, we took time zero for spreading based on the first time point at which we are able to detect spreading area with 20X magnification, and not the plating time. Once the cells start to spread, they do so at various speeds, and the spreading speed can vary also in time for each single cell, getting faster, or slower (see individual cell curves Figure S1I), but, on average, the spreading speed is faster in the first 10 to 20 minutes after the beginning of spreading, then it slows down (Figure 1G). Our analysis shows that faster spreading cells lose more volume, and that this happens mostly during the first 10 to 20 minutes after the start of spreading (Figure 1J).

Tension values were also measured via AFM at 30 min after spreading when dV/dt and dA/dt are zero. The authors should clarify how this time point is informative to argue about tension driving volume change. In this context, it remains to be tested how tension evolves at earlier time points during spreading. Given also the higher tension in cells for specific treatments at 30 min (e.g. Y-27632), why would this not lead to a longer period of cell volume decrease compared to control cells considering the authors' argument of tension homeostasis in cells? In this context, the authors could also extend their comments on the results of panel 6I in the main text and how far they match with the measured apparent membrane tension.

We agree with the reviewer that these points need to be clarified. Unfortunately, due to technical limitations, it is not possible to combine tether forces measurements and cell volume measure by FXm (one needs a cantilever, the other is performed in a closed chamber). The two measures were thus performed separately. In addition, because of the time it takes to measure tether force and the difficulty to do it on very early spreading cells, it was not possible to follow the tether force from the time zero of spreading on single cells. We thus showed two measures at two time points, one as early as possible, which corresponds to the 30-90 min time after cell plating and the second 4-5 h after plating. It is important to mention that time zero is not exactly the same for the curves showing volume measures and for the tether force. For the volume and spreading curves, time zero was taken when a contact area between the cell and the coverslip became visible by RICM, which could also vary from cell to cell, and occurred after flow equilibration in the FXm chamber. On the other hand, the time 0 for the tether force was taken as the time when cells were seeded on the dish. Also, for volume, the time 0 is set for each single cell, while for the tether force, it is a global time 0 for the dish and the measures are done across a window of time which in overall corresponds to the phase of spreading area increase, and we call it now “spreading phase” This is the best we could achieve technically. Overall, we estimate that there is about 10 minutes of delay on average between the time 0 of these two measures. We have now made it more clear in the text (p 8) and in the Methods (Supplementary file, p. 19-20).

The comment of the referee also made us realize that we do not provide a value for the initial tether force prior to spreading. Because, as explained above we cannot obtain a very early measure (when cells are not adhered at all, it is not possible to perform tether force measure), we decided to measure the tether force for adhered non-spread cells. To achieve rounded spherical cells, we used small round micropatterns of 20 µm in diameter. We found that the tether force values, for both control and Y-27632 treated cells, are the same for adhered non-spread cells on small micropatterns and for spread cells at steady-state (4h after seeding).

This control has now been added to the Figure 3E and commented in the main text (p. 8).

From the modeling point of view, our theory is based on the assumption that the membrane behaves as a viscoelastic material sensitive to the rate of change of total cell area. During the initial phase of cell spreading (timescale below the relaxation timescale \tau) the membrane is elastic, and the increase in tension is proportional to the increase in the cell area. This explains why the rate of tension increase is fastest during the initial stage of spreading, when the spreading rate is fastest. At longer timescales, the tension starts to decrease due to the viscous relaxation. Since the volume change is proportional to the negative of tension change, volume follows the same trend.

At longer timescales, the tension relaxes to its homeostatic value. At some intermediate time, which depends both on the spreading rate and the viscoelastic relaxation time, but not on the initial tension value, the tension will reach its maximum values, and the volume will be at its minimum. The time of this minimum volume varies between ten to twenty minutes for most of the experiments.

We apologize for the mislabeling of the former Figure 6I (now Figure 5I). This plot shows the theoretical estimate for the difference in tension (in the units of homeostatic tension) between the case when the cell loses its volume upon spreading (as observed in experiments) compared to the hypothetical situation when the cell does not lose volume upon spreading (α = 0). The positive value of the tension difference predicts that the cell tension would have been higher if the cell were not losing volume upon spreading, which suggests a possible physiological role for the mechano-osmotic feedback we uncovered. It also matches our experimental observations for drug treatments which reduce or abolish the volume loss during spreading and correspond to higher tether force only at short time.

The presentation of AFM measurements and data would benefit from a more detailed description of the methodology. Tether pulling experiments were indicated at 30min/4h but given the time it takes to perform a single experiment and probe multiple cells the authors should comment on the specific time duration of experiments to facilitate the interpretation of results. In addition, parameters as tether pulling positions (cell body versus protrusions) and tether pulling statistics would enable to assess the presented data more rigorously. Can the authors further comment on the highly variable contact times used for AFM tether pulling experiments, is it associated with specific experimental conditions? Were drug treatments further performed 30min prior to measurements or was the incubation performed for 4h, to clarify long-term effects of drugs and potential cell adaptations.

We apologize for this and, as explained in the point above, indeed the tether pulling experiments take some time, which is why we cannot have a better time resolution for these. We have now added a more detailed description, including the time 0 issue, as described in the answer to the point 1 of the referee (p. 8). We hope it clarifies the aspects raised by the referee. This is the updated section in Materials and methods:

“Tether pulling

For apparent membrane tension measurements, tether force was measured with single cell atomic force spectroscopy by extruding tethers from the plasma membrane on top of the nucleus of HeLa EMBL cells. Cellview glass bottom dishes (Greiner) were coated for 1 h with fibronectin (50 μg/ml; Σ-Aldrich). Cells were incubated for 30 min with in the presence of drugs or vehicle, then plated, and probed either during spreading (from 30 to 90 min after plating) or at steady state (fully spread; from 4 to 5 h after plating). To perform experiments on non-spread cells, Fibronectin-coated circles (Ø 20 um) were micropatterned onto Cellview glass bottom dishes (Greiner) using PRIMO (Alveole) following the manufacturer’s recommendations.

Tether extrusion was performed on a CellHesion 200 BioAFM (Bruker) integrated into an Eclipse Ti inverted light microscope (Nikon). OBL-10 Cantilevers (spring constant ~60 pN/nm; Bruker) were mounted on the spectrometer, calibrated using the thermal noise method (reviewed in^19^) and coated for 1 h at 37° C with 2.5 mg/ml Concanavalin A (Σ-Aldrich), which binds polysaccharides expressed on the surface of the cell^20^. Before the measurements, cantilevers were rinsed in PBS and cells were washed and probed in Dulbecco’s Modified Eagle Medium with Glutamax (DMEM/Glutamax; Gibco) supplemented with 2% FBS (Life Technologies) and 1% penicillin-streptomycin solution (Life Technologies). Measurements were run at 37° C with 5% CO_2_ and samples were used no longer than 1 h for data acquisition.

Tether force was measured at 0 velocity at which is linearly proportional to apparent membrane tension, assuming constant membrane bending rigidity^21^. In brief, approach velocity was set to 0.5 µm/s while contact force and contact time ranged between 100 to 200 pN and 100 ms to 10 s respectively. The latter two parameters were experimentally tuned before every tether pulling attempt, aiming to reach a tradeoff between the maximization of the probability to extrude single tethers, and the reduction of experimental stress on the cells. The larger the contact time and force, the higher is the probability of formation of bonds between the molecules of Concanavalin A on the surface of the cantilever and the polysaccharides on the surface of the cell. On the other side, the lower those two parameters, the lower is the stress experienced by the cell during the contact with the cantilever. As a general trend, contact force and time must be increased over the course of the experiment, owing to the depletion of Concanvalin A from the cantilever^22^.

To ensure tether force measurement at 0 velocity, after contacting the cell surface the cantilever was retracted for 10 µm at a velocity of 10 µm/s. The position was then kept constant for 30 s and tether force was recorded at the moment of tether breakage at a sampling rate of 2000 Hz. Each tether extrusion attempt last about 3 min and each cell was probed until 3 single tethers were successfully extruded or for a maximum of 10 min.

Force-time curves resulting from successful tether extrusions were analyzed using the JPK Data Processing Software. Tether force values from tethers extracted from the cell were then averaged, and each cell was accounted as a single data point. For every experimental condition the Shapiro–Wilk test was used to test for normality of data. Student t-test was chosen for statistical testing of normal distributed data, while Mann–Whitney U-test was performed on non-normal distributed data.”

Authors define categories of fast/intermediate and slow spreading cells. Can the authors clarify if initial cell volumes is relevant to consider and different in these categories? Does initial cell volume also change with drug treatments (e.g. Y-27632 treatment, a condition in which the cells spread faster)? The authors could further comment in the main text on potential parameters regulating spreading rate variability in cells.

We thank the referee for pointing to this initial volume parameter. In Author response image 1 we show that, for most drug treatments, as well as for the three categories of cells, the average initial volume differs by less than 10% (within the accuracy of the measure for different experiments, see Referee 1 point 1). The only exception is for treatment with NSC (the ezrin inhibitor), for which cells start with a larger volume by about 25%, and we do not have an explanation for that.

**Author response image 1. sa2fig1:** Graph showing difference in % of average initial volume of 3 groups of control cells and cells treated with drugs comparing with average initial volume of control cells.

We also plotted the degree of volume loss (in %) during spreading versus the initial volume for control cells. It showed that there is no correlation between these two measures, even for differences in volume over a factor of more than 2. Which makes the differences in initial volume that we observed between conditions (Author response image 1) irrelevant for the degree of volume loss during spreading.

We have now added these controls as Figure S1K and commented them on p 5

The authors use confinement and osmotic shocks to test model assumptions and to study cell volume changes at short time scales. However, the authors so far do not consider applying the modified PLM to volume adaption in osmotic shock conditions or potential long-term volume changes in confinement. Can the mechano-sensitive PLM also capture volume adaption in osmotic shock conditions or what are the potential limitations? In addition, do cells in confinement show a volume adaptation over time that would follow similar principles as in spreading cells or osmotic shock experiments?

We thank the referee for these suggestions. We performed the osmotic shock and confinement experiments to characterize the (almost) purely osmotic and mechanical responses and we used parameters that we extracted from the osmotic shock experiments to compute membrane permeability. Our model aims at understanding, and is limited to, the timescales at which there is an overlap between these two types of responses. The timescale of volume loss due to confinement deformation is extremely fast, less than 30 ms, while the adaptation to osmotic shocks take several tens of minutes (the PLM operates in the range of minutes, which corresponds to the typical rates of ion transport). For this reason, we believe that the volume change during the confinement involves different physics than the one involved in the current model as its timescale is too fast to only involve the standard ion transport. Our model does not include fast timescales. It is a limitation of the model that is now explicitly mentioned in the text (p.14-15).

On the other hand, the referee is right that a timescale at which the model could become relevant is the adaptation/recovery from the confinement experiment. The model would suggest that, following confinement, cells could recover their volume as membrane tension decreases, in addition to steady-state growth – this would be true only if there is no loss of non-permeable osmolytes through transient pores in the membrane during confinement (e.g. amino-acids, which are a large part of these and can play a significant role in cell osmolarity).

We thus checked what was the cell increase in volume in the 20 minutes following confinement, for cells which had been confined at 20 microns (control) and at 10 microns (which induces a volume loss). These experiments are very difficult, because the height of the confinement chamber has to remain stable enough over the measurement time to allow it to serve for FXm, in the % range. Because this did not work, we used the background level of fluorescence as a proxy for the height to correct for changes in height of the confiner over the duration of the experiment. This allowed us to obtain three independent experiments for both height (in many cases the height was too unstable and experiments had to be discarded). As shown in Author response image 2, the results are consistent with the model, with more volume increase for the cells which had been confined to 10 microns. While this experiment is interesting and opens new perspectives for our model, we believe it is out of the scope of this article and will need to be repeated more times and better controlled. We thus show it just for the referee’s benefit.

**Author response image 2. sa2fig2:** Graph showing the rate of volume increase after initial volume loss induced by confinement. Each point represents the average rate of individual experiment. Average amount of cells in each experiment n=73.

Similarly, cells display a transient phase of faster growth following the spreading phase, which, according to our model, corresponds to the combination of growth and relaxation of membrane tension leading to a volume recovery phase. This can be seen when recording cells after spreading for a longer time. However, this belongs to another ongoing study in the lab, that we do not wish to include here.

From the theoretical point of view, while our model could potentially capture some aspects of the osmotic and confinement experiments (the recovery phase for example), extending it would require to introduce elements which have not been investigated in this study and is thus beyond the scope of this work. The osmotic shock experiments do however provide us with valuable information needed to support various assumptions we make in the model, which we detail below.

From the osmotic shock experiment, we find that there are two timescales in the volume dynamics. At fast timescales of seconds, there is a passive influx of water, this allows us to take the water transport at a steady state during the slow-spreading regime. At a slower timescale, there is volume recovery due to regulatory volume increase or decrease. This has been studied previously in the literature and these regulatory volume changes are associated with the regulation of different ion co-transporters like NKCC, KCl, and NHE. The model presented in this work could be easily extended to include these co-transporters, but that would involve many more parameters (see our response to Reviewer#1 point 3). We now provide a formal description of what such an extended model would look like in the Supplementary file (p 3). From our model, we can ask if a purely mechano-osmotic coupling could be sufficient to give a volume recovery upon osmotic shock. The answer to that question is that the mechano-osmotic coupling will affect the volume dynamics after the shock, but it is not sufficient to give a volume recovery at long timescales. This is because at long times the membrane tension goes back to its homeostatic value and hence the ion transport parameters are the same before the shock and a long time after the shock. However, since the osmolarity of the external medium is different even at long time the volume will not go back to the value before the shock. The mechano-osmotic coupling will indeed affect the volume recovery dynamics, but not the asymptotic value of the volume.

In summary, the volume change at different timescale is possibly regulated by different mechanisms, with some overlapping regimes, and our model focuses on the case of cell spreading.

The authors argue for a role of myosin II in counteracting volume loss in spreading cells. Would this effect also be relevant in confinement as the authors previously showed a mechanosensitive activation of myosin II in confined cells? Does volume loss occur similarly when myosin II activity is inhibited? Do the authors expect a similar role of myosin II in regulating volume change in confinement and volume adaptation in osmotic shock conditions and spreading cells?

We first want to clarify that we do not think that myosin II (or actin cortex contractility) regulates cell volume directly, as explained in the answer to referee 1 point 2. What our experiments suggest is that reducing contractility has an effect on volume during cell spreading (and not on non-spreading cells) because it accelerates the spreading speed. So myosin II is effectively counteracting volume loss in spreading cells via its effect on reducing the spreading speed. As a consequence, there is no reason for myosin II to have an effect on volume modulation under confinement, as it has no direct effect on volume (see answer to referee 1, point 2). To test it experimentally, we performed confinement with control cells and cells incubated with Y-27632 30 min prior to the experiments. We found that there is no statistically significant difference between control and Y-27632 treated cells.

**Author response image 3. sa2fig3:** Graph showing average relative volume of control and Y-27 treated cells under different confinement heights. Each point is average relative volume of N=14-30 (average amount of cells n=165) experiments and N=4-5 (average amount of cells n=166) for Y-27 treated cells. The results of t-test for each height: h=20 µm p=0.13, h=11 µm p=0.41, h=5 µm p=0.29.

It is now discussed on p 14-15.

Similarly, there is no reason (from our model perspective) for Y-27 treated cells to display a different volume adaptation following osmotic shocks. We already showed that even disrupting the actin cortex with Lat A does not significantly affect the response to osmotic shocks. Because we had already a lot of experiments to perform, we chose to not address this point experimentally (which would have required performing an entire set of osmotic shock experiments in the presence of Y-27632).

The authors explain the Gadolinium effect by referring to the fact that calcium increase might activate myosin II. Direct measurements such as MLCK treatment or different calcium interference approaches would allow to support this argument directly.

As suggested by the referee, we performed cell spreading experiments in the presence of ML-7 treatment (MLCK inhibitor). We 25 µM as was suggested in ^23^.We did not see any effect on average cell volume and spreading upon this treatment:

**Author response image 4. sa2fig4:** Graph showing average relative volume and average spreading area of control and treated with 25 µM ML-7 cells (N=3, n=25).

Even when using 100 mM, we were not able to mimic the effect on spreading speed of Y27632 (or Gadolinium). Cell showed actually a slightly slower spreading speed, maybe due to a toxic effect of the drug at that dose. Consistently, initial volume loss was not increased compared to controls, but the cell did not recover and kept loosing volume, again maybe due to a toxic effect at this dose, since we did not observe that at lower doses.

Because these results are hard to interpret, since we have no positive control for the action of ML-7 at low doses, and high doses have little effect on spreading speed and potentially some off target effects, we cannot really interpret them and we chose to leave them out of the article.

What we wanted to test with the Gadolinium treatment was the involvement of Piezo channels, because they are classical mechanosensors. Our results show clearly that the inhibition of these channels lead to both a faster spreading speed and more volume loss, with an effect which remains on the trend defined by control cells and other treatments affecting spreading speed like Y-27632. Because we think that showing that this effect goes through the effect of Calcium on contractility (which is the most obvious from the literature), we chose to tone down our interpretation of this particular point by rephrasing our conclusion (p 11).

To complement that point, we added a novel experiment, to vary the spreading speed independently of a drug treatment. For that, we chose to add MnCl_2_, which is known to increase cell spreading^24^. The result gave a clear confirmation, with a faster spreading speed and more volume loss. This result is now added in Figure 2D and S2J and in the text (p 5).

Does NHE1 play a similar role during adaptation to osmotic stress conditions as observed in spreading cells? It would further be interesting to show whether the function of NHE1 for mechano-sensitive volume regulation is observed in the other cell types tested in this study.

We thank the reviewer for this comment. Indeed, we have not well justified our use of the NHE inhibitor EIPA. Our aim was not to directly affect the major ion pumps involved in volume regulation (which would indeed rather be the NaK exchanger), because that would likely strongly impact the initial volume of the cell and not only the volume response to spreading, making the interpretation more difficult. We based our choice on previous publication, e.g.^13^,showing that EIPA inhibited the main fast volume changes previously reported for cultured cells: it was shown to inhibit volume loss in spreading cells, as well as mitotic cell swelling ^15,25^. Using EIPA, we also found that, while the initial volume was only slightly affected, the volume loss was completely abolished even in fast spreading cells (Y-27 and EIPA combined treatment, Figure 5H). This clearly proves that the volume loss behavior can be abolished, without changing the speed of spreading, which was our main aim with this experiment.

The most direct effect of inhibiting NHE exchangers is to change the cell pH^16,17^, which, given the low number of H protons in the cell (negligible contribution to cells osmotic pressure), cannot affect the cell volume directly. A well-studied mechanism through which proton transport can have indirect effect on cell volume is through the effect of pH on ion transporters or due to the coupling between NHE and HCO3/Cl exchanger. The latter case is well studied in the literature^18^. In brief, the flux of proton out of the cell through the NHE due to Na gradient leads to an outflux of HC0_3_ and an influx of Cl. The change in Cl concentration will have an effect on the osmolarity and cell volume.

We thus performed hyperosmotic shocks with this drug and we found that, as expected, it had no effect on the immediate volume change (the Ponder’s relation), but affected the rate of volume recovery (combined with cell growth). Overall, the cells treated with EIPA showed a faster volume increase, which is what is expected if active pumping rate is reduced. This is in contrast with the above mentioned mechanism of volume regulation which will to lead to a reduced volume recovery of EIPA treated cells. This leads us to conclude that there is potentially another effect of NHE perturbation. Changing the pH will have a large impact on the functioning of many other processes, in particular, it can have an effect on ion transport^16^. Overall, the cells treated with EIPA showed a faster volume increase, which is what is expected if active pumping rate is reduced.

A full mechanistic explanation of the effect of this drug is beyond the scope of this work. Because of this we are not analyzing the effect of EIPA on the model parameter α in detail. We now clarified our interpretation of these results in the main text of the article (p 11, Figure 3D and S5B).

**Author response image 5. sa2fig5:** Graphs showing Ponder’s relation of control cells (the line indicated linear regression for control cells); only control cells, linear regression y=0. 67+0.33, R^2^=0.98 (left); only EIPA treated cells, linear regression y=0.71+0.32, R^2^=0.96 (right).

**Author response image 6. sa2fig6:** Graph showing the effect on RVI and RVD. We plotted average volume at 30 min after osmotic shock for control and EIPA treated cells (where P/Piso=0.61 and 0.72 are hypoosmotic shock). EIPA treatment, in a majority of cases, increased the cell volume gained during adaptation..

In conclusion, EIPA treatment tends to increase the rate of cell volume increase in all the conditions tested, which is consistent with a reduction of the pumping rate of ion pumps (Now shown in Figure S5B and p 11).

To answer the second part of the referee’s comment, we tested the role of EIPA and EIPA+Y-27 treatment on RPE-1 cells during spreading. We found that both EIPA and EIPA+Y-27 treated cells lost less volume than control, while their spreading speed was not significantly affected, which is coherent with our results for HeLa cells.

**Author response image 7. sa2fig7:** Graph the relation between dV/dt and dA/dt during lamellipodia formation.

These results are now shown as Figure S5C; p 11.

Arp2/3 is identified as a molecular regulator of cell volume in spreading cells and the authors speculate on its role in regulating ion channel/pump activity. The authors could further include a discussion on alternative functions of Arp2/3 such as regulating endo/exocytosis relevant for cell volume regulation.The discussion could further benefit from adding further arguments on the physiological relevance of cell volume regulation, considering the observed range of volume change and the focus on the specific mechanical stress conditions studied in this work.

We thank the reviewer for this suggestion. We have added a sentence on endo/exocytosis and Arp2/3 (p 13). Cell spreading was shown to be accompanied by an increase in exocytosis, providing more surface area to the plasma membrane and reducing membrane tension^26^. From previous literature, this is supposed to be triggered when the cell is already significantly spread, so probably later in the spreading process than the cell volume loss we are reporting here, which happens readily in the first minutes of spreading. In addition, the volumes related to endo-exocytosis events are very small compared to the volume changes considered here, so these processes are unlikely to be contributing significantly to the volume change in fast spreading cells. We also now suggest in the discussion that it might be interesting to investigate how affecting endo/exocytosis changes volume modulation during spreading, as it might contribute indirectly by modulating membrane tension (p 13).

In terms of physiological relevance, we propose a potential contribution to membrane tension homeostasis. Nevertheless, more generally, what we think our work reveals is the existence of a fundamental mechanism, which we call mechano-osmotic coupling, probably constantly at work in cells and coupling their volume change to their mechanical state. This means that it could introduce noise in the volume growth rate^27^ while not affecting their growth in mass (see Figure S1G). What our work suggests is thus that cell shape changes might be a source of density fluctuations. It is known that cells maintain a tight regulation of their global density, because it is a central parameter that can affect all the cell biochemistry (there is an ongoing work in our lab proving that point and showing how cells cope with that, and maintain density homeostasis despite fluctuations produced by cell shape changes). We discuss this aspect in the discussion of our article (p 14).

Statistical testing, the authors present N=1 in multiple panels of the presented data and should re-check the statistical significance of these results. Including p-values directly in the graphs would ease the reading of figure panels.

We apologize for the missing information. In most cases, experiments shown with N=1 in the main figure were representative examples, with further quantification. This was maybe not well explained and has been corrected (for most of the experiments with N=1 additional experiments were in supplementary dataset). In cases in which the number of shown experiments was really N=1, repeats are now added to the article. The changes are shown in Author response table 1. There are only few experiments left with N=1, which are more there for illustration purposes but have results not directly relevant for the core of the article. They could be removed if the referee feels that they are superfluous.

**Author response table 1. sa2table1:** 

Old panel	New panel	Statistics improvement
1A	1A	It was an error in the legend, actual N=3
1E	1E	There were 2 more experiments in the attached datasets. During the revision 2 additional experiment were done. N=5
3C-D,4A-C	3C-D,4A-C	The osmolarity of control medium and osmotic shock solution differ from experiment to experiment. Thus instead of averaging of experiments with similar osmolarity prior and after shock we plotted them as individual points representing the average of each individual experiment at the panels 3D, 4B and 4C. These panels also represent the total amount of osmotic shock experiment performed in this study. For the experiment with 30 s time lapse the total amount of experiments is: control N=28, Lat A N=4, EIPA N=8. For the experiment with 100 ms time lapse the total amount of experiments is: control N=11, N=5.Panels 3C and 4A are the representative examples.
5D	7B	The actual confinement height differs from experiment to experiment. Thus instead of averaging of experiments with similar heights we plotted them as individual points representing the average of each individual experiment. The total amount of experiments is N=9.
5E	7C	No improvements of statistics due to the limited revision time.
5F, 5I	7D, 7G	No improvements of statistics due to the limited revision time.
S1A	S1B	There were 2 more experiments in the attached datasets. We merged it with the experiment showed on the panel. N=3.
S1B	S1A	There were 2 more experiments in the attached datasets. We merged it with the experiment showed on the panel. N=3.
S1E	S1D	No improvements of statistics due to the limited revision time.
S1F	S1G	No improvements of statistics due to the limited revision time.
S2A	S2B	During the revision 2 additional experiment were done. N=3
S2D		There were 2 experiments for CK-666. An additional experiment was performed. N=3.
S2E	S2D-F	There were 2 more experiments for PLL-PEG in the attached datasets. We merged it with the experiment showed on the panel. N=3 for PLL-PEG.There were 2 experiments for CK-666. An additional experiment was performed. N=3.
S3B		We removed this panel.
S3C	S3B	No improvements of statistics due to the limited revision time.
S3E	S3D	During the revision 2 additional experiment were done. N=3
S5J	S7J	There was 1 more experiment in the attached datasets. We merged it with the experiment showed on the panel. N=2.
S6A	S3G	It was an error in the legend, actual N=2
S6B-C	6A	It was an error in the legend, actual N=2. But one more experiment was analyzed and added. N=3.
S6I	7K	No improvements of statistics due to the limited revision time

We also included more results of statistical tests in the figure panels or legends when it was possible.

Propidium iodide staining could be used to monitor plasma membrane rupture in confinement, considering the high leakage/volume loss in confined cells. It also appears that the authors did not comment on the high increase in surface area in Figure 5H in the main text.

We thank the referee for this suggestion. We had in fact also thought of the possibility that large holes form in the plasma membrane, and we had performed experiments with Propidium iodide, but we could not observe any cell displaying a positive staining, except for a small percentage of dying cells which served as positive control that the labelling worked. It is possible that membrane ruptures (if it is indeed the mechanism of volume loss) are too transient and close fast enough to prevent a significant π staining. The confining and volume loss time in these experiments is below 100 ms, which means that the holes (if there are any) probably do not remain open longer than that.

**Author response image 8. sa2fig8:** Graph showing percentage of cells with positive propidium iodide staining for different confinement heights (for each height N=2-3, average number of cells in each experiment n=196)*.*</Author response image 8 title/legend>.

There might have been a misunderstanding on the meaning of the surface area increase shown in Figure 5H (now Figure 7F). It is written in the main text: “Below 5 μm height, the cell surface significantly increased, which also corresponded to the formation of large blebs”, but it might be confusing and we changed it into “Below 5 μm height, the cell apparent surface area significantly increased due to the formation of large blebs”. The surface area increase is mostly the surface of blebs. Blebs are forming when the plasma membrane detaches from the underlying cell cortex, this releases a lot of membrane wrinkles which constitute most of the bleb surface area, as shown by others^28^. It is thus not surprising that the apparent surface area increases when blebs form. This is why we did not comment it more.

LatA treatments are often used at concentrations which only partially depolymerize the cell cortex but concentrations vary and could be more clearly commented in the main text. Would higher LatA concentrations increase cell spreading in confinement and alter the cell volume change?

For cell spreading experiments, we used a low concentration (100 nM), otherwise cells would not spread at all. The data show that for this concentration, the cells spread more slowly and accordingly loose less volume. For the confinement, we used much higher doses (2 and 5 µM), which disrupted strongly the cell cortex, leading to formation of large blebs even prior to confinement. In this case the cells did not lose any volume at all even when strongly confined. Our interpretation is that the loss of cortical actin, because of the formation of blebs and the release of membrane folds, made a large amount of free membrane available and thus membrane was not tensed and the volume remained constant. We do not think that a further increase in Lat A concentration would give a different result.

Related to Figure 1G, the authors state "causing a transient density increase". Panel S1G however indicates an increased density with a stable plateau, can the authors clarify this statement?

This is because the data presented here do not run for a long enough time. The density increases because the volume drops but mass keeps increasing at a normal rate. This density increase is eventually resorbed, but over a very long time (several hours), thanks to an adaptation of the mass growth rate. This is a process that we are currently studying in details, but it is out of the scope of this study.

The authors state that "with a loss of volume reaching up to 20% for fast-spreading cells" and "actin-dependent loss of volume (up to 30%)". The max volume loss of cells is not clearly visible in the current graphs and appears exaggerated in the text. The authors could include individual data points or change to a different data illustration to visualize the range of cell volume loss better.

We apologize for this and we now show graphs which display more clearly the end of the distribution (now Figure S2A).

The authors could list the specific cell types tested in the study in various statements in the main text that appear rather generic otherwise.

We apologize if it was not clear. We went through the text and tried to fix it when it was unclear which cells had been used. The figure legends have also been checked.

How does tau_a versus tau in the table relate to tau_eff and tau in the main text? Providing a clearer introduction and overview of model parameters would support the readability.

The model parameter \tau is defined in the main text above Equation 2, it is the viscoelastic relaxation timescale of tension. The experimental parameter \tau_a is the timescale of cell spreading, which is obtained by fitting the cell speeding curve with an exponentially saturating function. The model parameter \tau_eff is defined in the main text below Equation 4. This is an effective relaxation timescale for volume that depends on \tau, volume, and contact area. The difference between \tau and \tau_eff stems from the complicated geometrical relationship between contact area and total area in a spreading cell. The two fitting parameters A0 and tau_a are obtained by fitting the contact area. And the two parameters \xi and \tau are obtained from the volume dynamics.

Comments on figure panels:Figure 1A, cells are not well visible in this figure panel and a higher magnification or zoom in could be used.

Done.

Figure 1B, indicate channels in legend and check for consistency in figure panel.

Done.

Figure 1C, for the micropattern data indicate spreading time relevant for volume calculation.

The time for volume measurements on micropatterns was 4h after plating, the same as we used for measuring volume in cell population. It is indicated in the legend and Materials and methods now.

The authors also argue that they use micropatterns to reduce spreading area to a defined size compared to control conditions. Comparing Figure 1B and 1A, the cell areas are however similar and control cell spreading area appears to be higher in panel 1B versus 1A?

We apologize for the confusion. We now say more clearly in the text that these are different cell lines: HeLa EMBL control cells (also called HeLa Kyoto) for patterns and most of the experiments in the paper and a sub clone of HeLa cells expressing hgem-mcherry which turn out to have slightly different characteristics although they also are HeLa cells.

Figure 1I, J, indicate in the legend at which time point dV/dt was measured.

Done.

How was the scale bar in panel 1I calculated?

Color bar represents Kernel density. Kernel density estimation (KDE) is a non-parametric way to estimate the probability density function of a random variable. It was calculated with Python scipy.stats.gaussian_kde(). https://github.com/scipy/scipy/blob/v1.7.1/scipy/stats/kde.py#L41-L637

We replaced this color code now by 3 groups of control cells as was suggested by the Reviewer 3.

Figure 2C, there appears to be a gap in the data of Y-27632?

The referee is right. This is due to the averaging of several experiments with an outlier which pulled the average down and the corresponding data point was removed. The full graph is shown in Author response image 9. We also described the outlier removal procedure in the Material and Method and the Transparent report.

**Author response image 9. sa2fig9:** 

Figure 3A, the explanation of the schematic of the PLM could be explained in more detail to ease the reading.

We tried to improve it.

Figure 3D vs 4B, why is the Ponders equation in panel 3D independent of actin but not in panel 4B? LatA data in panel 4B appear to be not commented in main text.

We are not sure we understand this comment of the referee. Figure 4B does not correspond to the Ponder’s relation, but to the speed of volume change during the initial passive response to the osmotic shock. The referee probably thinks that there is an effect of Lat A because the points for Lat A do not cover all the P/Piso, but only a small range around 1. On the contrary, they show that there is no effect. To make it clearer, Lat A points have been removed from the panel 4B.

Figure 4B, check sign of dV/dt compared to other figure panels as it appears neg/pos for hypo/hyper conditions when cells swell/shrink.

Done. We checked but we have not detected an issue. Note that the x axis in Figure 4B is P/Piso meaning above 1 when there is an hyper osmotic shock (thus volume decreases and dv/dt is negative, as can be seen on the graph).

Figure 4C, over which duration was adaptation measured?

We apologize for the unclear sentence in Materials and methods: “Adaptation speed for osmotic shock recovery was calculated as a linear slope at 5 min intervals.” We changed it to make it clearer: “Adaptation speed for osmotic shock recovery was calculated as a linear slope starting from the minimum or maximum volume value achieved during passive response (for hyper or hypoosmotic shock) on 5 min interval”

Figure 5E, the legend indicates that a double expression line of myosin II/actin was used but only one channel is shown (membrane?). The visualization of myosin II and actin would be informative to include.

We used this cell line, but our aim was mostly to visualize the membrane in this particular experiment, which was labelled with CellMask Far Red (this is what was imaged and what is shown). We have removed the mention that the cell actually expresses the other markers, since we do not use them.

Figure 6K, indicating measured tension values in Figure 6K or matching the color code between panel 6J and 6K would ease the reading of figures.

Done – note that when data were shown before in another graph, we made them appear in grey, to make sure it is clear that they are the same dataset and not a new one for the particular panel

Figure S2G Why is the correlation before and after lamellopodia changing slope, appears to be not commented and the figure is not cited in the main text.

We observed that RPE1 cells display an initial short phase during which volume increases, which is dependent on Myosin II contractility (it disappears in Y-27632 treated cells). It seems that the extent of this initial small increase in volume also depends on the speed of the very initial spreading (in the first ~4 minutes). This is what is shown on the graph. Once the cell starts to spread and extend a lamellipodia, the volume decreases like in HeLa cells. We have no explanation for this initial bump in volume in RPE1 cells. We now say that more explicitly in the text (p 5).

Figure S2F, what do the lines indicate?

These are linear regressions. It is now mentioned in the legend

Reviewer #3 (Recommendations for the authors):It would be good to insert figure S4H in main figure, as it clearly how speed of deformation affects the volume change.

There is no Figure S4H, the referee is probably mentioning another panel but we could not identify which one it is. In general, we have tried to rethink which panels should be shown in the main figures and which should be placed in supplementary figures, to make it more clear that the focus of the article is on cell spreading. If the referee mentioned the panel S6I we now replaced it with the Model Figure 2 and it is the Figure 7K now.

The absence of correlation between volume and spreading area at a given spreading rate is interesting and new. It is probably worth discuss the authors' results in light of Ref 11, that conclude on a weak correlation between spread area and volume, and ref 9 that observe a strong correlation between area/volume.

We thank the reviewer for this comment, and we agree that this is actually the central result and message of our article. We apologize if it was not clear enough; we have now better highlighted this result by rewriting extensively some parts of the article.

Fig1FCould the authors consider including the spreading area change in time on PLL-PEG?

The spreading area that we measure in the article is more precisely the contact area measured by RICM. In the case of PLL-PEG there is no detectable contact area in RICM. The cells ‘float’ over the substrate, they cannot attach on it, at least in the duration of the experiments.

Fig1D-HColor code is not consistent and make the figure harder to read.

We tried our best to improve that point.

Page 5, first paragraphPlease explain which criteria you used to pool the cells according to their spreading speed: do you have 3 clear populations, well separated by spreading speed, or you had to apply cutoffs sort cells in different populations?

The cells were simply separated in three groups of equal sizes. This is now better illustrated showing the groups on the speed distribution. With this grouping, we still had enough cells in each group (~ 40 cells) and enough groups to argue some correlation between spreading speed and volume.

Could the authors indicate the three corresponding populations ('slow spreading', 'moderate spreading', 'fast spreading') on Figure 1I?

We thank the referee for the suggestion, it is actually more useful than the color code we had used originally, which was not very informative.

Consider clarifying the color code.

We have replaced the color code (which was giving the local density of points) and was not very useful, by the three groups of cells, as suggested by the referee.

Can the authors plot Figure 1J for each population ('slow spreading', 'moderate spreading', 'fast spreading')? Is the same tendency still observed?

The tendency is observed within each group, as shown on the graphs in Author response image 10, with of course less range of spreading speed for slower spreading cells. In all cases cells tend to spread faster at the beginning and loose more volume, then slow down and stop losing volume. We do not feel the graphs are necessary to add to the article, but they can be added if the referee thinks they are useful.

**Author response image 10. sa2fig10:** Graphs showing relation between average dV/dt and dA/dt for different time intervals of spreading for 3 groups of cells: fast, moderate and slow spreading.

Page 5, second paragraphIn RPE-1, dynamics of spreading area (Figure S2D) and volume (Figure S2E right) for control cell and under Y-27 are very similar, while it is not the case in HeLa cells. Can the authors comment on it?

We do not have a good explanation for that. Our simplest explanation is that RPE1 cells already have a very low basal contractility, as we reported before many times. So inhibiting myosin II activity has a mild effect on spreading speed. It seems to mostly affect the very early spreading dynamics – our interpretation of that is that the initial contact of the RPE1 cells induces the classical cascade directed by integrin binding and leading to a transient initial contraction followed by actual spreading and lamellipodia extension. On the other hand, HeLa cells have a high basal contractility level and do not display this initial contraction. They immediately spread, but with a high basal contractility which limits their spreading speed. To test these hypothesis, we would need to study in details the process of spreading in these two cells lines and the state of the acto-myosin cytoskeleton. It would be very interesting but is out of the scope of this study, which focuses on the coupling between spreading and volume changes.

Figure 2B-CAxes do not have the same minima/maxima so complicated to read and to compare with Figure 1H.

Note that in 1H, the goal is the comparison, in control cells, between different spreading speeds, while 2B shows the effect of Y-27 which produces faster spreading than the fastest spreading cells in the controls. All these are then placed together on a single graph for comparison in Figure 5K.

Figure 2DAxis do not have the same minima/maxima. It would be easier to read and compare with Figure 1I if they had the same range.

This is the same point as above. If we use the same range, all the points in figure 1J will be pushed together, it will not help the reading.

Are the linear fits in fig2D and fig1I giving the same results? They seemed to be different.

This is a good point from the referee. They were not the same. We have now changed that in all figures reporting this dV/dt versus dA/dt to always use the fit taken from the control cells. It shows that the Y27632 treated cells loose a little bit more volume than what control cells would at that speed of spreading if the relation was purely linear on the whole range of speeds. This could be due to an additional effect of Y27/contractility, but it is a small one and the point falls quite close to the control trend.

Page 5, third paragraphThe results of this part are very similar to the paper cited above (https://doi.org/10.1101/2021.01.22.427801), probably worth mentioning.

This is indeed true, since these are almost the same experiments (volume is just measured in a different way). We have now added the citation and mentioned the similarity (p6).

In the main text, authors should specify that you used PEG400 to apply hyperosmotic shocks.

Done (p 6)

Do the authors find differences if they apply osmotic shocks with salts or adding non-metabolized sugars such as sucrose or sorbitol?

We chose not to do this set of experiments, for the sake of time, because the central message of the article is not on the osmotic response. These experiments were done to characterize the response of the cells, to support some hypothesis of the model. However, it is well accepted in the literature, that Ponder’s relation is not affected by the choice of osmotic agent^29^.

Page 6, second paragraphUnder Y27 treatment, cell spread faster and their membrane tension measured using AFM is 30% higher than control 30 min after seeding. As LatA treated cells spread slower, wouldn't the authors expect a smaller membrane tension? Dai and Sheetz-1999 looked at the effect of cytochalasin D and indeed observed a reduced membrane tension.

The referee is right. However, we have not included that condition. As the referee states, this is an expected result, and given the number of conditions already included, we thought that it would not add much.

Page 8What are the differences, beside the timescale, between Figure 3C and Figure 4A? It would be worth mentioning the way these different were acquired in the figure legend.

The only difference is the time-lapse of the acquisition. In 3C, images were acquired every 30 second over 60 minutes, while in 4A they were acquired every 100 ms over ~7 seconds. This is now mentioned in the figure legends.

Page 9 last paragraph + page 10 first paragraphIs intensity of CellMask a proper tool to extrapolate on membrane folds and reservoirs (Figure 5E)? The spread around the contour line seems larger for 7.5um than for no confinement for control cells, and the reverse for Lat A treated cells. I am not sure how these intensity plot can be quantitatively linked to the size of the membrane reservoir.

We believe that CellMask dyes are a good proxy to quantify the amount of membrane. The spread around the contour line is actually shown quantitatively on the right graph next to the images. And it is very clear from the graph that the width at mid-height is not more for 7.5 than controls – it is mostly given by the optical resolution. We now give the non-confined. What matters is the integrated intensity. These dies contain a lipophilic moiety for membrane loading and a negatively charged hydrophilic dye for “anchoring” of the probe in the plasma membrane. These amphipathic dyes provide a uniform staining of the plasma membrane. While they cannot be used to assess the presence of nanofolds or domains, they can provide bulk information on the surface area of the plasma membrane in contact with the extracellular medium, and they stain large folds and membrane ruffles. Depletion of membrane ruffles and folds would result in less CellMask ^30–32^. In this article the drop of CellMask intensity at the plasma membrane serves an an indication of depletion of fold, ruffles and small filopodia. In agreement with this, in cells stained with CellMask, blebs display a thinner and dimmer CellMask signal compared to the cortex-bound membrane – this is because blebs have a bare flat membrane and thus contain less plasma membrane reservoirs than the cortex-bound^28,33^. We thus think that our measure, although maybe not perfectly quantitative for the amount of membrane, can be used to assess the difference in the amount of membrane folds between conditions.

Also, in the before to last paragraph in page 9, I kindly disagree with the statement that "membrane tension could not arise from a limitation in membrane availability". Isn't it that unfolding the membrane partially, not reaching the full size of the membrane reservoir, would still change membrane tension, because of opposing cytoskeletal and adhesive forces trying to fold the membrane?

We agree with the referee and we apologize for the sentence which was not clear enough. What we mean is that the increase in membrane tension could not arise from a limitation in the total amount of membrane, since there is a very large reservoir. And indeed, it increases due to the phenomenon mentioned by the referee. We modified the sentence accordingly (p 14)

Figure 5FLatA treated cell keep a constant volume after confinement while control cell decrease volume. Since you have a condition (Y27 treated cell) that have opposing effect of LatA on spreading area and volume, I wonder what would be the effect of confinement on Y27 treated cells. Would cells 'overreact?'

The condition of Y27 treatment does not have a direct effect on volume in the cell spreading experiments, but it has an effect only via the increase in spreading speed. We have now added experiments to prove that more directly (see Referee 1 point 2). As a consequence, it should not have any effect in the confinement experiments, and this is what we find experimentally (see Referee 1 point 2)

Figure 6 A-HY-axis scale is different for each graph, would be better to fit one range and keep it so that the reader can compare from one to another.

Because the effects of the various drugs have different amplitude, it is not really possible to use the same scales for all these graphs. The comparison can be made on the panel K, where all the values are placed together for that purpose.

Figure 6JDo you affect absolute cell volume when adding those drugs?

Because the effects of the various drugs have different amplitude, it is not really possible to use the same scales for all these graphs. The comparison can be made on the panel K, where all the values are placed together for that purpose.

[Editors' note: further revisions were suggested prior to acceptance, as described below.]

Essential revisions:We all agreed that the revised version of the manuscript is much improved and clearly presents a very interesting and compelling story, well done!Before the formal acceptance of the manuscript, one of the reviewers (rev. #1) made two final comments regarding (1) the contractile stress's role in volume control; and (2) the relative importance of membrane tension and voltage for the effects presented here. I think it might be a good idea to consider these comments and decide whether you think you can adapt the discussion part of the text so your points are clearly laid down.Reviewer #1 (Recommendations for the authors):The authors have addressed my comments and the manuscript is much improved. I can recommend publication. I think there is generally no disagreement in our thinking of contractility and cell volume regulation. It is true that contractile force, or pressure, alone cannot really change the volume very much. This was already shown in Yellin et al., Biophys. J. 2018, and Li and Sun, Frontiers Cell Dev. Bio., 2022, and I agree with the authors' general thinking. However, the discussion in Pg. 7, I think, can be improved. Right now, it sounds like contractile stress is being compared with the total osmotic pressure of impermeable solutes inside the cell. But in reality, the contractile stress should be compared with the total osmotic pressure difference between inside and outside of the cell, which is much smaller than 100mM. During 10% volume change during cell spreading, I think since it's happening slowly (order of minutes), the ion flux and contractile stress should change simultaneously, possibly with contractile stress changing slowly, while ion flux happening very fast. But, the contractile stress must change, since at minimum, the cell is changing shape, and therefore cortical stress must change to maintain relative force balance. I'm still not completely sure whether is ok to neglect it, since the steady-state result of the proposed model is zero osmotic pressure difference between inside and outside, but in reality, there is about 1kPa of hydraulic pressure difference and 1kPa osmotic pressure (~0.5mM) difference. This difference may be small enough to be negligible in the current context. But it may not be small in other contexts, such as during cell motility.

We have added the references that the referee mentioned, which support our argument that hydrostatic pressure has a negligible role in volume changes that are more than 1%. We have also emphasised the importance of membrane and cortical tension in other cellular phenomena such as shape change, which are not considered here since the spreading rate is a measured input of the model (p.7 2nd paragraph). We are not sure to understand the referee’s point about the significance of a 1kPa osmotic pressure difference in the context of cell motility. The motility data reported in our paper (for a particular cell type – dendritic cells – migrating in collagen) shows volume changes of order 10-20% (Figure 7), which cannot be explained by a 1% modulation of the osmotic pressure difference.

The other issue I hope the authors can clarify in the discussion is the equal importance of membrane voltage and membrane tension in this story. Ion exchangers such as NHE and NaK are affected by both. The current dependence of exchanger flux with respect to voltage is based on just passing an ion through a constant electric field. This is generally not true for ion exchangers. I think most studies of ion exchangers show a voltage gating effect. This, after all, is the basis of action potentials. Therefore, these important facts should be clearly stated.

We certainly agree with the referee that voltage-gating is very important for many type of channels and exchanger. We now more clearly state the assumptions behind our estimate of the coupling parameter \α, and their limitations (p.9 below Equation1). However, we are unsure about what the referee means when he mentions the "the equal importance of membrane voltage and membrane tension in this story”. In our mode, the ion concentrations and the potential difference are dynamical variables of the PLM. The only coupling between the PLM and cell spreading is through membrane tension. Changes of tension during spreading will certainly affect the transmembrane potential, which in turn affects the rates of ion transport, but this is capture by Equation1 at the linear level. we have now made this more explicit in the text (p.9 below Equation1).